# OptiVer: Unleashing the Power of LLMs for Optimization Modeling via Dual-Side Verification

## Abstract

Building mathematical optimization models is critical in operations research (OR), while it requires substantial human expertise. Recent advancements have utilized large language models (LLMs) to automate this modeling process. However, existing works often struggle to verify the correctness of the generated optimization models, without checking the rationality of the constraints and variables or the validity of solutions to the generated models. This hampers the subsequent verification and correction steps, and thus it severely hurts the modeling accuracy. To address this challenge, we propose a novel LLM-based framework with Dual-side Verification (OptiVer) from both structure and solution perspectives, thereby improving the modeling accuracy. The structure-side verification ensures that the modeling structure of the generated optimization models aligns with the original problem description, accurately capturing the problem's constraints and requirements. Meanwhile, the solution-side verification interprets and evaluates the validity of the solutions, confirming that the optimization models are logically and mathematically sound. Extensive experiments on several popular benchmarks demonstrate that our approach significantly outperforms the state-of-the-art, achieving over 20% improvement in accuracy.

## 1 Introduction

Optimization problems are foundational to operations research (OR), with wide-ranging applications in manufacturing (Jayal et al., 2010), transportation (Yin, 2002), and service industries (Berman et al., 1994). In practice, OR problem statements are typically specified in natural language. Practitioners must therefore (i) translate these descriptions into an appropriate mathematical optimization model (defining objectives, decision variables, and constraints) and (ii) implement the solver code (e.g., SCIP (Achterberg, 2009), Gurobi (Gurobi Optimization, 2021), or Pyomo (Bynum et al., 2021; Hart et al., 2011)) to obtain solutions. This workflow is labor-intensive, demands substantial domain expertise in problem context, mathematical modeling, and code-level implementation or debugging, and is consequently costly and time-consuming (Ahmaditeshnizi et al., 2024).

Given the impressive capabilities of large language models (LLMs) in natural-language understanding and domain knowledge acquisition, a growing number of works have employed LLMs to automate the processes of modeling, programming, and debugging. Existing approaches can be broadly grouped into two categories. The first category, prompt-based methods, relies on pre-trained LLMs (e.g., GPT-4 (OpenAI, 2023) and GPT-4o (OpenAI, 2024)), which are prompted to construct mathematical models incrementally. In practice, these methods are often implemented into a carefully designed framework, such as multi-agent cooperation (Xiao et al., 2024; Ahmaditeshnizi et al., 2024) and Monte Carlo tree search (Astorga et al., 2025). The second category enhances modeling capabilities through fine-tuning, which involves constructing a large, labeled dataset for LLM training (Huang et al., 2025; Wu et al., 2025; Jiang et al., 2025; Chen et al., 2025; Lu et al., 2025).

Beyond these two approaches, recent research has explored self-correction strategies to improve the modeling performance (Jiang et al., 2025; Xiao et al., 2024; Ahmaditeshnizi et al., 2024). These methods trigger correction primarily from error messages produced during code execution. However, such strategies confine self-correction to code-level issues, and the underlying model can remain

flawed even when the code runs without failure. To develop more effective model verification methods, we characterizes incorrect models by the following features. First, incorrect models often overlook indispensable constraints that the textual problem description does not explicitly state. For example, when formulating a maximum flow problem, an LLM might omit flow-balance constraints at intermediate nodes simply because they are not explicitly mentioned, yielding a structurally incomplete model. Second, incorrect models can yield solutions that are infeasible or violate basic logical principles, even though the solver executes successfully and reports an improved objective.

To address these challenges, we propose a novel multi-agent framework with Dual-side Verification (OptiVer) from both the modeling structure and solution perspectives, improving the modeling accuracy. This approach moves beyond simple code-execution signals by translating both the optimization model and its resulting solutions into natural language and evaluating their semantic correctness. To the end, OptiVer introduces two novel evaluation metrics for self-correction: modeling structure consistency and solution validity. (1) **Consistency in the modeling structure** ensures that the mathematical formulation (its variables, constraints, and objective) is a complete and faithful translation of the original problem description. To evaluate this metric, one LLM agent performs a back-translation that abstracts the generated model into a compact, multi-level description of its components. A second agent then aligns this abstraction with the structure derived from the original specification to reveal omissions or mismatches. (2) **Solution validity** assesses whether the solution of the produced model is logically

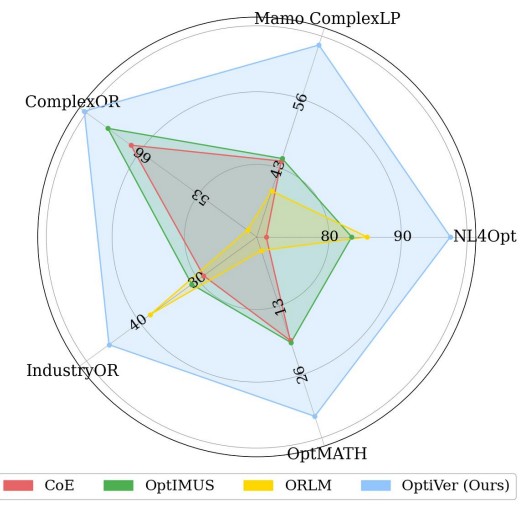

Figure 1: OptiVer outperforms other baselines in solving accuracy (SA) across the benchmarks.

and contextually sound for the real-world task. To assess it, one agent interprets the numeric solution in natural language, explaining its meaning in the context of the real-world problem. Another agent then critiques that interpretation to expose logical absurdities or mathematical violations that code-execution checks miss. Finally, we use the verification feedback for model refinement.

As illustrated in Figure 1, extensive experiments on five popular benchmarks showcase that our approach significantly outperforms the state-of-the-art, achieving an average improvement of approximately 10% in solving accuracy. Notably, OptiVer is designed as a plug-and-play framework, capable of effectively verifying and refining optimization models generated by any existing pre-trained or fine-tuned OR LLMs.

## 2 RELATED WORK

**Automated Optimization modeling** In practice, the OR problems often arise from real-world situations, which are typically described in natural language. Consequently, automated optimization modeling has emerged as a critical area aimed at reducing the labor and time costs associated with the modeling process (Chen et al., 2023; Li et al., 2023). Notable early efforts in this field include the NL4Opt competition (Ramamonjison et al., 2021). Since then, several benchmarks have been introduced to evaluate performance, such as ComplexOR (Xiao et al., 2024), NLP4LP (Ahmaditeshnizi et al., 2024), Mamo (Huang et al., 2024), IndustryOR (Huang et al., 2025) and Optibench Yang et al. (2025); Wang et al. (2024). Recent research primarily falls into two categories: prompt-based methods and fine-tuned methods. Prompt-based approaches utilize pre-trained large language models (LLMs) with carefully crafted prompts to iteratively construct models. For instance, Chain-of-Experts (Xiao et al., 2024) and OptIMUS (Ahmaditeshnizi et al., 2024) frameworks employ multi-agent cooperation, while some other methods Astorga et al. (2025) leverage Monte Carlo tree search techniques to explore potential models. To further enhance the modeling capabilities of LLMs, researchers also work on fine-tuning these models with extensive OR and modeling knowledge (Huang et al., 2025; Wu et al., 2025; Jiang et al., 2025; Chen et al., 2025). For example, LLaMoCo (Ma et al., 2024) utilizes an instruction tuning framework to adapt LLMs for solving optimization

problems in a code-to-code manner. ORLM (Huang et al., 2025) trains open-source LLMs specifically designed for optimization modeling and solver code development. Additionally, advanced techniques such as KTO (Ethayarajh et al., 2024) and data augmentation have been introduced to improve model training (Wu et al., 2025; Jiang et al., 2025).

## 3 MOTIVATED RESULTS AND CASE ANALYSIS

**Challenges**   We identify two key challenges in existing LLM-based optimization modeling methods. (1) LLMs struggle to identify the modeling structure within the problems, including missing or incorrect constraints and errors in variable definitions. The prevalence of such mistakes is notable in benchmarks, with 36.0% in NL4Opt and 12.6% in ComplexOR as pointed out in OptiMUS (Ahmaditeshnizi et al., 2024). (2) LLMs can only find the errors in the solver codes, but can hardly find the errors in the optimization models. OptiMUS points out that "Coding errors are easier to identify and fix. In contrast, identifying bugs in the formulation requires deeper reasoning and is harder." In existing methods, the debugging module is typically activated only when solver codes produce execution errors. To illustrate these challenges, we use GPT4o-mini in Figure 2, involving a maximum flow problem (MF).

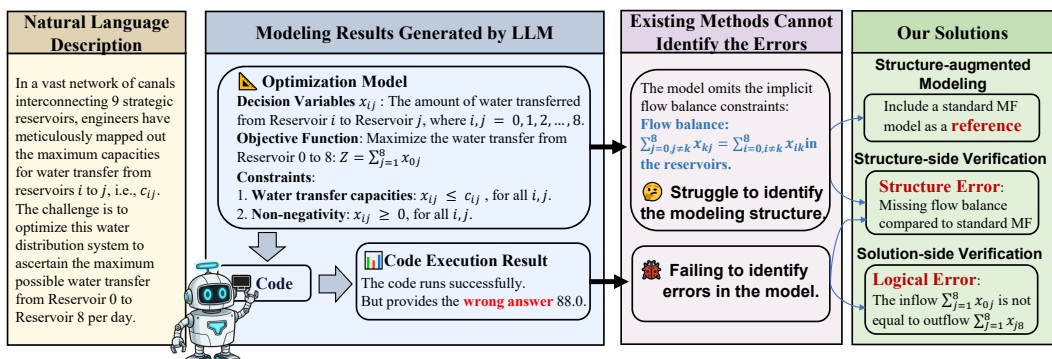

Figure 2: The two challenges we observed in existing optimization modeling methods.

**Observations on the Modeling Structures**   To specify the definition and the usage of modeling structures, we have the following observation. The LLM cannot find the flow balance constraints at first. However, the model can correctly identify the relevant problem classifications. When we prompt the model to formulate the relevant problem classification (Maximum Flow Problem in this case), it successfully identifies the flow balance constraints.

The core principle of structure-augmented modeling is to **leverage similar standard optimization models as a reference** to identify a problem's implicit constraints. In Operations Research, many problems in similar scenarios share characteristics with optimization models in conventional problem classifications, such as the Vehicle Routing Problem or the Maximum Flow Problem. These classic

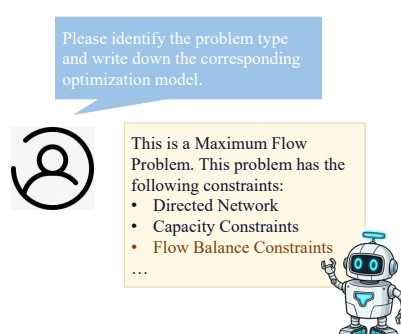

Figure 3: The influence of recalling a problem classification.

types have conventional mathematical formulations——including standard variables and assumed constraints——which this paper refers to as **modeling structures**. Even when a new problem does not neatly fit a standard problem classification, referencing the modeling structure of a similar, well-understood problem helps the LLM uncover these implicit relationships, which are often crucial for a correct formulation.

## 4 METHODOLOGY

Our work investigates how the modeling process can be enhanced through effective verification methods on both the structural and solution sides. An overview of the framework is presented in Figure 4. We define multi-level modeling structures in Section 4.1, followed by a detailed explanation of

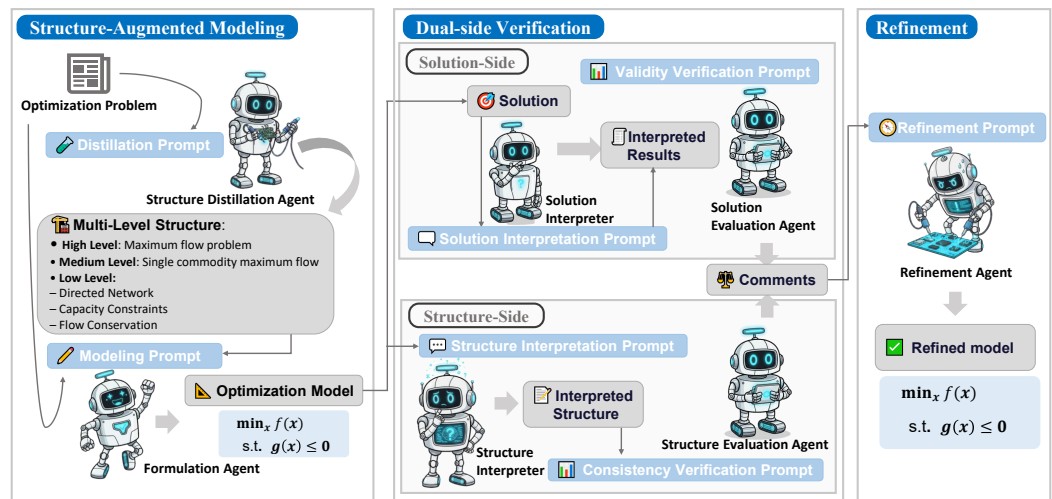

Figure 4: Our OptiVer framework begins by distilling the multi-level structures from the natural language description. These extracted structures are then combined, allowing the formulator to generate an initial model. Then, OptiVer conducts a dual-side verification and refinement process.

structure-side verification in Section 4.2 and solution-side verification using a multi-agent cooperation framework in Section 4.3. We first introduce some notations in this work as follows.

Let $\mathcal{D}$ represent the space of natural problem descriptions, and let $\mathcal{M}$ denote the model space encompassing all possible optimization models. The modeling process can be viewed as a mapping from the problem description $D \in \mathcal{D}$ to an optimization model $M \in \mathcal{M}$. The descriptions follow the distribution $p(D)$, and we explicitly define the modeling space as the Cartesian product of the variable space, constraint space, and objective function space. Let $p(M)$ be the distribution representing the optimization model. The mutual information is a widely used theoretical metric for the similarity of the distributions. The mutual information $I(D, M)$ is a canonical metric used to quantify the dependence between such distributions. In information theory, mutual information is defined as

$$I(M, D) = \sum_{M \in \mathcal{M}, D \in \mathcal{D}} p(M, D) \log \left( \frac{p(M, D)}{p(M)p(D)} \right) \tag{1}$$

A high mutual information indicates that the model accurately captures the constraints and requirements stated in the text. Conversely, information loss (low mutual information) corresponds to common modeling errors, such as missing implicit constraints or incorrect variable definitions.

## 4.1 STRUCTURE-AUGMENTED MODELING

**(1) Motivation of Multi-Level Structure: Coarse-to-fine structure Analysis** Before the modeling process, human experts first analyze the problem description to identify a similar conventional problem classification as a reference. Next, they determine the variant of the classification that best aligns with the description. Finally, they assess special requirements in the description. This analysis follows a coarse-to-fine approach, from high-level to low-level structure analysis.

**(2) Multi-Level Modeling Structure** Our framework begins by distilling the modeling structures from the natural language descriptions. As discussed in Section 3, understanding the problem type is crucial in the modeling process, as it serves as a foundational template for developing optimization models. Inspired by the coarse-to-fine structure analysis process used by human experts, we further refine the concept of modeling structures by introducing the idea of multi-level modeling structures.

- High-Level Structure: This represents the fundamental problem type within OR, such as the maximum flow problem, set covering problem, vehicle routing problem, and knapsack problem. Each of these problem types is associated with a basic optimization model.

- Medium-Level Structure: This pertains to the classical classification or variants of fundamental problem types. For instance, variations of the maximum flow problem include multi-source MF, multi-commodity MF, minimum-cost MF, and MF in undirected graphs. Each variant is associated with a specific modified optimization model derived from the basic model.

- Lower-Level Structure: This level encompasses constraints in the classical optimization model as well as specific requirements that extend beyond classical models. For instance, standard constraints might include capacities and flow balance constraints in the MF, while special requirements could involve flow capacities that fluctuate over time.

As we mentioned in Section 3, even highly complex and unique industrial problems are often variants or combinations of fundamental problem classifications recognized in operations research. The **high-level** and **medium-level structures** in our framework are designed to capture this foundational core, providing a solid starting point for the modeling process. Second, and most critically, the framework's **low-level structure** is specifically designed to provide the necessary flexibility to handle unique, real-world contexts. This level is not confined to a specific problem classification and is intended to capture the nuanced, problem-specific constraints and requirements that extend beyond classical formulations. This design allows the framework to represent the unique aspects of any given problem, rather than forcing it into a rigid, predefined category. We denote the multi-level modeling structure as $S$. Below, we provide an example of the modeling structure we have defined.

---

**Example: The Structure Schema Extracted by LLMs**

- **High Level**: Maximum flow problem
- **Medium Level**: Single commodity maximum flow
- **Low Level**:
  - **Directed Network**: The flow is directed from one reservoir to another.
  - **Capacity Constraints**: Each edge (connection between reservoirs) has a maximum capacity.
  - **Flow Conservation**: The amount of water entering any intermediate reservoir must equal the amount leaving, except for the source and sink.

---

**(3) Structure Distillation and Structure-Augmented Modeling**    We use two pre-trained LLMs (implemented by GPT4o-mini in this work) as agents to complete the structure distillation and initial modeling tasks, guided by designed prompts. To distill the multi-level modeling structure $S$ from the natural language description $D$, we introduce an LLM agent, called the structure distillation agent. The agent takes as input the problem description and outputs the formatted structure context. Then, we call a formulation agent to generate an initial optimization model $M$ guided by prompts combining the problem description and modeling structure, i.e.,

$$S = \texttt{Distillation\_Agent}(D), \quad M = \texttt{Formulation\_Agent}(D, S). \tag{2}$$

## 4.2 Structure-Side: Structure Interpretation and Consistency Verification

**(1) Motivation**    Structure-side verification finds the modeling errors by detecting any deviation from the established, correct formulation for a known class of problems, catching errors of omission where the LLM may overlook fundamental constraints and variables required for that problem classification. Inspired by dual learning in machine translation (He et al., 2016), we assert that a correct model must meet the following consistency criterion: when we translate the optimization model back into the space of modeling structure, the resulting context should semantically correspond to the modeling structure directly derived from the problem description.

**(2) Structure Interpretation and Consistency Verification**    We introduce a structure interpretation agent and an evaluation agent to complete the structure verification task. The two agents are also guided with specific prompts. First, a structural interpretation agent performs a "back-translation". It takes the generated mathematical model $M$ and converts it back into its abstract modeling structure $\tilde{S}$. Next, a structural evaluation agent acts as a critic. It compares the interpretation agent's output $\tilde{S}$ with the original structure $S$ derived from the problem description to check for semantic consistency.

The evaluation agent's output is twofold: a binary consistency score $c_c$ and a detailed comment that highlights any discrepancies. This comment provides specific, actionable feedback that is later used to refine the model. The process can be formally summarized as

$$\tilde{S} = \texttt{StruInterp\_Agent}(M), \quad (com, c_c) = \texttt{StruEval\_Agent}(S, \tilde{S}), \qquad (3)$$

where $c_c = 1$ indicates consistency, while the comment $com$ details any differences found between the structures, guiding the subsequent refinement step.

Finally, we propose an analysis of the structure-side verification using mutual information. Suppose that $\tilde{S}$ is the interpreted structure from optimization model $M$. The structure-side verification aims to improve the consistency between structures $S$ from the natural language description and that $\tilde{S}$ interpreted from the optimization model, i.e., the mutual information $I(S, \tilde{S})$.

**Proposition 4.1.** *We have $I(S, \tilde{S}) \leq I(D, M)$. Thus, the structure-side verification optimizes the lower bound of the mutual information between the problem description and the optimization model.*

The structure-side verification is improving the lower bound.

**Proposition 4.2.** *Let $\varepsilon > 0$ be a fixed threshold representing the minimum information overlap required for a model to be deemed structurally consistent. Define the Bernoulli random variable $c_c$ corresponding to the binary output of our Structure Evaluation Agent, $c_c \triangleq \mathbf{1}_{\{I(S,\tilde{S}) \geq \varepsilon\}}$, where $I(S, \tilde{S})$ denotes the mutual information between the distilled structure $S$ and the interpreted structure $\tilde{S}$. Then, the expected mutual information satisfies, $\mathbb{E}[I(S, \tilde{S})] \geq \varepsilon \mathbb{E}[c_c]$.*

### 4.3 SOLUTION-SIDE: SOLUTION INTERPRETATION AND VALIDITY VERIFICATION

**(1) Motivation** This method works because it grounds the abstract mathematical model by assessing whether its solution is logically feasible. A model may be syntactically correct and yield a numerical answer, yet that answer could violate the fundamental logic of the original problem (e.g., suggesting more water flows out of a reservoir than flows in). We argue that the semantic content of the solution itself is a far richer source for identifying errors. Solution-side verification enhances performance because it is designed to catch logical errors that are invisible to systems that only check for solver execution errors. The core of this verification is to leverage the **common-sense and logical reasoning capabilities** of such LLMs for improved error detection.

**(2) Solution Interpretation and Validity Verification** Given an optimization model $M$, OptiVer executes the solver code and obtains the optimal solution $\boldsymbol{x}$. Then, OptiVer performs solution-side verification using two LLM agents, guided by designed prompts. The first agent, a solution interpreter, translates the raw numerical solution $\boldsymbol{x}$ into a meaningful natural language description $\tilde{D}$ based on the original problem context $D$. Next, the second agent, a solution evaluation agent, scrutinizes this description to identify any logical or mathematical errors. This agent's output includes a binary validity score $c_v$ and, crucially, a detailed comment

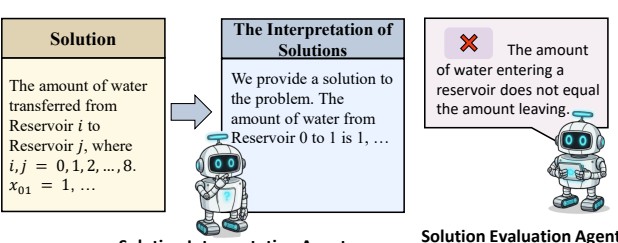

Figure 5: An example of solution verification.

$com$ that provides specific feedback on any flaws found. The score value is 1 if the evaluator recognizes the validity of solution $\boldsymbol{x}$, and 0 otherwise. This verification process can be formally summarized as:

$$\tilde{D} = \texttt{SolInterp\_Agent}(\boldsymbol{x}, D), \quad (com, c_v) = \texttt{SolEval\_Agent}(D, \tilde{D}). \qquad (4)$$

Similar to the analysis of the structure-side verification, we have the following analysis of the solution-side verification. Suppose that $\tilde{D}$ is the interpreted solution from the optimization model $M$. During the solution-side verification, we improve the mutual information $I(D, \tilde{D})$.

**Proposition 4.3.** *We have $I(D, \tilde{D}) \leq I(D, M)$. Thus, the solution-side verification optimizes the lower bound of the mutual information between the problem description and the optimization model.*

Table 1: Comparison of our method and the baselines across five popular benchmarks. Throughout the experiments, we compare the solving accuracy (SA) of the methods.

| | NL4Opt | Mamo ComplexLP | ComplexOR | IndustryOR | OptMATH |
|---|---|---|---|---|---|
| Reasoning LLMs | | | | | |
| DeepSeek-R1 | 82.6 | 67.2 | 68.4 | 32.0 | 33.1 |
| OpenAI-o1 | 87.1 | 66.3 | 68.4 | 36.0 | 32.5 |
| Fine-tuned Method | | | | | |
| ORLM | 85.1* | 38.8* | 42.1* | 38.0* | 2.6* |
| Evo-Step | 84.4* | 61.6* | - | 36.3* | - |
| LLMOPT | 80.3* | 44.1* | 72.7* | 29.0* | 12.5* |
| OptMATH | 95.9* | 54.1* | - | - | 34.9* |
| SIRL | 96.3* | 62.1* | - | 33.0* | 29.0* |
| Prompt-based Method | | | | | |
| Standard | 64.6 | 27.9 | 31.5 | 24.0 | 15.6 |
| CoT | 69.3 | 34.5 | 36.8 | 27.0 | 18.6 |
| CoE | 71.3 | 44.5 | 68.4 | 29.0 | 19.8 |
| OptiMUS | 83.0 | 45.0 | 73.6 | 31.0 | 20.2 |
| OptiVer (Ours) | **96.5** | **66.7** | **78.9** | **45.0** | 34.3 |

Values marked with * are from the original or reproduced papers. , and - are with missing data because the model has not been publicly released.

**Proposition 4.4.** *Let $\varepsilon' > 0$ be a threshold for solution validity and define $c_v \triangleq \mathbf{1}_{\{I(D,\tilde{D}) \geq \varepsilon'\}}$, representing the binary output of the Solution Evaluation Agent. Then $\mathbb{E}[I(D,\tilde{D})] \geq \varepsilon'\mathbb{E}[c_v]$.*

### 4.4 REFINEMENT

Based on the feedback, we refine the optimization model. The refinement agent within the OptiVer framework is a specialized LLM-based component responsible for correcting and enhancing the initial optimization model based on insights from the dual-side verification process. Guided by a refinement prompt, the agent takes the current formulation as input and produces a refined optimization model, represented as $M' = \text{Ref\_Agent}(D, S, M, com)$.

## 5 EXPERIMENTS

**Benchmarks** We use five real-world operations research benchmarks: NL4Opt (Ramamonjison et al., 2021), Mamo ComplexLP (Huang et al., 2024), ComplexOR (Xiao et al., 2024), IndustryOR (Huang et al., 2025) and OptMATH (Lu et al., 2025). The NL4Opt benchmark, released for the NeurIPS 2022 NL4Opt competition, consists of 289 elementary linear programming problems. Mamo ComplexLP 211 problems. ComplexOR is a comprehensive dataset including linear and mixed-integer programming. In alignment with the studies by (Ahmaditeshnizi et al., 2024) and (Jiang et al., 2025), we focus on 19 specific problems from this dataset. IndustryOR has 100 challenging problems from various industry scenarios. OptMATH has 166 challenging problems.

**Implementation and Baselines** In our experiments, we utilized the GPT4o-mini to implement the agents in our method and all the prompt-based baselines. For the implementation of OptiVer, please see Appendix F for the prompts of each agent. In our experiments, we compare OptiVer with four available prompt-based methods and five fine-tuned operations research LLMs. The four prompt-based baselines include Standard, Chain-of-Thoughts (CoT) (Wei et al., 2022), Chain-of-Experts (CoE) (Xiao et al., 2024), and OptiMUS (Ahmaditeshnizi et al., 2024). The Standard baseline represents the output of GPT without any optimization of its reasoning processes. We include five fine-tuned open-source operations research language models as baselines, including ORLM (Huang et al., 2025) (based on LLaMA-3-8B model), Evo-Step (Wu et al., 2025) (based on LLaMA-3-8B model), LLMOPT (Jiang et al., 2025) (based on Qwen1.5-14B), OptMATH (Lu et al.,

Table 2: Alation studies on (1) each component and (2) each level of modeling structures in OptiVer.

| Method | NL4Opt | Mamo ComplexLP | ComplexOR | IndustryOR |
|---|---|---|---|---|
| Ablation for the Components | | | | |
| OptiVer w/o struaug | 91.8 | 54.2 | 63.1 | 34.0 |
| OptiVer w/o stru-side | 91.5 | 55.2 | 68.4 | 29.0 |
| OptiVer w/o sol-side | 91.8 | 53.1 | 68.4 | 41.0 |
| Ablation for Each Level of the Structure | | | | |
| OptiVer w/o high | 92.9 | 59.9 | 78.9 | 41.0 |
| OptiVer w/o medium | 91.1 | 57.0 | 73.6 | 38.0 |
| OptiVer w/o low | 91.1 | 54.9 | 73.6 | 30.0 |
| **OptiVer (full)** | **96.5** | **66.7** | **78.9** | **45.0** |

2025) (based on Qwen2.5-32B), and SIRL (Chen et al., 2025) (based on Qwen2.5-7B) trained with reinforcement learning. Additionally, we also compare our results with the pre-trained reasoning model DeepSeek-R1 (DeepSeek-AI, 2025) and OpenAI-o1 (OpenAI, 2024).

**Metrics** Consistent with existing research, we employed solving accuracy (SA) to evaluate performance. Specifically, SA represents the proportion of problems for which the methods successfully identify the optimal solutions. The higher value of SA implies better performance.

## 5.1 MAIN RESULTS

To demonstrate the effectiveness of our method, we conduct experiments comparing solving accuracy (SA) between our approach and baseline methods across various benchmarks. The results presented in Table 1 indicate that our method significantly outperforms the baselines, achieving an approximate 20% improvement in solving accuracy compared to Standard. For the challenging benchmarks, our method consistently delivers outstanding performance. This demonstrates that OptiVer exhibits strong generalization capabilities across both easy and difficult scenarios. Furthermore, OptiVer achieves performance better than state-of-the-art reasoning LLMs, such as DeepSeek-R1 and OpenAI-o1, despite relying on a much weaker base model, GPT4o-mini. Please see Appendices D and E for case and error analysis.

## 5.2 ABLATION STUDIES

**(1) The Effects of Each Component of OptiVer** In this section, we examine the effects of the three components of OptiVer: structure-augmented modeling, structure-side verification, and solution-side verification. To assess their contributions, we implement three variants of OptiVer. The first variant, OptiVer w/o stru-aug, omits the introduction of a modeling structure to enhance the modeling process. For structure-side verification, instead of interpreting the model in structural terms, we instruct an LLM agent to provide a narrative explaining the meaning of the variables, constraints, and objectives. The second variant, OptiVer w/o stru-side, does not implement structure-side verification at all. The third variant, OptiVer w/o sol-side, excludes the solution-side verification process. The results, presented in Table 2, reveal a significant drop in performance in the absence of any of these components, highlighting their essential roles in the modeling process.

**(2) The Effects of Each Level of Modeling Structures** Next, we investigate the impact of each level within our proposed modeling structure. The variant OptiVer w/o high/medium/low level excludes the use of high, medium, and low-level structures. The experimental results in Table 2 demonstrate that all three levels contribute positively to overall performance, with the medium and low-level structures showing particularly pronounced improvements.

**Takeaway** Critically, the framework's "low-level structure" is specifically designed to provide the necessary flexibility to handle unique, real-world contexts. This level is not confined to a specific problem type and is intended to capture the nuanced, problem-specific constraints and requirements that extend beyond classical formulations. This design allows the framework to represent the unique aspects of any given problem, rather than forcing it into a rigid, predefined category.

Table 3: We build OptiVer on different baselines and backbone models.

| Method | NL4Opt | Mamo ComplexLP | ComplexOR | IndustryOR |
|---|---|---|---|---|
| Different Baselines | | | | |
| ORLM | 78.2 | 41.2 | 36.8 | 39.0 |
| ORLM+OptiVer | 92.3 | 59.6 | 73.6 | 42.0 |
| OptiMUS | 83.0 | 45.0 | 73.6 | 31.0 |
| OptiMUS+OptiVer | 96.1 | 61.0 | 78.9 | 45.0 |
| Different Backbones | | | | |
| GPT-4o | 79.4 | 45.0 | 57.8 | 27.0 |
| GPT-4o+OptiVer | 97.5 | 66.3 | 78.9 | 48.0 |
| Qwen2.5-14B | 70.3 | 41.2 | 57.8 | 31.0 |
| Qwen2.5-14B+OptiVer | 85.8 | 56.3 | 68.4 | 39.0 |

## 5.3 BUILDING ON DIFFERENT BASELINES AND LLMS

**(1) Improving different Baselines:** OptiVer was applied to the outputs of three foundational baselines: OptiMUS and the fine-tuned ORLM model. In each case, OptiVer's verification and refinement process enhanced the initial models generated by these baseline methods. **(2)Improving different Base LLMs:** To illustrate that the framework is not reliant on a specific backbone model, we conducted experiments using various base LLMs with OptiVer. This approach highlights how performance scales with the capabilities of the underlying model, including stronger models (e.g., GPT-4o) and weaker models (e.g., Qwen2.5-14B) The results consistently indicated significant performance gains, as shown in Table 3. Please refer to Appendix C for detailed experiment settings.

## 5.4 QUANTITY ANALYSIS OF VERIFICATIONS

**Critical Components of Verifications** The interpretation and evaluation agents are essential components of the verification process, as they determine whether OptiVer can effectively identify errors in the modeling process. We conducted extensive ablation studies to quantitatively assess the accuracy and reliability of these agents. Our experiments were specifically designed to evaluate their ability to distinguish between correct and incorrect models.

Table 4: Verification Precision

| Verification Type | Easy | Medium | Hard |
|---|---|---|---|
| Structure Verification | 92% | 89% | 83% |
| Solution Verification | 93% | 91% | 86% |

Table 5: Verification Recall

| Verification Type | Easy | Medium | Hard |
|---|---|---|---|
| Structure Verification | 86% | 79% | 68% |
| Solution Verification | 83% | 85% | 73% |

**Experiment design** We utilized the IndustryOR dataset for evaluation, which consists of three difficulty levels (easy, medium, and hard) that allow us to test the generalization capabilities of OptiVer across varying problem complexities. *The hard problems can be general problems with complex structures that fall out of the conventional problem classifications.* However, this analysis was labor-intensive, as the IndustryOR dataset does not provide detailed, step-by-step ground-truth labels necessary for our analysis. To ensure the correctness of this evaluation, we resorted to a manual checking process, which is time-consuming. We first manually annotated the optimization models for the sampled problems to establish a ground truth. To facilitate our evaluation, we randomly selected ten problems from each difficulty level. For generating incorrect models, we initially labeled the models and extracted the structures. We then created nine negative samples for each labeled model by randomly deleting or rewriting some of the variables and constraints. This resulted in 30 positive and 270 negative modeling samples. For **structure evaluation**, we collected the interpreted structures from both the positive and negative samples. The evaluator compared these interpreted structures with the ground-truth structures and generated a binary score. For **solution evaluation**, we used the positive and negative samples to generate solutions, which we then interpreted and assessed for reliability.

Table 6: Inference efficiency comparison of different methods.

| Metric | Method | NL4Opt | MAMO ComplexLP | ComplexOR | IndustryOR |
|---|---|---|---|---|---|
| Inference Time | CoE | 58.2 | 72.5 | 98.6 | 79.9 |
| | OptiMUS | 64.2 | 80.3 | 101.3 | 88.2 |
| | Ours | 52.8 | 67.6 | 68.2 | 59.4 |
| Agent Calls | CoE | 13.6 | 15.7 | 16.6 | 14.1 |
| | OptiMUS | 10.4 | 13.8 | 15.3 | 13.9 |
| | Ours | 9.0 | 9.1 | 9.4 | 9.2 |
| Token Usage | CoE | 6745.8 | 7705.0 | 9469.8 | 8751.5 |
| | OptiMUS | 7039.4 | 8248.4 | 10796.1 | 9062.7 |
| | Ours | 5320.7 | 7385.2 | 8457.4 | 6819.8 |

Table 7: The token usage of each step.

| Step | NL4Opt | MAMO ComplexLP | ComplexOR | IndustryOR |
|---|---|---|---|---|
| Structure Distillation | 492.1 | 441.1 | 665.4 | 501.1 |
| Modeling | 1175.5 | 1944.8 | 2147.1 | 1564.7 |
| Structure-side Verification | 474.6 | 597.4 | 594.2 | 662.7 |
| Solution-side Verification | 302.4 | 422.1 | 521.3 | 347.4 |
| Refinement | 728.7 | 915.5 | 1013.5 | 759.0 |
| Coding and Debugging | 2147.4 | 3064.4 | 3515.9 | 2984.9 |

**Results** The evaluation accuracy and recall rates are presented in Tables 4 and 5. For each difficulty level, we evaluated 10 positive samples and 90 negative samples. Both precision and recall rates for the negative samples are high across the difficulty levels, demonstrating the reliability of the scores. We find that the verification process still performs well for hard problems that cannot be classified into a specific problem type, indicating the strong generalization to general problems.

## 5.5 COMPARISON OF SOLVING EFFICIENCY

**Efficiency Definition** We examine the solving efficiency of OptiVer in comparison to the prompt-based baselines CoE and OptiMUS by analyzing the average time, token usage and agent calls to solve a problem. We use the **same solver (Gurobi)** for all methods. This ensures fairness in efficiency comparisons. The solving time in Table 6 contains the modeling time using LLMs and the execution time of the solver. The solver execution time is short (under 0.01 seconds) and can be neglected during this process. Thus, the solving time in Table 6 reflects the modeling time by LLMs. The results presented in Table 6 indicate that OptiVer achieves significantly shorter solving times, showcasing its high efficiency. Furthermore, we analyzed the internal cost of our verification steps. The key finding in Table 7 is that the extra verification steps are computationally friendly. The modeling and coding part takes over 60% of the tokens.

**The Reason why OptiVer is efficient** Compared to other prompt-based baselines, **OptiVer has a simpler workflow**. CoE and OptiMUS are based on the multi-agent cooperation framework. The workflow of these methods is automatically controlled by a management agent. The insufficient decision-making ability of the management agent may lead to suboptimal decision chains. In the experiments, we find that this method may repeatedly call the same agent. For example, for certain complex problems, CoE may call the terminology interpreter again and again. In contrast, OptiVer does not include such a management agent, leading to a simpler workflow.

## 6 CONCLUSION

In this paper, we propose an LLM-based verification framework designed to enhance the accuracy of automated mathematical modeling tasks. In the structure-side verification, we assess the modeling structures of the current model to ensure structural consistency. Meanwhile, in the solution-side verification, we interpret the solution within the context of the problem descriptions, aiming to identify any logical or mathematical errors in the models. Extensive experiments demonstrate the effectiveness of our method across a wide range of benchmarks.

ETHICS STATEMENT.

This work is designed to explore the significance of the verification process in LLM optimization modeling. We do not foresee any direct, immediate, or negative societal impacts of our research.

REPRODUCIBILITY STATEMENT.

All the results in this work are reproducible. We have discussed the implementation details in Section 5. We also present our prompts for each agent in Appendix F.

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

## USE OF LLMs

Large language models (LLMs) were used to aid writing polish, including refining sentence phrasing, logical flow, and prose clarity, without altering original meanings or technical details. We use LLM to generate the robot logos in Figure 2, 3, 4, and 5. LLMs did not participate in core research tasks (e.g., experiment design, data processing, model training, result analysis, or drafting key technical content).

## A    PROOF OF PROPOSITIONS

### A.1    PROOF OF PROPOSITION 4.1

*Proof.* We first prove the inequality $I(D, M) \geq I(S, M)$. Using the chain rule, we obtain $I(D, M) = I(S, M) + I(D, M \mid S) - I(S, M \mid D)$. We analyze each term in this decomposition within the context of our framework.

**We show** $I(S, M \mid D) = 0$. In our framework, the structure $S$ is derived deterministically (or via a high-fidelity extraction) from the description $D$ by the Distillation Agent. Therefore, given $D$, $S$ is fixed and provides no additional uncertainty reduction for $M$. Thus, the conditional mutual information is zero. We show $I(D, M \mid S) \geq 0$. By the non-negativity property of mutual information, this term is non-negative. Substituting these values back into the decomposition yields the fundamental inequality: $I(D, M) \geq I(S, M)$.

**Then, we prove the inequality** $I(S, M) \geq I(S, \tilde{S})$. Using the chain rule, we obtain $I(S, M) = I(S, \tilde{S}) + I(S, M \mid \tilde{S}) - I(S, \tilde{S} \mid M)$. We analyze each term in this decomposition within the context of our framework. We show $I(S, \tilde{S} \mid M) = 0$. In our framework, the structure $\tilde{S}$ is derived deterministically from the description $M$. Thus, the conditional mutual information is zero. We show $I(S, M \mid \tilde{S}) \geq 0$. By the non-negativity property of mutual information, this term is non-negative. Substituting these values back into the decomposition yields the fundamental inequality: $I(S, M) \geq I(S, \tilde{S})$.

$\square$

### A.2    PROOF OF PROPOSITION 4.2

*Proof.* We prove that the mutual information between the problem description and the model is an upper bound for the mutual information between the description and the interpreted solution. Using the chain rule for mutual information, we obtain $I(D, M) = I(D, \tilde{D}) + I(D, M \mid \tilde{D}) - I(D, \tilde{D} \mid M)$. We analyze each term in this decomposition within the context of our framework:

$I(D, \tilde{D} \mid M) \geq 0$ **(Information Leakage)**. In our implementation, the Solution Interpretation Agent uses the original description $D$ to ensure the interpreted solution $\tilde{D}$ uses the correct terminology (e.g., mapping variable $x_{ij}$ back to "Reservoir $i$ to $j$"). Thus, there is some information flow from $D$ to $\tilde{D}$ that is not mediated by the symbolic model $M$. By the non-negativity property, this term is non-negative.

$I(D, M \mid \tilde{D}) \gg 0$ **(Information Loss)**. This term represents the information $M$ contains about $D$ that is lost when summarizing the solution into $\tilde{D}$. The model $M$ represents the entire set of logical constraints and relationships defined in $D$. The interpreted solution $\tilde{D}$ describes only one specific solution derived from $M$. Since the general model logic $M$ contains vastly more information about the problem definition $D$ than a single solution instance $\tilde{D}$, this Information Loss term is strictly positive and significantly dominates the information leakage term ($I(D, M \mid \tilde{D}) \gg I(D, \tilde{D} \mid M)$).

Substituting these values back into the decomposition yields the fundamental inequality $I(D, M) \geq I(D, \tilde{D})$.

$\square$

## A.3 PROOF OF PROPOSITION 4.3

*Proof.* For any non-negative random variable $Z \triangleq I(S, \tilde{S})$ and any threshold $\varepsilon > 0$, the pointwise inequality $Z \geq \varepsilon \mathbf{1}_{\{Z \geq \varepsilon\}}$ holds. Substituting our terms yields:

$$I(S, \tilde{S}) \geq \varepsilon \mathbf{1}_{\{I(S, \tilde{S}) \geq \varepsilon\}} = \varepsilon c_c \quad \text{a.s.}$$

Taking expectations on both sides gives $\mathbb{E}[I(S, \tilde{S})] \geq \varepsilon \mathbb{E}[c_c]$. □

## A.4 PROOF OF PROPOSITION 4.4

*Proof.* Let $Z' \triangleq I(D, \tilde{D})$. Applying the inequality $Z' \geq \varepsilon' \mathbf{1}_{\{Z' \geq \varepsilon'\}}$ and taking expectations yields the result. □

# B  MORE EXPERIMENT RESULTS

## B.1 THE PROBLEMS WE TRY TO ADDRESS IS CRITICAL IN OPTIMIZATION MODELING

The OptiVer framework is designed and validated for broad applicability across a wide range of problem domains and model types. Our experimental results provide compelling evidence for this generalizability.

Table 8: The proportion of errors.

|  | NL4Opt | Mamo ComplexLP |
|---|---|---|
| Missing Constraints | 37.3 | 20.0 |
| Failure Model Debugging | 51.8 | 40.0 |

We demonstrate that the motivations and challenges we address are common and critical in the optimization modeling field.

**The missing constraints**  Section 4.5 of the OptiMUS paper (Ahmaditeshnizi et al., 2024) has summarized and classified common errors, including missing or wrong constraints, incorrect model, and coding errors. Missing or wrong constraints mean the model fails to extract all the constraints from the model or generates wrong constraints. An incorrect model means errors, such as defining binary variables for visiting cities instead of links in TSP. The prevalence of such mistakes is notable in benchmarks, with 36.0% in NL4Opt and 12.6% in ComplexOR.

**Incorrect model debugging**  This is also a common challenge. The OptiMUS paper (Ahmaditeshnizi et al., 2024) points out that "Coding errors are easier to identify and fix. In contrast, identifying bugs in the formulation requires deeper reasoning and is harder." In existing methods, the debugging module is called only when the solver codes raise execution errors.

# C  EXPERIMENT SETTING IN SECTION 5.3

The plug-and-play capability of OptiVer is supported by both its architectural design and empirical results across diverse setups. We have built OptiVer based on two modeling baselines (OptiMUS and ORLM) and two pretrained backbone LLMs (GPT-4o and Qwen2.5-14B). OptiMUS are prompt-based methods with general LLMs as backbones (we use GPT4o-mini here), which can process any text inputs. We first extract multi-level structures for the problems using a structure distillation agent. These extracted structures are then appended to the problem descriptions and sent as input to the OptiMUS. Once the baselines generate an initial formulation, we proceed with OptiVer's verification step. However, ORLM is a fine-tuned model designed to handle only specific input formats. Therefore, the ORLM model is used solely to provide an initial optimization model, while we perform the verification and refinement processes using the GPT4o-mini model.

# D CASE STUDY

For the example to explain why OptiVer can mitigate the errors, we provide the following optimization problem with output of CoT and OptiVer.

**Problem** (simplified version) In a vast network of canals interconnecting 9 strategic reservoirs, engineers have meticulously mapped out the maximum capacities for water transfer from reservoirs $i$ to $j$, i.e., $c_{ij}$. The challenge is to optimize this water distribution system to ascertain the maximum possible water transfer from Reservoir 0 to Reservoir 8 per day.

---

**Optimization Formulation**

**Optimization Model Given by CoT:**

- **Decision Variables:** $x_{ij}$: The amount of water transferred from Reservoir $i$ to Reservoir $j$, where $i, j = 0, 1, 2, \ldots, 8$.

- **Objective Function:** Maximize the water transfer from Reservoir 0 to 8:

$$Z = \sum_{j=1}^{8} x_{0j}$$

- **Constraints:**
  1. Water transfer capacities: $x_{ij} \leq c_{ij}$, for all $i, j$.
  2. Non-negativity: $x_{ij} \geq 0$, for all $i, j$.

---

This model is incorrect due to missing flow balance constraints. The verification process is outlined as follows:

- **Structure-Augmented Modeling:** The model references a maximum flow problem. It correctly formulates the flow balance constraint when recalling the standard model.

- **Structure-Side Verification:** The model interprets the current optimization model and compares it with the structure of the original problems.

- **Solution-Side Verification:** If the model lacks flow balance constraints, the obtained solution is represented as $x_{ij} = c_{ij}$. The evaluation agent in OptiVer analyzes the solutions and determines that the inflow does not equal the outflow within the system. Consequently, the evaluation agent identifies this discrepancy as an error.

---

**Optimization Formulation**

**Optimization Model Given by OptiVer:**

- **Modeling structures:**
  - **High Level:** Maximum flow problem
  - **Medium Level:** Single commodity maximum flow
  - **Low Level:**
    1. Directed Network: The flow is directed from one reservoir to another.
    2. Capacity Constraints: Each edge has a maximum capacity.
    3. Flow Conservation: The amount of water entering any intermediate reservoir must equal the amount leaving.
- **Decision Variables:** $x_{ij}$: The amount of water transferred from Reservoir $i$ to Reservoir $j$.
- **Objective Function:** Maximize the water transfer:

$$Z = \sum_{j=1}^{8} x_{0j}$$

- **Constraints:**
  1. $x_{ij} \leq c_{ij}$, for all $i, j$.
  2. $x_{ij} \geq 0$, for all $i, j$.
  3. Flow Conservation: $\sum_{\substack{j=0 \\ j \neq k}}^{8} x_{kj} = \sum_{\substack{i=0 \\ i \neq k}}^{8} x_{ik}$ for $k$ in the reservoirs

---

**Analysis**  The modeling structures are proposed to address the challenges of missing constraints. The core of structure-augmented modeling is to identify a similar standard optimization model, and identify the implicit constraints using the standard optimization model as a reference.

# E   ERROR ANALYSIS ON DIFFERENT PROBLEM TYPES

The results presented in Table 9 clearly demonstrate OptiVer's strong generalization capabilities, as it consistently and significantly outperforms the CoT baseline across five distinct and challenging problem categories. This robust performance is particularly evident in problem types where standard prompting methods struggle. For instance, on the Capacitated TSP, where CoT achieves a mere 5.13% accuracy, OptiVer boosts performance to 48.72%. Similarly, for Diet, Transportation, and Maximum Flow problems, OptiVer elevates accuracy from the 16-27% range to a much more effective 55-82% range. This shows that OptiVer's verification process can successfully navigate complex problem structures that are difficult for LLMs to model correctly. Furthermore, even in cases where the CoT baseline is already strong in some problems, such as the Facility Location-Allocation Problem (80.65%), OptiVer still provides a significant improvement, pushing the accuracy to 93.55%. The consistent and substantial performance lift across this diverse set of problems underscores that OptiVer's adaptive verification framework is a broadly applicable and effective strategy, rather than a technique tailored to a specific problem type.

Table 9: The performance on each problem category on the MAMO ComplexLP dataset

| Problem Category | CoT | OptiVer |
|---|---|---|
| Diet Problem | 27.27% | 81.82% |
| Transportation Problem | 23.53% | 70.59% |
| Capacitated TSP | 5.13% | 48.72% |
| Maximum Flow Problem | 16.28% | 55.81% |
| Facility Location-Allocation Problem | 80.65% | 93.55% |

# F THE PROMPT DESIGN

## F.1 STRUCTURE DISTILLATION

```
    interpretation_prompt=[
"""
You are a mathematical formulator working with a team of optimization
    ↪ experts. The objective is to tackle a complex optimization problem.
""",
"""
Please interpret and explain the following problem description.

{problem}

- What is the specific problem type of this OR and CO problem? What
    ↪ specific kind of OR problem?
""",
"""
This is the base formulation of the problem

{base_formulation}

- What is the subdivision of different kinds of this problem?
- Is this base formulation correct?
""",
"""
- Is there any implicit constraints in the problem, including but not
    ↪ limited to the logical selection relation, if/else and if/then
    ↪ relation?
""",
"""
Please summarize and write in JSON Format. For 'subdivision', please find
    ↪  the ones matching this problem description

```json
{{
  "problem_type": ..,
  "specific_type": ...,
  "subdivisions": {{
    subdivision 1: description,
    subdivision 2: description,
    ...
  }},
  "implicit_constraints": {{
    implicit constraint 1: description,
    implicit constraint 2: description,
    ...
  }},
}}
```

- Note that I'm going to use python json.loads() function to parse the
    ↪ json file, so please make sure the format is correct (don't add ','
    ↪  before enclosing '}}' or ']' characters.
- Generate the complete json file and don't omit anything.
- Use '```json' and '```' to enclose the json file.
"""
]
```

## F.2 STRUCTURE-AUGMENTED MODELING

```
formulation_prompt = [
"""
You are an expert mathematical formulator and an optimization professor
    ↪ at a top university. Your task is to model the problem in the
    ↪ standard LP or MILP form.
""",
"""
Here is the description of the problem to be formulated.

{problem}

- Please summarize the parameters and their tensor sizes.
- Please explain the definition of the parameters.
- Please keep the answer brief and concise.
""",
"""
please write in JSON Format. Make sure the bracket is closed, especially
    ↪ when processing the matrices. Do not transpose the matrices and
    ↪ keep the shape of the matrices.

{{
    "parameters": [
        {
            "symbol": "mathematical symbol of the parameters",
            "definition": "definition of the parameters","
            "value": the value of the parameters,
            "shape": [],
        },
        {
            "symbol": "mathematical symbol of the parameters",
            "definition": "definition of the parameters",
            "value": the value of the parameters,
            "shape": [],
        },
        ...
    ],
}}

- Use CamelCase and full words for new variable symbols, and do not
    ↪ include indices in the symbol (e.g. ItemsSold instead of itemsSold
    ↪ or items_sold or ItemsSold_i)
- Note that I'm going to use python json.loads() function to parse the
    ↪ json file, so please make sure the format is correct (don't add ','
    ↪  before enclosing '}}' or ']' characters.
- Use '```json' and '```' to enclose the json file.
""",
"""
Here are some of the cases when we need auxiliary variables. Do we need
    ↪ to include auxiliary binary variables in the formulation?

- Logical Conditions: When a decision depends on a binary condition (e.g.,
    ↪  whether to open a facility or not, use a kind of transportation or
    ↪  not ,and so on), auxiliary binary variables can represent these
    ↪ conditions.
- Modeling step costs: Using binary variables involves creating a
    ↪ mathematical formulation where costs change based on specific
    ↪ thresholds or levels of activity.
- Disjunctive Constraints: When a problem involves "either-or" situations,
    ↪  binary variables can be used to model these disjunctions
    ↪ effectively (Combined with the big M method).
- Capacity Constraints: In problems involving limited resources, binary
    ↪ variables can indicate whether a resource is being utilized or not,
    ↪  allowing for better modeling of capacity.
```

```
- Selection Problems: In scenarios where a fixed set of items or
    ↪ variables can be selected (e.g., choosing a subset of projects to
    ↪ fund), binary variables indicate the selection status.
- Scheduling Order: When determining the sequence in which tasks are
    ↪ performed, binary variables can indicate the order of tasks (e.g.,
    ↪ Task A before Task B). This is often used in job-shop scheduling or
    ↪  project scheduling.
- Penalty Costs: In scheduling with penalties for delays (like tardiness
    ↪ or unmet deadlines), binary variables can help track whether a task
    ↪  incurs a penalty, allowing for cost minimization.
- Job Switching: In scenarios where workers or machines can switch
    ↪ between tasks, binary variables can indicate if a switch occurs,
    ↪ helping to manage transition times and costs.
""",
"""
This problem is a {problem_type} problem with structures

{structure}

To analyze the description carefully, here is the base formulation of
    ↪ this problem (which can be correct or needs to be modified)

{base_formulation}

Now take a deep breath and formulate this problem according to the
    ↪ description and base formulation.

- Consider whether we need to introduce auxiliary binary variables, note
    ↪ that do not include redundant variables.
- For variables, use integer type for discrete items (such as production,
    ↪  unit, people) and continuous ones for continuous items (water,
    ↪ land, time, grams, and so on).
- Your formulation should be in LaTeX mathematical format (do not include
    ↪  the $ symbols).
- Important: You can not define new parameters. You can only define new
    ↪ variables. Use CamelCase and full words for new variable symbols,
    ↪ and do not indices in the symbol (e.g. ItemsSold instead of
    ↪ itemsSold or items_sold or ItemsSold_i). You can include indices in
    ↪  the constraint and objective formulations.
- Make sure that you do not use the numeric number in the formulation
    ↪ except when necessary, instead, you use the parameter name (you can
    ↪  include indices in the constraint and objective formulations).
- Always use non-strict inequalities (e.g. \\leq instead of <), even if
    ↪ the constraint is strict.

Take a deep breath and solve the problem step by step.
"""
]
```

## F.3 STRUCTURE INTERPRETATION AND STRUCTURE CONSISTENCY VERIFICATION

```
modification_prompt = [
"""
You are an expert mathematical formulator and an optimization professor
    ↪ at a top university. Your task is to model and fix the problem in
    ↪ the standard LP or MILP form.
""",
"""
This is a {problem_type} problem with parameters

{parameters}

The formulation is as follows

{formulation_interpretation}

Does this problem consistent with the characteristics of the following
    ↪ structure description? If yes, please say "Yes" directly.
If not, please give your comments to modify the formulation.

{original_problem_interpretation}
""",
"""
Please reformulate the problem to make the formulation consistent with
    ↪ the structure description.

- Consider whether we need to introduce extra binary variables or
    ↪ linearization for a piece-wise linear function.
- Your formulation should be in LaTeX mathematical format (do not include
    ↪  the $ symbols).
- Important: You can not define new parameters. You can only define new
    ↪ variables. Use CamelCase and full words for new variable symbols,
    ↪ and do not include indices in the symbol (e.g. ItemsSold instead of
    ↪  itemsSold or items_sold or ItemsSold_i). You can include indices
    ↪ in the constraint and objective formulations.
- Make sure that you do not use a numeric number in the formulation
    ↪ except where necessary; instead, you use the parameter name (you
    ↪ can include indices in the constraint and objective formulations).
- Always use non-strict inequalities (e.g. \\leq instead of <), even if
    ↪ the constraint is strict.

Take a deep breath and solve the problem step by step.
"""
]
```

### F.4 SOLUTION INTERPRETATION AND SOLUTION VALIDITY VERIFICATION

```
solution_prompt = [
"""
You are an expert mathematical formulator and an optimization professor
    ↪ at a top university. Your task is to model and fix the problem
    ↪ using the solution information in the standard LP or MILP form.
""",
"""
This is a {problem_type} problem with solutions

{solutions}

The formulation is as follows

{formulation_interpretation}

Please interpret the meaning of the solution.
""",
"""
Here is the problem description.

{original_problem_interpretation}

Is this solution the optimal solution? The optimal solution should be
    ↪ mathematical sound and logical coherence:
- We cannot find a better solution.
- The solution should meet the constraints of the problem description.

If yes, please say "Yes" directly.
If not, please give your comments to modify the formulation.
""",
"""
Please reformulate the problem to make the formulation consistent with
    ↪ the structure description.

- Consider whether we need to introduce extra binary variables or
    ↪ linearization for a piece-wise linear function.
- Your formulation should be in LaTeX mathematical format (do not include
    ↪  the $ symbols).
- Important: You can not define new parameters. You can only define new
    ↪ variables. Use CamelCase and full words for new variable symbols,
    ↪ and do not include indices in the symbol (e.g. ItemsSold instead of
    ↪  itemsSold or items_sold or ItemsSold_i). You can include indices
    ↪ in the constraint and objective formulations.
- Make sure that you do not use a numeric number in the formulation
    ↪ except where necessary; instead, you use the parameter name (you
    ↪ can include indices in the constraint and objective formulations).
- Always use non-strict inequalities (e.g. \\leq instead of <), even if
    ↪ the constraint is strict.

Take a deep breath and solve the problem step by step.
"""
]
```

# G   MORE EXAMPLES

## G.1   NL4OPT

---

**Example in NL4Opt**

**Natural Language Description:**
A fishery wants to transport their catch. They can either use local sled dogs or trucks. Local sled dogs can take 100 fish per trip while trucks can take 300 fish per trip. The cost per trip for sled dogs is $50 while the cost per trip for a truck is $100. The budget is at most $1000 and the number of sled dog trips must be less than the number of truck trips. Formulate an LP to maximize the number of fish that can be transported.

- - - - - - - - - - - - - - - - - - - - - - - - - - - - - - - - - - - - - - - - - - - - - -

**Modeling Structure Given by OptiVer:**

```
1  {
2    "problem_type": "Linear Programming",
3    "specific_type": "Resource Allocation Problem",
4    "subdivisions": {
5      "transportation optimization": "Optimizing the transport
         ↪ of goods or resources using various modes of
         ↪ transport.",
6      "resource allocation": "Distributing limited resources to
         ↪ achieve the best possible outcome.",
7      "cost minimization and profit maximization": "Balancing
         ↪ costs against outputs to maximize profit or benefits
         ↪ ."
8    },
9    "implicit_constraints": {
10     "non-negative trips": "The number of trips (x and y)
         ↪ cannot be negative.",
11     "integer trips": "The number of trips should generally be
         ↪ an integer value."
12   }
13 }
```

- - - - - - - - - - - - - - - - - - - - - - - - - - - - - - - - - - - - - - - - - - - - - -

**Optimization Model Given by OptiVer:**

```
1  {
2      "constraints": [
3          {
4              "name": "Budget Constraint",
5              "description": "The total cost of trips must not
                   ↪ exceed the budget.",
6              "formulation": "CostPerSledDogTrip \\cdot
                   ↪ NumberOfSledDogTrips + CostPerTruckTrip \\cdot
                   ↪  NumberOfTruckTrips \\leq TotalBudget"
7          },
8          {
9              "name": "Trip Comparison Constraint",
10             "description": "The number of sled dog trips must
                   ↪ not exceed the number of truck trips.",
11             "formulation": "NumberOfSledDogTrips \\leq
                   ↪ NumberOfTruckTrips"
12         },
```

```
13        {
14            "name": "Non-negativity Constraints",
15            "description": "The number of trips must be non-
                ↪ negative.",
16            "formulation": "NumberOfSledDogTrips \\geq 0"
17        },
18        {
19            "name": "Non-negativity Constraints",
20            "description": "The number of trips must be non-
                ↪ negative.",
21            "formulation": "NumberOfTruckTrips \\geq 0"
22        },
23        {
24            "name": "Integer Constraints for Sled Dog Trips",
25            "description": "The number of sled dog trips must be
                ↪  an integer.",
26            "formulation": "NumberOfSledDogTrips \\in \\mathbb{Z
                ↪ }"
27        },
28        {
29            "name": "Integer Constraints for Truck Trips",
30            "description": "The number of truck trips must be an
                ↪  integer.",
31            "formulation": "NumberOfTruckTrips \\in \\mathbb{Z}"
32        }
33    ],
34    "variables": [
35        {
36            "definition": "Integer variable representing the
                ↪ number of sled dog trips.",
37            "symbol": "NumberOfSledDogTrips",
38            "type": "integer",
39            "shape": []
40        },
41        {
42            "definition": "Integer variable representing the
                ↪ number of truck trips.",
43            "symbol": "NumberOfTruckTrips",
44            "type": "integer",
45            "shape": []
46        }
47    ],
48    "objective": [
49        {
50            "description": "Maximize the total number of fish
                ↪ transported.",
51            "formulation": "Z = FishPerSledDogTrip \\cdot
                ↪ NumberOfSledDogTrips + FishPerTruckTrip \\cdot
                ↪  NumberOfTruckTrips",
52            "objective_sense": "Maximize"
53        }
54    ]
55 }
```

**Solver Code Given by OptiVer:**

```python
import json
import numpy as np
import math
import gurobipy as gp
from gurobipy import GRB

# Create a new model
model = gp.Model('model')

# Define parameters
parameters = {
    "CostPerSledDogTrip": 50,
    "CostPerTruckTrip": 100,
    "FishPerSledDogTrip": 100,
    "FishPerTruckTrip": 300,
    "TotalBudget": 1000
}

# Define variables
NumberOfSledDogTrips = model.addVar(vtype=GRB.INTEGER, name="
    ↪ NumberOfSledDogTrips")
NumberOfTruckTrips = model.addVar(vtype=GRB.INTEGER, name="
    ↪ NumberOfTruckTrips")

# Add variables to the model
model.update()

# Constraints
# Budget Constraint
model.addConstr(
    parameters["CostPerSledDogTrip"] * NumberOfSledDogTrips +
    parameters["CostPerTruckTrip"] * NumberOfTruckTrips <=
    parameters["TotalBudget"],
    name="BudgetConstraint"
)

# Trip Comparison Constraint
model.addConstr(
    NumberOfSledDogTrips <= NumberOfTruckTrips,
    name="TripComparisonConstraint"
)

# Non-negativity Constraints
model.addConstr(NumberOfSledDogTrips >= 0, name="
    ↪ NonNegativitySledDogTrips")
model.addConstr(NumberOfTruckTrips >= 0, name="
    ↪ NonNegativityTruckTrips")

# Set the objective
model.setObjective(
    parameters["FishPerSledDogTrip"] * NumberOfSledDogTrips +
    parameters["FishPerTruckTrip"] * NumberOfTruckTrips,
    GRB.MAXIMIZE
)

# Optimize the model
model.optimize()
status = model.status

obj_val = None
# Check whether the model is infeasible, has infinite solutions, or
    ↪  has an optimal solution
```

```python
if status == gp.GRB.INFEASIBLE:
    obj_val = "infeasible"
elif status == gp.GRB.UNBOUNDED:
    obj_val = "unbounded"
elif status == gp.GRB.OPTIMAL:
    obj_val = model.objVal

print("Objective Value:", obj_val)
```

## G.2 MAMO COMPLEXLP

---

**Example in Mamo ComplexLP**

**Natural Language Description:**
In a scenario involving a salesperson who needs to visit five different cities to conduct business, each city is uniquely numbered from 1 to 5. The salesperson's objective is to minimize the total travel expenses, which could be influenced by factors such as distance, fuel costs, or transportation fees. The salesperson can start their journey from any of these cities but must ensure they visit each city exactly once before returning to their starting point.

The travel costs between the cities are as follows:
- From City 1, the travel costs are 58 units to City 2, 15 units to City 3, 75 units to City 4, and 91 units to City 5.
- From City 2, it costs 58 units to City 1, 54 units to City 3, 85 units to City 4, and 11 units to City 5.
- Traveling from City 3, the expenses are 15 units to City 1, 54 units to City 2, 28 units to City 4, and 61 units to City 5.
- From City 4, the costs are 75 units to City 1, 85 units to City 2, 28 units to City 3, and 47 units to City 5.
- Lastly, from City 5, it costs 91 units to City 1, 11 units to City 2, 61 units to City 3, and 47 units to City 4.

Given this setup, what is the minimum total travel cost for the salesperson to visit each city exactly once and then return to the starting city?

- - - - - - - - - - - - - - - - - - - - - - - - - - - - - - - - - - - - - - - - - - - - - - - - - - - - - - - - -

**Modeling Structure Given by OptiVer:**

```
1  {
2    "problem_type": "Combinatorial Optimization Problem",
3    "specific_type": "Traveling Salesman Problem (TSP)",
4    "subdivisions": {
5      "Hamiltonian cycle": "Path that visits each vertex exactly
          ↪  once and returns to the starting vertex",
6      "Weighted graph": "Graph with weights on edges
          ↪ representing travel costs between cities",
7      "Directed graph": "Graph where edges have a direction,
          ↪ indicating the cost of travel from one city to
          ↪ another"
8    },
9    "implicit_constraints": {
10     "subtour elimination": "Explicit constraints to prevent
          ↪ subtours in the solution",
11     "start_end city constraint": "Salesperson must start and
          ↪ end at the same city"
12   }
13 }
```

- - - - - - - - - - - - - - - - - - - - - - - - - - - - - - - - - - - - - - - - - - - - - - - - - - - - - - - - -

**Optimization Model Given by OptiVer:**

```
1  {
2      "constraints": [
3          {
4              "name": "Each City Visited Once",
```

```json
 5              "description": "Each city must be visited exactly
                   ↪ once by the salesperson.",
 6              "formulation": "\\sum_{j \\in Cities} x_{ij} = 1\\
                   ↪ quad \\forall i \\in Cities"
 7          },
 8          {
 9              "name": "Return to Start City",
10              "description": "The salesperson must return to the
                   ↪ starting city after visiting all cities.",
11              "formulation": "\\sum_{i \\in Cities} x_{ji} = 1\\
                   ↪ quad \\forall j \\in Cities"
12          },
13          {
14              "name": "Subtour Elimination",
15              "description": "Constraints to prevent subtours in
                   ↪ the solution.",
16              "formulation": "u_i - u_j + (|Cities|) \\cdot x_{ij}
                   ↪  \\leq |Cities| - 1\\quad \\forall i, j \\in
                   ↪ Cities, \\; i \\neq j"
17          },
18          {
19              "name": "Position Constraints",
20              "description": "Position variables must be within
                   ↪ valid range.",
21              "formulation": "2 \\leq u_i \\leq |Cities| \\quad \\
                   ↪ forall i \\in Cities"
22          }
23      ],
24      "variables": [
25          {
26              "definition": "Binary variable indicating whether
                   ↪ the salesperson travels from city i to city j
                   ↪ .",
27              "symbol": "x_ij",
28              "type": "binary",
29              "shape": [
30                  5,
31                  5
32              ]
33          },
34          {
35              "definition": "Auxiliary continuous variable
                   ↪ representing the position of city i in the
                   ↪ tour.",
36              "symbol": "u_i",
37              "type": "continuous",
38              "shape": [
39                  5
40              ]
41          }
42      ],
43      "objective": [
44          {
45              "description": "Minimize the total travel cost for
                   ↪ the salesperson.",
```

```
46              "formulation": "Z = \\sum_{i \\in Cities} \\sum_{j
                    ↪ \\in Cities} d_{ij} \\cdot x_{ij}",
47              "objective_sense": "Minimize"
48          }
49      ]
50  }
```

---

**Solver Code Given by OptiVer:**

```python
import json
import numpy as np
import math
import gurobipy as gp
from gurobipy import GRB

# Create a new model
model = gp.Model('model')

# Define parameters
parameters = {
    "Cities": [1, 2, 3, 4, 5],
    "TravelCosts": [
        [0, 58, 15, 75, 91],
        [58, 0, 54, 85, 11],
        [15, 54, 0, 28, 61],
        [75, 85, 28, 0, 47],
        [91, 11, 61, 47, 0]
    ]
}

Cities = parameters["Cities"]
TravelCosts = parameters["TravelCosts"]
num_cities = len(Cities) # Number of cities

# Define variables
x = model.addVars(num_cities, num_cities, vtype=GRB.BINARY, name="x
    ↪ ") # Binary variables for routes
u = model.addVars(num_cities, vtype=GRB.CONTINUOUS, name="u") #
    ↪ Continuous variables for subtour elimination

# Constraints: Each City Visited Once
for i in range(num_cities):
    model.addConstr(gp.quicksum(x[i, j] for j in range(num_cities)
        ↪ if i != j) == 1, f"VisitOnce_Cities[{i}]")

# Constraints: Return to Start City
for j in range(num_cities):
    model.addConstr(gp.quicksum(x[i, j] for i in range(num_cities)
        ↪ if i != j) == 1, f"ReturnToStart_Cities[{j}]")

# Subtour Elimination Constraints
for i in range(1, num_cities):
    for j in range(1, num_cities):
        if i != j:
            model.addConstr(u[i] - u[j] + num_cities * x[i, j] <=
                ↪ num_cities - 1, f"SubtourElimination_Cities[{i}][{j
                ↪ }]")

# Position Constraints
```

```python
for i in range(num_cities):
    model.addConstr(u[i] >= 2, f"LowerBound_u[{i}]")
    model.addConstr(u[i] <= num_cities, f"UpperBound_u[{i}]")

# Objective: Minimize total travel cost
model.setObjective(gp.quicksum(TravelCosts[i][j] * x[i, j] for i in
    ↪  range(num_cities) for j in range(num_cities)), GRB.MINIMIZE)

# Optimize the model
model.optimize()
status = model.status

obj_val = None
# Check whether the model is infeasible, has infinite solutions, or
    ↪  has an optimal solution
if status == gp.GRB.INFEASIBLE:
    obj_val = "infeasible"
elif status == gp.GRB.UNBOUNDED:
    obj_val = "unbounded"
elif status == gp.GRB.OPTIMAL:
    obj_val = model.objVal

print("Objective Value:", obj_val)
```

## G.3 COMPLEXOR

> **Example in ComplexOR**
>
> **Natural Language Description:**
> The capacitated warehouse location problem involves determining the optimal locations for a set number of warehouses to service customers at minimum cost, taking into account warehouse capacities, operating costs, and customer demand.
> The capacitated warehouse location problem is the problem of locating NumberOfLocations warehouses which have to service NumberOfCustomers customers, at minimum cost. Each customer has an associated demand CustomerDemand. There are constraints on the total demand that can be met from a warehouse, as specified by WarehouseCapacity. Costs are incurred when allocating service to customers from warehouses ServiceAllocationCost, and warehouses have a fixed operating cost WarehouseFixedCost. Additionally, there is a lower limit MinimumDemandFromWarehouse on the amount of demand that a warehouse must meet if it is opened, as well as constraints on the minimum MinimumOpenWarehouses and maximum MaximumOpenWarehouses number of warehouses that can be operational.
> The total number of potential warehouse locations is 10. The total number of customers to be serviced is 20. The demand of each customer is [117, 86, 69, 53, 110, 74, 136, 140, 126, 79, 54, 86, 114, 76, 136, 73, 144, 51, 53, 120]. The cost of allocating service from each warehouse to each customer is [[80, 94, 44, 51, 190, 44, 129, 178, 129, 91, 172, 119, 177, 150, 90, 51, 53, 97, 184, 87], [139, 33, 104, 135, 50, 176, 97, 121, 47, 29, 186, 163, 149, 108, 156, 169, 100, 160, 153, 85], [153, 36, 18, 170, 18, 181, 178, 68, 171, 106, 159, 110, 21, 106, 91, 29, 144, 140, 155, 116], [103, 59, 78, 125, 14, 11, 152, 95, 76, 173, 36, 148, 75, 132, 59, 153, 113, 74, 185, 71], [193, 186, 130, 145, 114, 150, 33, 154, 20, 75, 103, 30, 137, 131, 167, 32, 53, 150, 176, 166], [159, 130, 156, 65, 36, 59, 199, 124, 104, 72, 180, 73, 43, 152, 143, 90, 161, 65, 172, 141], [173, 121, 110, 127, 22, 159, 195, 137, 47, 10, 87, 11, 154, 66, 126, 60, 152, 54, 20, 25], [181, 34, 186, 152, 109, 195, 133, 198, 30, 65, 69, 19, 109, 143, 108, 196, 59, 133, 10, 123], [82, 113, 147, 21, 88, 24, 38, 16, 70, 122, 148, 192, 116, 108, 18, 20, 143, 18, 116, 142], [176, 170, 87, 91, 195, 183, 124, 89, 72, 97, 89, 23, 45, 196, 97, 27, 83, 81, 171, 148]]. The total capacity for each warehouse is [3010, 2910, 4530, 4720, 4920, 3750, 4930, 2970, 3310, 2460]. The lower limit on the demand that must be met from a warehouse if it is to be operational is [64, 55, 27, 71, 93, 90, 89, 87, 43, 50]. The minimum number of warehouses that need to be operational is 3. The maximum number of warehouses that can be operational is 8. The fixed operating cost of each warehouse is [8517, 5068, 9433, 6127, 6033, 5966, 7762, 9406, 6602, 7040].
>
> ------------------------------------------------------------
>
> **Modeling Structure Given by OptiVer:**
>
> ```
> 1  {
> 2    "problem_type": "Mixed Integer Linear Programming",
> 3    "specific_type": "Capacitated Warehouse Location Problem",
> 4    "subdivisions": {
> 5      "1": "Location Optimization",
> 6      "2": "Capacity Planning",
> 7      "3": "Cost Minimization"
> 8    },
> 9    "implicit_constraints": {
> 10     "1": "Each customer's demand must be fully met.",
> 11     "2": "A warehouse that is opened must meet a specified
>            ↪ minimum demand."
> 12   }
> 13 }
> ```
>
> ------------------------------------------------------------
>
> **Optimization Model Given by OptiVer:**

```json
{
    "constraints": [
        {
            "name": "Demand Meeting",
            "description": "Each customer's demand must be fully
                ↪  met.",
            "formulation": "\\sum_{j=1}^{NumberOfLocations} y_{
                ↪ ij} = CustomerDemand[i] \\quad \\forall i"
        },
        {
            "name": "Capacity Limit",
            "description": "The total demand served from each
                ↪ warehouse cannot exceed its capacity.",
            "formulation": "\\sum_{i=1}^{NumberOfCustomers} y_{
                ↪ ij} \\leq WarehouseCapacity[j] \\cdot x_j \\
                ↪ quad \\forall j"
        },
        {
            "name": "Minimum Demand",
            "description": "A warehouse that is opened must meet
                ↪  a specified minimum demand.",
            "formulation": "y_{ij} \\geq
                ↪ MinimumDemandFromWarehouse[j] \\cdot x_j \\
                ↪ quad \\forall j, \\forall i"
        },
        {
            "name": "Minimum Open Warehouses",
            "description": "At least a minimum number of
                ↪ warehouses must be operational.",
            "formulation": "\\sum_{j=1}^{NumberOfLocations} x_j
                ↪ \\geq MinimumOpenWarehouses"
        },
        {
            "name": "Maximum Open Warehouses",
            "description": "At most a maximum number of
                ↪ warehouses can be operational.",
            "formulation": "\\sum_{j=1}^{NumberOfLocations} x_j
                ↪ \\leq MaximumOpenWarehouses"
        }
    ],
    "variables": [
        {
            "definition": "Binary variable indicating whether
                ↪ warehouse j is opened (1) or not (0)",
            "symbol": "x_j",
            "type": "binary",
            "shape": [
                10
            ]
        },
        {
            "definition": "Continuous variable representing the
                ↪ amount of demand served from warehouse j to
                ↪ customer i",
            "symbol": "y_{ij}",
            "type": "continuous",
```

```
42            "shape": [
43                20,
44                10
45            ]
46        }
47    ],
48    "objective": [
49        {
50            "description": "Minimize the total cost of servicing
                ↪  customers, including service allocation and
                ↪ operating costs of warehouses.",
51            "formulation": "Z = \\sum_{j=1}^{NumberOfLocations}
                ↪ WarehouseFixedCost[j] \\cdot x_j + \\sum_{i=1}
                ↪ ^{NumberOfCustomers} \\sum_{j=1}^{
                ↪ NumberOfLocations} ServiceAllocationCost[j][i]
                ↪  \\cdot y_{ij}",
52            "objective_sense": "Minimize"
53        }
54    ]
55 }
```

**Solver Code Given by OptiVer:**

```python
import json
import numpy as np
import math
import gurobipy as gp
from gurobipy import quicksum # Import quicksum

# Create a Gurobi model
model = gp.Model('model')

# Parameters
NumberOfLocations = 10
NumberOfCustomers = 20

CustomerDemand = [117, 86, 69, 53, 110, 74, 136, 140, 126, 79,
              54, 86, 114, 76, 136, 73, 144, 51, 53, 120]

ServiceAllocationCost = [
   [80, 94, 44, 51, 190, 44, 129, 178, 129, 91, 172, 119, 177, 150,
       ↪  90, 51, 53, 97, 184, 87],
   [139, 33, 104, 135, 50, 176, 97, 121, 47, 29, 186, 163, 149, 108,
       ↪  156, 169, 100, 160, 153, 85],
   [153, 36, 18, 170, 18, 181, 178, 68, 171, 106, 159, 110, 21, 106,
       ↪  91, 29, 144, 140, 155, 116],
   [103, 59, 78, 125, 14, 11, 152, 95, 76, 173, 36, 148, 75, 132,
       ↪ 59, 153, 113, 74, 185, 71],
   [193, 186, 130, 145, 114, 150, 33, 154, 20, 75, 103, 30, 137,
       ↪ 131, 167, 32, 53, 150, 176, 166],
   [159, 130, 156, 65, 36, 59, 199, 124, 104, 72, 180, 73, 43, 152,
       ↪  143, 90, 161, 65, 172, 141],
   [173, 121, 110, 127, 22, 159, 195, 137, 47, 10, 87, 11, 154, 66,
       ↪  126, 60, 152, 54, 20, 25],
   [181, 34, 186, 152, 109, 195, 133, 198, 30, 65, 69, 19, 109, 143,
       ↪  108, 196, 59, 133, 10, 123],
```

```
    [82, 113, 147, 21, 88, 24, 38, 16, 70, 122, 148, 192, 116, 108,
        ↪ 18, 20, 143, 18, 116, 142],
    [176, 170, 87, 91, 195, 183, 124, 89, 72, 97, 89, 23, 45, 196,
        ↪ 97, 27, 83, 81, 171, 148]
]

WarehouseCapacity = [3010, 2910, 4530, 4720, 4920, 3750, 4930, 2970,
    ↪ 3310, 2460]

WarehouseFixedCost = [8517, 5068, 9433, 6127, 6033, 5966, 7762,
    ↪ 9406, 6602, 7040]

MinimumDemandFromWarehouse = [64, 55, 27, 71, 93, 90, 89, 87, 43,
    ↪ 50]

MinimumOpenWarehouses = 3
MaximumOpenWarehouses = 8

# Variables
x = model.addVars(NumberOfLocations, vtype=gp.GRB.BINARY, name="x")
y = model.addVars(NumberOfCustomers, NumberOfLocations, vtype=gp.
    ↪ GRB.CONTINUOUS, name="y")

# Objective function
model.setObjective(
    quicksum(WarehouseFixedCost[j] * x[j] for j in range(
        ↪ NumberOfLocations)) +
    quicksum(quicksum(ServiceAllocationCost[j][i] * y[i, j] for j in
        ↪  range(NumberOfLocations)) for i in range(
        ↪ NumberOfCustomers)),
    gp.GRB.MINIMIZE
)

# Constraints
for i in range(NumberOfCustomers):
    model.addConstr(
        quicksum(y[i, j] for j in range(NumberOfLocations)) ==
            ↪ CustomerDemand[i],
        name=f"demand_meeting_{i}"
    )

for j in range(NumberOfLocations):
    model.addConstr(
        quicksum(y[i, j] for i in range(NumberOfCustomers)) <=
            ↪ WarehouseCapacity[j] * x[j],
        name=f"capacity_limit_{j}"
    )

for j in range(NumberOfLocations):
    model.addConstr(
        quicksum(y[i, j] for i in range(NumberOfCustomers)) >=
            ↪ MinimumDemandFromWarehouse[j] * x[j],
        name=f"minimum_demand_{j}"
    )

model.addConstr(
    quicksum(x[j] for j in range(NumberOfLocations)) >=
        ↪ MinimumOpenWarehouses,
    name="minimum_open_warehouses"
)

model.addConstr(
```

```
    quicksum(x[j] for j in range(NumberOfLocations)) <=
        ↪ MaximumOpenWarehouses,
    name="maximum_open_warehouses"
)

# Optimize the model
model.optimize()

# Check the optimization status
status = model.status

obj_val = None
if status == gp.GRB.INFEASIBLE:
    obj_val = "infeasible"
elif status == gp.GRB.UNBOUNDED:
    obj_val = "unbounded"
elif status == gp.GRB.OPTIMAL:
    obj_val = model.objVal

print("Objective Value:", obj_val)
```

## G.4 COMPLEXOR

---

**Example in ComplexOR**

**Natural Language Description:**
The Knapsack Problem involves selecting the most valuable combination of items to fit in a knapsack without exceeding its weight limit. The Knapsack Problem is a classic optimization problem in operations research and computer science. The problem is to determine the most valuable combination of items to include in a knapsack, given a set of TotalItems with different values and weights represented by ItemValues and ItemWeights respectively, and a maximum weight capacity of the knapsack MaxKnapsackWeight. The goal is to maximize the total value of the items in the knapsack, represented by ItemValues, without exceeding its weight capacity MaxKnapsackWeight. The available kinds of items is 6. The value of each kind of item is [17, 4, 10, 21, 12, 18]. The weight of each item is [23, 6, 14, 30, 15, 25]. The maximum weight capacity of the knapsack is 60. The total weight of the selected items must not exceed MaxKnapsackWeight. Multiple items in one kind can be selected. No more than TotalItems kinds of items can be considered for selection. Maximize the total value of the items in the knapsack.

- - - - - - - - - - - - - - - - - - - - - - - - - - - - - - - - - - - - - - - - - - - - -

**Modeling Structure Given by OptiVer:**

```
1  {
2    "problem_type": "Combinatorial Optimization",
3    "specific_type": "Bounded Knapsack Problem",
4    "subdivisions": {
5      "Subdivision 1": "0/1 Knapsack Problem",
6      "Subdivision 2": "Bounded Knapsack Problem",
7      "Subdivision 3": "Multi-dimensional Knapsack Problem"
8    },
9    "implicit_constraints": {
10     "implicit constraint 1": "x_i must be an integer",
11     "implicit constraint 2": "MaxKnapsackWeight must be non-
          ↪ negative"
12   }
13 }
```

- - - - - - - - - - - - - - - - - - - - - - - - - - - - - - - - - - - - - - - - - - - - -

**Optimization Model Given by OptiVer:**

```
1  {
2      "constraints": [
3          {
4              "name": "Weight Limit Constraint",
5              "description": "The total weight of selected items
                  ↪ must not exceed the maximum weight capacity of
                  ↪  the knapsack.",
6              "formulation": "\\sum_{i=1}^{TotalItems}
                  ↪ ItemWeights_i \\cdot ItemQuantities_i \\leq
                  ↪ MaxKnapsackWeight"
7          },
8          {
9              "name": "Non-negativity and Integer Constraints",
10             "description": "The quantity of each item selected
                  ↪ must be non-negative and integer.",
```

```
11              "formulation": "ItemQuantities_i \\geq 0\\quad \\
                    ↪ text{and integer for }i = 1,2,\\ldots,
                    ↪ TotalItems"
12          }
13      ],
14      "variables": [
15          {
16              "definition": "Number of items of type i selected (i
                    ↪  = 1to TotalItems)",
17              "symbol": "ItemQuantities_i",
18              "type": "integer",
19              "shape": []
20          }
21      ],
22      "objective": [
23          {
24              "description": "Maximize the total value of the
                    ↪ selected items in the knapsack.",
25              "formulation": "Z = \\sum_{i=1}^{TotalItems}
                    ↪ ItemValues_i \\cdot ItemQuantities_i",
26              "objective_sense": "Maximize"
27          }
28      ]
29  }
```

---

**Solver Code Given by OptiVer:**

```python
import json
import numpy as np
import math
import gurobipy as gp

# Create a new model
model = gp.Model('model')

# Parameters
TotalItems = 6
ItemValues = [17, 4, 10, 21, 12, 18]
ItemWeights = [23, 6, 14, 30, 15, 25]
MaxKnapsackWeight = 60

# Variables: ItemQuantities_i (integer variables)
ItemQuantities = model.addVars(TotalItems, vtype=gp.GRB.INTEGER,
    ↪ name="ItemQuantities")

# Objective: Maximize Z = sum(ItemValues_i * ItemQuantities_i)
model.setObjective(gp.quicksum(ItemValues[i] * ItemQuantities[i]
    ↪ for i in range(TotalItems)), gp.GRB.MAXIMIZE)

# Constraints
# Weight Limit Constraint: sum(ItemWeights_i * ItemQuantities_i) <=
    ↪  MaxKnapsackWeight
model.addConstr(gp.quicksum(ItemWeights[i] * ItemQuantities[i] for
    ↪ i in range(TotalItems)) <= MaxKnapsackWeight, "WeightLimit")

# Non-negativity and Integer Constraints are inherently defined by
    ↪ the variable type
```

```python
# ItemQuantities_i >= 0 and ItemQuantities_i in Z
# Gurobi automatically treats integer variables as non-negative, so
    ↪  no additional constraint is needed for non-negativity.

# Optimize the model
model.optimize()
status = model.status

obj_val = None
# Check whether the model is infeasible, has infinite solutions, or
    ↪  has an optimal solution
if status == gp.GRB.INFEASIBLE:
    obj_val = "infeasible"
elif status == gp.GRB.UNBOUNDED:
    obj_val = "unbounded"
elif status == gp.GRB.OPTIMAL:
    obj_val = model.objVal

print("Objective Value:", obj_val)
```

