# OpenReview forum: "OptiVer: Unleashing the Power of LLMs for Optimization Modeling via Dual-Side Verification"
_ICLR.cc/2026/Conference — Submitted to ICLR 2026_

### Official Review · Reviewer_SBAA · 2025-10-28

**Soundness:** 2
**Presentation:** 2
**Contribution:** 2
**Rating:** 4
**Confidence:** 4

**Summary:**

- The paper proposes Optiver, an LLM-based framework to improve the formulation of optimization models through dual-side verification - checking both the model structure and the solution
- It uses a multi-agent approach: one set of agents performs structure-based consistency verification (via back-translation-type approaches) and another handles solution-side validity verification (by interpreting the actual solution itself for logical consistency)
- Results on benchmarks show notable improvements compared to prior methods

**Strengths:**

- The paper addresses a timely problem, namely the verification issue of LLM-based optimization modeling
- The empirical results are comprehensive, considering multiple benchmarks and ablation studies to indicate the importance of each algorithmic component
- The approach of dual-side verification, and its inspiration in mutual information, is well-explained

**Weaknesses:**

- While the dual-side verification framing is novel, much of the structure (multi-agent framework, back-translation, verification prompts) resembles existing self-correction or multi-agent LLM paradigms. The paper may be more of an incremental refinement than a conceptual breakthrough.
- The mutual information arguments are superficial—they’re not formally derived or validated empirically.
- The performance analysis focuses on solving accuracy (SA), while the main focus of the paper is on verification. It would benefit from a more rigorous demonstration of verification performance, failure modes, and types of errors that are detected
- Also, it is not clear what proportion of the reported gains comes from structure vs solution-side verification

**Questions:**

- Since GPT-4o-mini is used for both generation and evaluation, how do you avoid circular bias (the model verifying its own output)?
- How would this generalize beyond simple OR tasks (e.g., LP/MIP) on stochastic/combinatorial problems
- The paper feels more engineering, although this is not a major weakness IMO, it would be worth highlighting more clearly the methodological contributions.
- I might have missed this, but are all baselines evaluated using the same underlying LLM?
- Can the authors also report the token consumption to get a sense for the normalized performance improvements?

---

> ### Author Response · Authors · 2025-11-21
> **Response to Reviewer SBAA---Part 1/6**
>
> We would like to express our sincere gratitude once again for Reviewer SBAA's valuable feedback and constructive suggestions. We have made detailed clarifications. **We sincerely hope that our additional response has adequately addressed your concerns.**. If so, we would greatly appreciate your consideration in **raising the score**. If there are any remaining concerns, **please let us know**, and we will **continue to actively address your comments and work on improving our submission**.
>
> ### Weakness 1&Question3
>
> > W1&Q3:   The paper may be more of an incremental refinement than a conceptual breakthrough.
>
> Thank you for this comment. We'd like to clarify our core methodological contribution, which we believe is a conceptual advance, not just an incremental one. You are correct that components like "multi-agent frameworks" and "back-translation" are familiar paradigms. Our novelty, however, lies not in inventing these general-purpose tools, but in **designing a new, domain-specific verification paradigm** grounded in the unique challenges of Operations Research.
>
> - **Existing Self-Correction vs. OptiVer:** Existing self-correction methods for optimization modeling overwhelmingly rely on **code-level debugging**. They check if the code produces a syntax error or a runtime error. This is fundamentally insufficient for OR, as a model can be **syntactically perfect (code runs) but logically flawed**, producing a wrong, infeasible, or unbounded solution.
> - **Our Conceptual Shift:** Our paper's contribution is to shift the focus from **code-level verification** to **model-level verification**. OptiVer is the first framework, to our knowledge, to systematically check the logical and mathematical soundness of the optimization model itself.
>
> Our two verification sides are novel, domain-specific contributions that do not exist in general self-correction:
>
> - **Structure-Side Verification:** This is not generic "back-translation." It is a specific OR-grounded process that compares the abstracted structure of the generated model against the distilled requirements of the problem. This is explicitly designed to catch common, hard-to-find OR modeling errors like missing implicit constraints.
> - **Solution-Side Verification:** This is a conceptually new idea. We are the first to propose using the semantic meaning of the solution as a verification signal. By interpreting the solution (e.g., "we are shipping a negative amount of goods") and checking it for real-world absurdity, we create a powerful new debugging tool that code-level checks cannot provide.
>
> In summary, the dual-side verification is a new paradigm tailored for optimization modeling.

---

> ### Author Response · Authors · 2025-11-21
> **Response to Reviewer SBAA---Part 2/6**
>
> ### Weakness 2
>
> > W2: The mutual information arguments are superficial—they're not formally derived or validated empirically.
>
> - Thank you for your insightful comments regarding the theoretical framing of mutual information (MI). We clarify that we employ MI not as a calculated loss function, but as the **theoretical foundation** to characterize the alignment between the distributions of two distinct semantic spaces.
>
>   - We conceptualize the modeling process as a mapping between probability distributions in two spaces:
>
>     - **Language Space $\mathcal{D}$.** The space of natural language descriptions. We conceptually decompose the distribution $P_D$ into three distinct components:
>       - **Background-related information $P_{D_{\text{bg}}}$.** Represents the narrative scenarios and real-world contexts that are irrelevant to the mathematical formulation.
>       - **Modeling-related information $P_{D_{\text{model}}}$.** Represents the semantic intent, logical requirements, constraint types, and objectives essential for the optimization problem.
>       - **Data Value information $P_{D_{\text{data}}}$.** Represents the parameter data values in the problems.
>
>     - **Modeling Space $\mathcal{M}$.** We explicitly define the modeling space as the Cartesian product of the variable space, constraint space, and objective function space. Let $P_M$ be the distribution representing the symbolic structures generated by the model.
>
>     **MI is a widely used theoretical metric for the similarity of the distributions**. The mutual information $I(D, M)\approx I(\{D_{\text{model}}, D_{\text{data}}\}, M)$ is a canonical metric used to quantify the dependence (or similarity) between such distributions. It measures the reduction in uncertainty about the distribution in $\mathcal{M}$ given the observation.
>
> - We improve the representation of our theoretical framework by detailing the binary judgment as an MI proxy. We explain that the consistency binary scores $c$ forms the inequalities $\mathbb{E}[I(M, D)] \geq\mathbb{E}[I(S, \tilde{S})] \geq \varepsilon \mathbb{E}[c_c]$ and $\mathbb{E}[I(M, D)] \geq\mathbb{E}[I(S, \tilde{S})] \geq \varepsilon \mathbb{E}[c_c]$. Thus, **optimizing the consistency scores is in fact optimizing the lower bound**. Calculating the exact MI involves integrating over these high-dimensional, implicit distributions, which is computationally intractable. Our verification agents perform a **binary classification task**, outputting a consistency score $c \in \{0, 1\}$ (corresponding to the scores $c_c$ and $c_v$ in our framework). This binary judgment reflects the magnitude of the mutual information: a consistent judgment ($c=1$) indicates that the alignment between the distributions exceeds a required confidence threshold. To demonstrate this rigor, we present two derived propositions.
>
>   > **Proposition.**
>   >
>   > Let $\varepsilon > 0$ be a fixed threshold representing the minimum information overlap required for a model to be deemed structurally consistent. Define the Bernoulli random variable $c_c$ corresponding to the binary output of our Structure Evaluation Agent:
>   >
>   > $$c_c \triangleq \mathbf{1}_{\{I(S, \tilde{S}) \geq \varepsilon\}},$$
>   >
>   > where $I(S, \tilde{S})$ denotes the mutual information between the distilled structure $S$ and the interpreted structure $\tilde{S}$. Then, the expected mutual information satisfies:
>   >
>   > $$\mathbb{E}[I(S, \tilde{S})] \geq \varepsilon \mathbb{E}[c_c].$$
>   >
>   > **Proof.**
>   >
>   > For any non-negative random variable $Z \triangleq I(S, \tilde{S})$ and any threshold $\varepsilon > 0$, the pointwise inequality $Z \geq \varepsilon \mathbf{1}_{\{Z \geq \varepsilon\}}$ holds. Substituting our terms yields:
>   >
>   > $$I(S, \tilde{S}) \geq \varepsilon \mathbf{1}_{\{I(S, \tilde{S}) \geq \varepsilon\}} = \varepsilon c_c \quad \text{a.s.}$$
>   >
>   > Taking expectations on both sides gives $\mathbb{E}[I(S, \tilde{S})] \geq \varepsilon \mathbb{E}[c_c]$.
>
>   > **Proposition.**
>   >
>   > Similarly, let $\varepsilon' > 0$ be a threshold for solution validity and define $c_v \triangleq \mathbf{1}_{\{I(D, \tilde{D}) \geq \varepsilon'\}}$, representing the binary output of the Solution Evaluation Agent. Then:
>   >
>   > $$\mathbb{E}[I(D, \tilde{D})] \geq \varepsilon' \mathbb{E}[c_v].$$
>   >
>   > **Proof.**
>   >
>   > Following the same argument as above, let $Z' \triangleq I(D, \tilde{D})$. Applying the inequality $Z' \geq \varepsilon' \mathbf{1}_{\{Z' \geq \varepsilon'\}}$ and taking expectations yields the result.
>
>   These propositions provide the rigorous link between our method's mechanics and the theoretical objective. They show that our refinement process—which aims to maximize the probability of passing verification ($\mathbb{E}[c]$)—is mathematically guaranteed to monotonically increase a **linear lower bound** on the mutual information between the language and modeling distributions.

---

> ### Author Response · Authors · 2025-11-21
> **Response to Reviewer SBAA---Part 3/6**
>
> - To address your concerns, we empirically validated with mutual information calculations. The computation of mutual information is intractable. To evaluate the similarity, we compute the cosine similarity between natural language descriptions and modeling structures, as well as the consistency between hierarchical structures and their reconstructed counterparts before and after the verification process. The results presented in **Table 1** demonstrate that our method can improve the mutual information and thus reduce the errors in missing constraints.
>
>   **Table 1**: The mutual information and the error ratios in missing constraints.
>
>   | | NL4Opt | MAMO ComplexLP | IndusstryOR |
>   | -| -- | -| -|
>   | $CosSim(S,\tilde{S})$ Before Verification | 0.995  | 0.994 | 0.973|
>   | $CosSim(S,\tilde{S})$ After Verification | 1.000  | 0.997 | 0.988 |
>   | $CosSim(D,\tilde{D})$ Before Verification | 0.542  | 0.556 | 0.492|
>   | $CosSim(D,\tilde{D})$ After Verification | 0.542  | 0.556  | 0.496 |
>   | Missing Constraint Errors, Before Verification | 10.0%  | 24.2% | 35.0% |
>   | Missing Constraint Errors, After Verification| 1.7% | 16.1% | 21.0%|
>
> - To further address the concern of the reviewer, we empirically show that the verification process can improve the consistency. Specifically, we use a strong LLM (DeepSeek-R1) as a judge to compare the semantic consistency of the model's structure before and after our verification step. The results (which we will add as **Table 2**) show a significant improvement in the judge's consistency rating, confirming our hypothesis.
>
>   **Table 2**: The improvement in consistency before and after verifications.
>
>   | | NL4Opt | MAMO ComplexLP | IndusstryOR |
>   |-|-| -| - |
>   |Before Stru-Ver | 0.82   | 0.43 | 0.31 |
>   |After Stru-Ver | 0.98   | 0.71 | 0.54 |
>   |Before Sol-Ver| 0.95   | 0.32 | 0.35  |
>   | After Sol-Ver | 0.99   | 0.69 | 0.49|

---

> ### Author Response · Authors · 2025-11-21
> **Response to Reviewer SBAA---Part 4/6**
>
> ### Weakness 3&4
>
> > W3&4: It would benefit from a demonstration of verification performance, failure modes, and types of errors that are detected.
>
> Thanks for your comments! We conduct more experiments to demonstrate the benefits of the verification process, failure modes, and errors that are detected.
>
> - To address your concerns, we conduct ablation studies on verification performance.
>
>   - **Ablation studies on the overall performance**. We conduct ablation studies to demonstrate the effectiveness of the verifications. The results in **Table 3** show that without each of the verifications, the performance suffers from a drop.
>
>     **Table 3**: The ablation studies on the verifications.
>
>     | | NL4Opt | MAMO ComplexLP | ComplexOR | IndustryOR |
>     | - | -| -- | - | - |
>     | w/o Structure-side Verification| 91.5| 55.2| 68.4| 29.0|
>     | w/o Solution-side Verification| 91.8| 53.1  | 68.4 | 41.0|
>     | OptiVer | 96.5| 66.7| 78.9| 45.0|
>
>   - **Reliability analysis for the verification**. We report the precision and recall for the evaluation in the verification process. We utilized the IndustryOR dataset for evaluation, which consists of three difficulty levels (easy, medium, and hard) that allow us to test the generalization capabilities across varying problem complexities (details can be found in Section 5.4). The evaluation accuracy and recall rates are presented in **Table 4**. Both precision and recall rates for the negative samples are high, demonstrating the reliability.
>
>     **Table 4**: The precision and recall of the structure-side verification in IndustryOR.
>
>     | | Easy | Medium | Hard |
>     | -| - | - | --|
>     | Precision| 92%  | 89%| 83%  |
>     | Recall| 86%| 79%| 68%  |
>
>     **Table 5**: The precision and recall of the solution-side verification in IndustryOR.
>
>     | | Easy | Medium | Hard |
>     | - | -| -| - |
>     | Precision | 93%| 91%| 86%|
>     | Recall| 83%| 85%| 73%|
>
> - Types of Errors that are detected. To explain why OptiVer can mitigate the errors, we provide the following optimization problem with the output of CoT and OptiVer.
>
>   > - Problem:  (simplified version) In a vast network of canals interconnecting 9 strategic reservoirs, engineers have meticulously mapped out the maximum capacities for water transfer from reservoirs $i$ to $j$, i.e., $c_{ij}$. The challenge is to optimize this water distribution system to ascertain the maximum possible water transfer from Reservoir 0 to Reservoir 8 per day.
>   >
>   > - Optimization Model Given by CoT:
>   >
>   >   - Decision Variables $x_{ij}$: The amount of water transferred from Reservoir $i$ to Reservoir $j$, where $i,j=0,1,2,...,8$.
>   >   - Objective Function: Maximize the water transfer from Reservoir 0 to 8: $Z =\sum^8_{j=1}  x_{0j}$.
>   >   - Constraints:
>   >     1. Water transfer capacities: $x_{ij} \le c_{ij}$, for all $i,j$.
>   >     2. Non-negativity: $x_{ij} \ge0$, for all $i,j$.
>   >
>   > - Optimization Model Given by OptiVer:
>   >
>   >   Modeling structures:
>   >
>   >   > - High Level: Maximum flow problem
>   >   > - Medium Level: Single commodity maximum flow
>   >   > - Low Level:
>   >   >   1. Directed Network: The flow is directed from one reservoir to another.
>   >   >   2. Capacity Constraints: Each edge (connection between reservoirs) has a maximum capacity.
>   >   >   3. Flow Conservation: The amount of water entering any intermediate reservoir must equal the amount leaving, except for the source and sink.
>   >
>   >   - Decision Variables $x_{ij}$: The amount of water transferred from Reservoir $i$ to Reservoir $j$, where $i,j=0,1,2,...,8$.
>   >   - Objective Function: Maximize the water transfer from Reservoir 0 to 8: $Z =\sum^8_{j=1}  x_{0j}$.
>   >   - Constraints:
>   >     1. Water transfer capacities: $x_{ij} \le c_{ij}$, for all $i,j$.
>   >     2. Non-negativity: $x_{ij} \ge0$, for all $i,j$.
>   >     3. **Flow Conservation:  $\sum^8_{j=0,j\not=k} x_{kj}= \sum^8_{i=0,i\not=k} x_{ik}$ for $k$ in the reservoirs**
>   >
>   > - Analysis: **The modeling structures are proposed to address the challenges of missing constraints. The core of structure-augmented modeling is to identify a similar standard optimization model, and identify the implicit constraints using the standard optimization model as a reference**.
>
> - Failure modes. We present the failure mode in **Table 6**. OptiVer can identify many of the missing constraints and variable errors. Most of the unaddressed errors are the problems that require extensive common sense and logical reasoning for problem understanding. We believe that with stronger underlying LLMs, these problems can be effectively addressed.
>
>   **Table 6**: The error modes of OptiVer.
>
>   || MAMO ComplexLP |  | IndustryOR  |  |
>   | - | - | - | -- | - |
>   | | Before Verification | After Verification | Before Verification | After Verification |
>   | Missing Constraints | 24.6%| 15.7% | 33.0% | 22.0% |
>   | Incorrect modeling  | 15.2%  | 7.4% | 13.0% | 8.0% |
>   | Coding errors | 5.2% | 4.1%| 3.0%| 2.0% |

---

> ### Author Response · Authors · 2025-11-21
> **Response to Reviewer SBAA---Part 5/6**
>
> ### Question 1
>
> > Q1: Since GPT-4o-mini is used for both generation and evaluation, how do you avoid circular bias (the model verifying its own output)?
>
> Thank you for this insightful question. Our method is specifically designed to avoid this "circular bias" by ensuring the evaluation agents **do not simply re-evaluate the generator's original output**. Instead, we break the circle by introducing **new, external information and contexts** for verification.
>
> - Existing method LLMOPT leverages a single model for both generation and self-correction, achieving 20% performance gain. This demonstrates that the self-correction method can significantly improve the overall performance.
>
> - **Solution-Side**. This is a strong defense against circular bias. The **solution $x$ is not LLM-generated content**; it is the objective output from an **external tool (the solver)**. When an LLM agent identifies a logical absurdity in the solution (e.g., "inflow does not equal outflow," "a negative number of items are shipped"), it is forced to conclude its original model $M$ was flawed. In this way, the LLM is not judging its own reasoning; it is using its reasoning to critique the results generated by the solver.
>
> - **Structure-Side**. This process also avoids circularity. The `Structure_Evaluation_Agent` is not asked to re-check the model it just wrote. Instead, its task is to compare two separate inputs:
>
>   1. **$S$**: The structure distilled from the original problem $D$.
>   2. **$\tilde{S}$**: The interpreted structure from the generated model $M$.
>
>   The agent's job is to find the mismatch between these two distinct texts. This is a comparative task, not a self-referential one.
>
> ### Question 2
>
> > Q2: How would this generalize beyond simple OR tasks (e.g., LP/MIP) on stochastic/combinatorial problems
>
> Yes, our method can easily generalize to stochastic combinatorial problems. To address your concerns, we collect and construct a small testing bed with 10 harder problems, including stochastic problems and non-linear problems. We test our methods on this testing bed with results in **Table 7**. The results demonstrate that our method can effectively generalize to and improve harder problems. We also provide an example as follows.
>
> > A tech retailer stocks wireless earbuds monthly. The monthly demand for earbuds is a discrete random variable: 50 units with probability 0.15, 60 units with probability 0.35, 70 units with probability 0.3, and 80 units with probability 0.2. Each unit costs the retailer \\$40, and holding unsold units for a month incurs a \\$3 storage cost per unit. If demand exceeds stock, the retailer offers a \\$10 discount on the next purchase to each customer who experiences a shortage. The retailer restocks once a month. Develop a stochastic optimization model to determine the optimal monthly order quantity that minimizes the expected total monthly cost (acquisition cost + storage cost + shortage compensation cost).
>
> > Decision Variables:
> > $$
> > \begin{aligned}
> > &x \quad \text{: Monthly order quantity (non-negative continuous variable)} \\\\
> > &I_i \quad \text{: Inventory level under the } i\text{-th demand scenario (non-negative continuous variable)} \\\\
> > &S_i \quad \text{: Shortage quantity under the } i\text{-th demand scenario (non-negative continuous variable)}
> > \end{aligned}
> > $$
> >
> >
> > Constraints:
> > $$
> > \begin{align*}
> > &I_i \geq x - D_i \quad \forall i = 1,2,3,4 \quad \text{(Relationship between inventory, order quantity and demand)} \\\\
> > &I_i \geq 0 \quad \forall i = 1,2,3,4 \quad \text{(Non-negativity of inventory)} \\\\
> > &S_i \geq D_i - x \quad \forall i = 1,2,3,4 \quad \text{(Relationship between shortage, demand and order quantity)} \\\\
> > &S_i \geq 0 \quad \forall i = 1,2,3,4 \quad \text{(Non-negativity of shortage)} \\\\
> > &x \geq 0 \quad \text{(Non-negativity of order quantity)}
> > \end{align*}
> > $$
> >
> > Objective Function: Minimize Expected Total Monthly Cost
> > $$
> > \min Z = c \cdot x + \sum_{i=1}^{4} p_i \left( h \cdot I_i + s \cdot S_i \right)$$
> >
> > where the total cost consists of three components:
> >
> > - Acquisition cost: $c \cdot x $
> > - Expected holding cost: $ \sum_{i=1}^{4} p_i \cdot h \cdot I_i $
> > - Expected shortage compensation cost: $ \sum_{i=1}^{4} p_i \cdot s \cdot S_i $
>
> **Table 7**: The Performance on the more complex datasets.
>
> | | Accuracy || Error Modes ||
> | - | - | -| -| - |
> ||  | Modeling Definition Errors | Missing Constraints | Coding Errors |
> | Standard | 30%| 28.6| 57.1 | 14.3|
> | CoT | 50% | 60.0| 40.0 | 0.0 |
> | Ours  | 100%| 0.0  | 0.0 | 0.0 |

---

> ### Author Response · Authors · 2025-11-21
> **Response to Reviewer SBAA---Part 6/6**
>
> ### Question 4
>
> > Q4: I might have missed this, but are all baselines evaluated using the same underlying LLM?
>
> Yes, all the prompt-based methods (Standard, CoT, CoE, OptiMUS, and OptiVer) are implemented using the same LLM (GPT4o-mini).
>
> ### Question 5
>
> > Q5: Can the authors also report the token consumption to get a sense for the normalized performance improvements?
>
> - To provide the detailed token analysis you suggested, we conduct additional experiments. We report the inference time, average agent calls, and token counts across the datasets. We will add the following **Table 8** to the Appendix. The results show that our method uses **fewer agent calls and fewer total tokens** on average than the other multi-agent frameworks.
>
>   **Table 8**: The inference cost across the benchmarks.
>
>   |                 |         | NL4Opt | MAMO ComplexLP | ComplexOR | IndusstryOR |
>   | --------------- | ------- | ------ | -------------- | --------- | ----------- |
>   | Inference  Time | CoE     | 58.2   | 72.5           | 98.6      | 79.9        |
>   |                 | OptiMUS | 64.2   | 80.3           | 101.3     | 88.2        |
>   |                 | Ours    | 52.8   | 67.6           | 68.2      | 59.4        |
>   | Agent Calls     | CoE     | 13.6   | 15.7           | 16.6      | 14.1        |
>   |                 | OptiMUS | 10.4   | 13.8           | 15.3      | 13.9        |
>   |                 | Ours    | 9.0    | 9.1            | 9.4       | 9.2         |
>   | Token Usage     | CoE     | 6745.8 | 7705.0         | 9469.8    | 8751.5      |
>   |                 | OptiMUS | 7039.4 | 8248.4         | 10796.1   | 9062.7      |
>   |                 | Ours    | 5320.7 | 7385.2         | 8457.4    | 6819.8      |
>
> - While OptiVer does have a multi-step workflow, we find it is **significantly more efficient in practice** than other multi-agent baselines. The reason is that existing methods like **CoE and OptiMUS use an LLM manager agent** to automate their workflow. We observed that this manager, due to the stochastic nature of LLMs **repeatedly calls the same agent** or falls into sub-optimal decision chains. OptiVer, by contrast, uses a **fixed, directed workflow**. This eliminates the overhead of the manager agent and prevents these costly, ineffective loops.
>
> - Furthermore, we analyzed the internal cost of our verification steps. The key finding is that the extra verification steps are computationally friendly. The modeling and coding part takes over 60% of the tokens. In summary, we believe this small computational investment is the key to our performance, enabling OptiVer to catch errors that other methods miss, as shown in our ablation studies.
>
>   **Table 9**: The token usage of each step.
>
>   |                             | NL4Opt | MAMO ComplexLP | ComplexOR | IndusstryOR |
>   | --------------------------- | ------ | -------------- | --------- | ----------- |
>   | Structure Distillation      | 492.1  | 441.1          | 665.4     | 501.1       |
>   | Modeling                    | 1175.5 | 1944.8         | 2147.1    | 1564.7      |
>   | Structure-side Verification | 474.6  | 597.4          | 594.2     | 662.7       |
>   | Solution-side Verification  | 302.4  | 422.1          | 521.3     | 347.4       |
>   | Refinement                  | 728.7  | 915.5          | 1013.5    | 759.0       |
>   | Coding and Debugging        | 2147.4 | 3064.4         | 3515.9    | 2984.9      |

---

> ### Author Response · Authors · 2025-11-25
>
> Dear Reviewer SBAA,
>
> We deeply appreciate your valuable feedback and the time you've taken to review our work, especially during this busy period.
>
> We are reaching out to kindly inquire about the current status of your review regarding our submission. We sincerely hope that our responses have adequately addressed your concerns. Furthermore, we are eager to address any additional queries you might have, which will enable us to enhance our work further.
>
> Once again, thank you for your guidance and support.
>
> Best, Authors

---

> ### Author Response · Authors · 2025-11-26
> **A More Complex Example of Stochastic Programming Problem**
>
> Thanks for your suggestions on evaluating OptiVer on more general stochastic programming problems. We provide a more complex example to demonstrate the strong generalization ability.
>
> The problem is as follows.
>
> > Consider a grid-connected photovoltaic (PV) system integrated with an energy storage system (ESS) operating over a 24-hour scheduling horizon $T = \{1, 2, ..., 24\}$, where the system aims to maximize the expected daily net profit considering uncertainties in PV generation, time-varying electricity prices, and local load demand. The PV hourly output $P_t^{PV}$ is a continuous random variable uniformly distributed over the interval [80, 550] kW representing the uncertain solar generation at hour $t$, with probability density function $f_{P_t^{PV}}(p) = 1/(550-80) = 1/470$ for $80 \leq p \leq 550$ and 0 otherwise; the electricity price $\lambda_t$ is a mixed random variable, with off-peak hours $t \in T_{off} = \{1\text{-}8, 22\text{-}24\}$ following a truncated normal distribution $\lambda_t \sim N(0.9, 0.1^2)$ yuan/kWh (truncated to [0.7, 1.1] to avoid non-physical values), peak hours $t \in T_{peak} = \{9\text{-}17\}$ being a deterministic high price of 1.8 yuan/kWh, and shoulder hours $t \in T_{shd} = \{18\text{-}21\}$ following a discrete-continuous mixture distribution with 60\% chance to be $U(1.3, 1.5)$ and 40\% chance to be 1.2 yuan/kWh; the local load demand $D_t \sim U(100, 300)$ kW is the uncertain electricity demand of the local load. The ESS is specified with rated capacity $E_{max} = 1000$ kWh, minimum state of charge (SoC) $E_{min} = 100$ kWh (to avoid deep discharge), charging/discharging efficiency $\eta_c = 0.92$ and $\eta_d = 0.90$, maximum charging/discharging power $P_{max}^c = 200$ kW and $P_{max}^d = 200$ kW, and initial SoC $E_0 = 300$ kWh; grid constraints limit the maximum hourly export power to the grid to $P_{max}^{grid} = 400$ kW; penalty costs include a curtailment penalty of $C_{cur} = 0.5$ yuan/kWh for PV power not used for load/ESS/grid and a load shedding penalty of $C_{shed} = 5.0$ yuan/kWh for unmet load demand (strictly minimized).
>
> The generated optimization model is as follows.
>
>
>
> > ##### Decision Variables
> >
> > - $x_t^c$: ESS charging power at hour $t$ (kW)
> > - $x_t^d$: ESS discharging power at hour $t$ (kW)
> > - $x_t^{grid}$: Power exported to grid at hour $t$ (kW)
> > - $x_t^{cur}$: Curtailed PV power at hour $t$ (kW)
> > - $x_t^{shed}$: Shed load at hour $t$ (kW)
> > - $E_t$: ESS state of charge at end of hour $t$ (kWh)
> >
> > ##### Mathematical Formulation
> >
> > $$
> > \begin{aligned}
> > \text{maximize } &\mathbb{E}\left[ \sum_{t=1}^{24} \left( \lambda_t x_t^{grid} - C_{cur} x_t^{cur} - C_{shed} x_t^{shed} \right) \right] &\\\\
> > P_t^{PV} + \eta_d x_t^d &= D_t - x_t^{shed} + \frac{x_t^c}{\eta_c} + x_t^{grid} + x_t^{cur} &\forall t \in T \quad \text{(Power balance)} \\\\
> > E_t &= E_{t-1} + x_t^c \cdot 1 - \frac{x_t^d}{\eta_d \cdot \eta_c} &\forall t \in T \quad \text{(SoC dynamics)} \\\\
> > E_{min} \leq E_t &\leq E_{max} &\forall t \in T \quad \text{(SoC limits)} \\\\
> > E_{0} &= 300 &\text{(Initial SoC)} \\\\
> > x_t^c \cdot x_t^d &= 0 &\forall t \in T \quad \text{(Non-simultaneous charge/discharge)} \\\\
> > 0 \leq x_t^c &\leq P_{max}^c &\forall t \in T \quad \text{(Charging limit)} \\\\
> > 0 \leq x_t^d &\leq P_{max}^d &\forall t \in T \quad \text{(Discharging limit)} \\\\
> > 0 \leq x_t^{grid} &\leq P_{max}^{grid} &\forall t \in T \quad \text{(Grid export limit)} \\\\
> > x_t^{cur} &\geq 0 &\forall t \in T \quad \text{(Curtailment)} \\\\
> > x_t^{shed} &\geq 0 &\forall t \in T \quad \text{(Load shedding)}
> > \end{aligned}
> > $$
> >
> >

---

> ### Author Response · Authors · 2025-11-26
> **A Detailed Example of the Failure Mode**
>
> To further explain the failure mode of OptiVer, we provide an example that OptiVer fails to capture the constraints with sophisticated logical reasoning.
>
> > **Problem**
> >
> > A sudden heavy rainfall has caused flash floods in a region, affecting three towns: Town A, Town B, and Town C. Emergency supplies need to be dispatched from a central warehouse to these towns, with the rescue headquarters operating 5 heavy-duty trucks. Each truck has a maximum load capacity of 8 tons, a maximum daily travel distance of 400 kilometers, and a maximum cumulative operational time including loading and unloading of 12 hours per day. The operation must adhere to a strict priority rule for supply distribution: Lifesaving Supplies > Temporary Shelter Supplies > Quality-of-Life Improvement Supplies. The primary objective is to maximize the total emergency response satisfaction score while respecting constraints on road capacity, vehicle limitations, and delivery timelines. The one-way distances from the warehouse to Town A, Town B, and Town C are 80 km, 120 km, and 150 km respectively. Due to road damage, the maximum number of round trips allowed per day to Town A is 2, while Town B and Town C each permit 3 round trips per day. The round-trip driving time for each truck to Town A is approximately 2.67 hours, to Town B is 4 hours, and to Town C is 5 hours, with a consistent loading and unloading time of 2 hours per trip resulting in total one-way trip times of 4.67 hours for Town A, 6 hours for Town B, and 7 hours for Town C. The supplies are categorized into three types: Lifesaving Supplies such as first-aid kits and life jackets that require specialized insulated containers limiting each truck to a maximum of 2 tons per trip, must be delivered in full to all towns within 12 hours, and have a demand of 5 tons for Town A, 6 tons for Town B, and 4 tons for Town C. Temporary Shelter Supplies including tents and folding beds allow a shortage of up to 30% of the total demand and have a demand of 8 tons for Town A, 10 tons for Town B, and 7 tons for Town C. Quality-of-Life Improvement Supplies like drinking water and ready-to-eat meals have no mandatory delivery requirement with distribution based on remaining capacity and a demand of 10 tons for Town A, 12 tons for Town B, and 9 tons for Town C. The satisfaction score per ton is 10 points for Lifesaving Supplies, 5 points for Temporary Shelter Supplies, and 2 points for Quality-of-Life Improvement Supplies.
> >
> >
> >
> > **Variable Definitions**
> >
> > - $x_{ij}$: Tons of supply type $j$ by truck $i$ ($j=1$: Lifesaving, $j=2$: Temporary Shelter, $j=3$: Quality-of-Life)
> > - $y_{ik}$: Round trips by truck $i$ to town $k$ ($k=A,B,C$)
> > - $z_{jk}$: Tons of supply type $j$ delivered to town $k$
> >
> > **Mathematical Formulation**
> > $$
> > \begin{aligned}
> > \text{maximize } & 10 \sum_{k=A,B,C} z_{1k} + 5 \sum_{k=A,B,C} z_{2k} + 2 \sum_{k=A,B,C} z_{3k} & \\\\
> > &\sum_{i=1}^{5} x_{i1} & = z_{1A} + z_{1B} + z_{1C} = 15 & \text{(Total lifesaving supplies)} \\\\
> > &x_{i1}  \leq 2 & \forall i=1,2,...,5 \quad \text{(Max lifesaving per truck)} \\\\
> > &z_{1A}  = 5, \quad z_{1B} = 6, \quad z_{1C} = 4 & \text{(Lifesaving allocation)} \\\\
> > &\sum_{k=A,B,C} z_{2k}  \geq 17.5 & \text{(Min temporary shelter)} \\\\
> > &\sum_{i=1}^{5} x_{i2}  = z_{2A} + z_{2B} + z_{2C} & \text{(Temporary shelter balance)} \\\\
> > &z_{2A}  \leq 8, \quad z_{2B} \leq 10, \quad z_{2C} \leq 7 & \text{(Temporary shelter limits)} \\\\
> > &\sum_{i=1}^{5} x_{i3}  = z_{3A} + z_{3B} + z_{3C} & \text{(Quality-of-life balance)} \\\\
> > &z_{3A}  \leq 10, \quad z_{3B} \leq 12, \quad z_{3C} \leq 9 & \text{(Quality-of-life limits)} \\\\
> > &4.67y_{iA} + 6y_{iB} + 7y_{iC}  \leq 12 & \forall i=1,2,...,5 \quad \text{(Time constraint)} \\\\
> > &160y_{iA} + 240y_{iB} + 300y_{iC}  \leq 400 & \forall i=1,2,...,5 \quad \text{(Fuel constraint)} \\\\
> > &x_{i1} + x_{i2} + x_{i3}  \leq 8 & \forall i=1,2,...,5 \quad \text{(Capacity constraint)} \\\\
> > &y_{iA}  \in \{0,1,2\} & \forall i=1,2,...,5 \quad \text{(Town A trips)} \\\\
> > &y_{iB}, y_{iC}  \in \{0,1,2,3\} & \forall i=1,2,...,5 \quad \text{(Towns B,C trips)} \\\\
> > &x_{ij}  \geq 0 & \forall i=1,2,...,5; j=1,2,3 \quad \text{(Non-negativity)} \\\\
> > &z_{jk}  \geq 0 & \forall j=1,2,3; k=A,B,C \quad \text{(Non-negativity)}
> > \end{aligned}
> > $$
>
> The current mathematical formulation fails to capture the critical priority relationship among supply types: **Lifesaving Supplies > Temporary Shelter Supplies > Quality-of-Life Improvement Supplies**. This priority structure requires logical reasoning that current LLM-based automated modeling cannot adequately handle.

---

> ### Author Response · Authors · 2025-11-27
>
> Dear Reviewer SBAA,
>
> We would like to extend our sincere gratitude for the time and effort you have devoted to reviewing our submission. Your positive feedback, insightful comments, and constructive suggestions have been invaluable to us, guiding us in improving the quality of our work!
>
> We eagerly await your feedback to understand if our responses have adequately addressed all your concerns. Once again, thank you for your guidance and support.
>
> Best,
>
> Authors

---

> ### Author Response · Authors · 2025-11-27
>
> Dear Reviewer SBAA,
>
> We sincerely thank you for your time and efforts during the rebuttal process. We are looking forward to your feedback to understand if our responses have adequately addressed your concerns. If so, we would deeply appreciate it if you could consider raising your score. If not, please let us know your further concerns, and we will continue actively responding to your comments. We sincerely thank you once more for your insightful comments and kind support.
>
> Best,
>
> Authors

---

### Official Review · Reviewer_FVPi · 2025-10-31

**Soundness:** 3
**Presentation:** 3
**Contribution:** 2
**Rating:** 6
**Confidence:** 4

**Summary:**

This paper proposes the Dual-side Verification (OptiVer) framework to improve the modeling accuracy from both structure and solution perspectives.

**Strengths:**

- Dual-side verification is a good idea to improve the solving accuracy of LLMs.
- This paper proposes the OptiVer framework, which novelly implements bidirectional verification and improves the accuracy of optimization problem modeling.

**Weaknesses:**

- Repeated interactions and calls between multi-agents may result in excessively long call times.
- Similarly, when the problem is complex (long descriptions, many numerical values, etc.), repeated interactions with LLMs may introduce data errors. Have the authors considered this issue? And what is the maximum scale of problems OptiVer can handle?
- For simple problems, such as simple MILP problems with only two variables, overly complex validation is unnecessary and may even introduce more errors.

**Questions:**

As described in weaknesses.

---

> ### Author Response · Authors · 2025-11-21
> **Response to Reviewer FVPi---Part 1/4**
>
> We would like to express our sincere gratitude once again for Reviewer FVPi's valuable feedback and constructive suggestions. We have made detailed clarifications. **We sincerely hope that our additional response has adequately addressed your concerns**. If so, we would greatly appreciate your consideration in **raising the score**. If there are any remaining concerns, **please let us know**, and we will **continue to actively address your comments and work on improving our submission**.
>
> ### Weakness 1
>
> > W1: Repeated interactions and calls between multi-agents may result in excessively long call times.
>
> Thank you for this insightful comment. In our analysis, we found that OptiVer is **significantly more efficient** (in time, tokens, and agent calls) than existing multi-agent baselines like CoE and OptiMUS.
>
> - We report the inference cost for our method and the baselines. We will add **Table 1** to the appendix, comparing OptiVer to the baselines on agent calls, token usage, and inference time. This table demonstrates that OptiVer requires **fewer agent calls, fewer total tokens, and less inference time** on average than both CoE and OptiMUS.
>
>   **Table 1**: The inference cost across the benchmarks.
>
>   |                 |         | NL4Opt | MAMO ComplexLP | ComplexOR | IndusstryOR |
>   | --------------- | ------- | ------ | -------------- | --------- | ----------- |
>   | Inference  Time | CoE     | 58.2   | 72.5           | 98.6      | 79.9        |
>   |                 | OptiMUS | 64.2   | 80.3           | 101.3     | 88.2        |
>   |                 | Ours    | 52.8   | 67.6           | 68.2      | 59.4        |
>   | Agent Calls     | CoE     | 13.6   | 15.7           | 16.6      | 14.1        |
>   |                 | OptiMUS | 10.4   | 13.8           | 15.3      | 13.9        |
>   |                 | Ours    | 9.0    | 9.1            | 9.4       | 9.2         |
>   | Token Usage     | CoE     | 6745.8 | 7705.0         | 9469.8    | 8751.5      |
>   |                 | OptiMUS | 7039.4 | 8248.4         | 10796.1   | 9062.7      |
>   |                 | Ours    | 5320.7 | 7385.2         | 8457.4    | 6819.8      |
>
> - While OptiVer does have a multi-step workflow, we find it is **significantly more efficient in practice** than other multi-agent baselines. The reason is that existing methods like CoE and OptiMUS use an LLM manager agent to automate their workflow. We observed that this manager, due to the stochastic nature of LLMs, often repeatedly calls the same agent and falls into sub-optimal decision chains. OptiVer, by contrast, uses a **fixed, directed workflow**. This eliminates the overhead of the manager agent and prevents these costly, ineffective loops.
>
> - Furthermore, we analyzed the internal cost of our verification steps. The key finding is that the extra verification steps are computationally friendly. The modeling and coding part takes over 60% of the tokens. In summary, we believe this small computational investment is the key to our performance, enabling OptiVer to catch errors that other methods miss, as shown in our ablation studies.
>
>   **Table 2**: The token usage of each step.
>
>   |                             | NL4Opt | MAMO ComplexLP | ComplexOR | IndusstryOR |
>   | --------------------------- | ------ | -------------- | --------- | ----------- |
>   | Structure Distillation      | 492.1  | 441.1          | 665.4     | 501.1       |
>   | Modeling                    | 1175.5 | 1944.8         | 2147.1    | 1564.7      |
>   | Structure-side Verification | 474.6  | 597.4          | 594.2     | 662.7       |
>   | Solution-side Verification  | 302.4  | 422.1          | 521.3     | 347.4       |
>   | Refinement                  | 728.7  | 915.5          | 1013.5    | 759.0       |
>   | Coding and Debugging        | 2147.4 | 3064.4         | 3515.9    | 2984.9      |
>
> [1] Chain-of-Experts: When LLMs Meet Complex Operations Research Problems. ICLR 2024.
>
> [2] OptiMUS: Scalable Optimization Modeling with (MI)LP Solvers and Large Language Models. ICML 2024.

---

> ### Author Response · Authors · 2025-11-21
> **Response to Reviewer FVPi---Part 2/4**
>
> ### Weakness 2
>
> > W2: When the problem is complex (long descriptions, many numerical values, etc.), repeated interactions with LLMs may introduce data errors. Have the authors considered this issue? And what is the maximum scale of problems OptiVer can handle?
>
> This is a very practical and important question about scalability. You've correctly identified two main aspects of a complex problem: (1) long, complex descriptions and (2) a large volume of numerical values. We have designed our framework to handle both, and we believe it is not susceptible to these issues.
>
> - We first specify the **abstract model** and **concrete model**, as referred in Pyomo [3]. A mathematical model can be defined using symbols that represent data values. For example, the following equations represent a linear program to find optimal values for the solutions with parameters $V, S$, and $d_{ij}$.
> $$
> \begin{aligned}
>   \text{min}\_x &Z = \sum\_{i=1}^n \sum_{j=1}^n d_{ij} \cdot x_{ij}\\\\s.t. &\sum_{i=1, i \neq j}^n x_{ij} = 1 \quad \forall j \in V\\\\&\sum_{j=1, j \neq i}^n x_{ij} = 1 \quad \forall i \in V\\\\&\sum_{i \in S} \sum_{j \in S} x_{ij} \leq |S| - 1
> \end{aligned}
> $$
>
>   We call this an abstract model since it relies on unspecified parameter values. The concrete model is created when specific data values are used to instantiate the abstract model.
>
> - **Handling Large-Scale Data Values**. Our method **avoids data errors by not passing large numerical arrays to the LLM** during the verification and refinement loops. Following best practices (also used in baselines like CoE and OptiMUS), numerical data is extracted once and is separated from the problem descriptions. We use the abstract symbols to represent the parameters in the descriptions.
>
>   - For structure-side verification, we do not need the parameter data and can verify the modeling structure using the abstract model.
>
>   - For solution-side verification, the large arrays are loaded by the solver code at runtime, but they are not part of the LLM's prompts. Furthermore, we may encounter situations when the large-scale parameters lead to a long solving time. Our solution-side verification is designed to validate the logic of the abstract model, not necessarily the final solution of a large-scale concrete model. To do this, we simply instantiate the abstract model with a small, toy set of data values. This creates a small, solvable concrete model that is fast to execute (as noted in Appendix B.1, solver time is under 0.01 seconds for the benchmark instances). A logical flaw in the abstract model (e.g., a missing constraint, a flawed objective) will almost always be exposed by this small-scale test.
>
>     > **Example:** In our motivating case (Figure 2), the missing flow balance constraint is a flaw in the abstract model. This logical error is detectable whether the concrete model is instantiated with 9 reservoirs or 9,000 reservoirs.
>
> - **Handling long problem description**. Even in complex problems, the abstract modeling logic is often concise, with at most a few dozen types of constraints. Existing LLMs are able to read a long, complex description once and extract this modeling framework. All subsequent verification steps then operate on this concise, symbolic structure, not the full, error-prone text.
>
> - Here we present a complex problem description (simplified version) as well as the optimization models. We find OptiVer can handle such a complex problem.
>
>
>
> [3] Pyomo: modeling and solving mathematical programs in Python. Mathematical Programming Computation 3(3) (2011): 219-260.

---

> ### Author Response · Authors · 2025-11-21
> **Response to Reviewer FVPi---Part 3/4**
>
> Here we present a complex problem description (simplified version) as well as the optimization models. We find OptiVer can handle such a complex problem.
>
> > A regional power system, comprising $N$ generating units, $L$ transmission lines, and $B$ buses, requires an optimal day-ahead generation schedule spanning $T$ consecutive time periods. The operational objective is to minimize the total system cost, which includes the cumulative fuel costs—modeled as quadratic functions of real power output for each generating unit with coefficients $a_i$, $b_i$, and $c_i$—along with the start-up costs $SUC_i$ and shut-down costs $SDC_i$ incurred by the units throughout the scheduling horizon.
> >
> > The optimization must ensure strict adherence to all physical laws governing power system operation and all security standards. This requires that for each time period, the total real power injected into the system by all online generators must precisely equal the sum of the total real power demand across all loads, $\sum_b D_{b,t}^P$, and the total real power losses dissipated across the entire transmission network. Simultaneously, at every bus and for every time period, the net injection of reactive power—comprising the reactive power output from local generators, minus the local reactive load demand $D_{b,t}^Q$, and adjusted for the net reactive power flow on all connected lines—must be balanced to zero.
> >
> > Each generating unit must operate within its defined technical capabilities. When a unit is committed online, its real power output must be maintained between its minimum $P_i^{\min}$ and maximum $P_i^{\max}$ stable generation limits, and its reactive power output must be kept within its specified minimum $Q_i^{\min}$ and maximum $Q_i^{\max}$ reactive capability bounds. Furthermore, the combined real and reactive power output of the unit must always reside within its physical capability envelope, defined by the apparent power capacity limit $S_i^{\max}$, forming a circular constraint in the P-Q plane. The temporal variation of a unit's real power output is restricted by its dynamic ramp-rate limits; the increase in output from one period to the next cannot exceed the ramp-up rate $RU_i$, and the decrease cannot exceed the ramp-down rate $RD_i$.
> >
> > The commitment schedule of the generating units must respect their inherent mechanical and thermal constraints. Once a unit is started up, it must remain online for a minimum continuous duration of $T_i^{on}$ periods, and once shut down, it must remain offline for a minimum continuous duration of $T_i^{off}$ periods. The logical transitions between operational states are governed by the condition that a change in the unit's on/off status from one period to the next must be consistently represented by a single start-up or shut-down action, which are mutually exclusive events within any given period.
> >
> > The security of the transmission network is paramount. The power flow on any line $l$ must not exceed its thermal rating $F_l^{\max}$ in either direction. The voltage magnitude at each bus $b$ must be maintained within its secure operating bounds, $V_b^{\min}$ and $V_b^{\max}$, at all times. The physical power flows on the lines are governed by the non-linear AC power flow equations, which relate the active and reactive power flow on a line connecting two buses to the voltage magnitudes $V_{b,t}$, $V_{b',t}$ and the phase angle difference $\theta_{b,t} - \theta_{b',t}$ at its terminal buses, using the line's conductance $G_{bb'}$ and susceptance $B_{bb'}$. To ensure a unique solution to the power flow, the voltage phase angle at a pre-selected reference bus $b_{\text{ref}}$ is fixed to zero for all time periods. The total system real power losses, which are a function of all bus voltages and phase angles as per the AC power flow model, must be fully accounted for in the system-wide real power balance equation.

---

> ### Author Response · Authors · 2025-11-21
> **Response to Reviewer FVPi---Part 4/4**
>
> ### Weakness 3
>
> > W3: For simple problems, such as simple MILP problems with only two variables, overly complex validation is unnecessary and may even introduce more errors.
>
> Thanks for your comments! Our empirical results show that this verification is both **necessary** and **reliable**, even on simple problems.
>
> - Existing method also find that self-correction can improve the modeling accuracy, even on the simple datasets NL4Opt. The LLMOPT uses 12 rounds of self-correction and improves over 20% performance on the simple datasets.
>
> - To address your concerns, we conduct ablation studies. We also report the structure and solution evaluation results in the IndustryOR datasets. The dataset are classified into easy, medium and hard problems. In **Tables 3 and 4**, we report the evaluation precision and recall of the verification process (the results and experiments settings can be found in Section 5.4). We find the evaluation step in the verification process has a high precision and recall across the easy and hard problems, which demonstrates the reliable of verification.
>
>   **Table 3**: The precision and recall of the structure-side verification in IndustryOR.
>
>   |           | Easy | Medium | Hard |
>   | --------- | ---- | ------ | ---- |
>   | Precision | 92%  | 89%    | 83%  |
>   | Recall    | 86%  | 79%    | 68%  |
>
>   **Table 4**: The precision and recall of the solution-side verification in IndustryOR.
>
>   |           | Easy | Medium | Hard |
>   | --------- | ---- | ------ | ---- |
>   | Precision | 93%  | 91%    | 86%  |
>   | Recall    | 83%  | 85%    | 73%  |
>
> - To further address your concerns, we conduct additional experiments. We select 50 examples from the existing datasets with less than five variables. We compare the results on the dataset with and without verifications. The results in Table 5 demonstrate that verification can still help the LLMs improve the performance. We provide an example that the verification detects the errors.
>
>   Table 5: Comparion between with and without verifications in easy problems.
>
>   |          | Without Verification | With Verification |
>   | -------- | -------------------- | ----------------- |
>   | Accuracy | 96%                  | 100%              |
>
>   - **Problem**: Suppose a certain animal needs at least 700 g of protein, 30 g of minerals, and 100 mg of vitamins daily. There are 5 types of feed available, and the nutritional content and price per gram of each type of feed are shown in the Table. Try to formulate a linear programming model that meets the animal's growth needs while minimizing the cost of selecting the feed.
>
>     | Feed | Protein (g) | Minerals (g) | Vitamins (mg) | Price (¥/kg) | Feed | Protein (g) | Minerals (g) | Vitamins (mg) | Price (¥/kg) |
>     | ---- | ----------- | ------------ | ------------- | ------------ | ---- | ----------- | ------------ | ------------- | ------------ |
>     | 1    | 3           | 1            | 0.5           | 0.2          | 4    | 6           | 2            | 2             | 0.3          |
>     | 2    | 2           | 0.5          | 1             | 0.7          | 5    | 18          | 0.5          | 0.8           | 0.8          |
>     | 3    | 1           | 0.2          | 0.2           | 0.4          |      |             |              |               |              |
>
>     $$\begin{aligned}
>     \text{minimize } &\sum_{i=1}^5 c_i \cdot x_i  &\\\\
>     \sum_{i=1}^5 p_i x_i &\geq P_{\text{min}} &\text{(Protein)} \\\\
>     \sum_{i=1}^5 m_i x_i &\geq M_{\text{min}} &\text{(Minerals)} \\\\
>     \sum_{i=1}^5 v_i x_i &\geq V_{\text{min}} &\text{(Vitamins)}\\\\
>     x_i \in \mathbb{R}_+ &\quad \forall i \in \{1, 2, 3, 4, 5\}&
>     \end{aligned}$$
>
>     The model has a logical error in the objective function. The cost, $c_i$, was given in yen per **kilogram**, while the variable is measured in **grams**. This unit inconsistency would lead to a wildly incorrect total cost. The **solution-side verification** identified this error.
>
>     $$\begin{aligned}
>     \text{minimize } &\sum_{i=1}^5 \frac{c_i}{1000} \cdot x_i  &\\\\
>     \sum_{i=1}^5 p_i x_i &\geq P_{\text{min}} &\text{(Protein)} \\\\
>     \sum_{i=1}^5 m_i x_i &\geq M_{\text{min}} &\text{(Minerals)} \\\\
>     \sum_{i=1}^5 v_i x_i &\geq V_{\text{min}} &\text{(Vitamins)}\\\\
>     x_i \in \mathbb{R}_+ &\quad \forall i \in \{1, 2, 3, 4, 5\}&
>     \end{aligned}$$

---

> ### Author Response · Authors · 2025-11-25
>
> Dear Reviewer FVPi,
>
> We sincerely thank you for your time and efforts during the rebuttal process. We are looking forward to your feedback to understand if our responses have adequately addressed your concerns. We sincerely thank you once more for your insightful comments and kind support.
>
> Best,
>
> Authors

---

### Official Review · Reviewer_j6Mz · 2025-11-01

**Soundness:** 2
**Presentation:** 2
**Contribution:** 2
**Rating:** 4
**Confidence:** 4

**Summary:**

This paper focuses on the unreliability in the process of modeling optimization problems with large language models and emphasizes the crucial role of process supervision and verification. This paper proposes OptiVer, a method that performs effective verification at both the structural and solution aspects to enhance the reliability of the modeling process. Experimental results demonstrate that OptiVer surpasses all Reasoning Models, Fine-tuning Methods, and Prompt-based Methods in terms of solving accuracy.

**Strengths:**

- Proposes verification of the modeling process from both structural and solution interpretation perspectives, introducing a novel approach for validating modeling.

 - Describes the measurement of reliable modeling from the perspective of mutual information, providing a theoretical metric for the semantic consistency of modeling structures.

 - The experiments in this paper successfully validate the effectiveness of both multi-level modeling and multi-perspective modeling.

**Weaknesses:**

- Conflicting Use of Baseline: The paper says the old NL4OPT and ComplexOR benchmark have some mistakes, but uses it as the main proof that the new one is better. This weakens the argument.

- The proposed Mutual Information metric lacks a specified estimation method and does not play a functional role in the methodology or experiments.

- Although an ingenious framework is designed to verify the reliability of the modeling process, the reliability of the verification process itself may not be guaranteed under the Prompt-based Multi-Agent framework.

**Questions:**

- The Challenges section in Chapter 3 mentions that NL4OPT and ComplexOR contain 36% and 12.6% erroneous cases, respectively. Does this potentially compromise the reliability of the reported 96.5% Solving Accuracy achieved by OptiVer(Full) on NL4OPT and 78.9% on ComplexOR?
- The paper frequently mentions the statistical measure of mutual information. However, the mutual information mentioned herein merely serves a descriptive role in the proposed framework, why is its value not quantitatively estimated to measure the consistency between natural language descriptions and modeling structures, as well as the consistency between hierarchical structures and their reconstructed counterparts?
- During the inference process for a single sample, how many times is the Refinement module executed? Is there a maximum triggering limit?
- In the results shown in Table 2, the ablation results for medium-level and low-level settings on NL4OPT and ComplexOR respectively show no difference. What might be the reason for this? Are these experimental results statistically stable after multiple repetitions?
- Regarding the results in Table 3, the ORLM outcomes are cited from the original or reproduced papers, while the ORLM+OptiVer results are obtained from experiments conducted in this paper. Does this potentially introduce inconsistencies in experimental conditions?

---

> ### Author Response · Authors · 2025-11-21
> **Response to Reviewer j6Mz---Part 1/5**
>
> We would like to express our sincere gratitude once again for Reviewer j6Mz's valuable feedback and constructive suggestions. We have made detailed clarifications. **We sincerely hope that our additional response has adequately addressed your concerns**. If so, we would greatly appreciate your consideration in **raising the score**. If there are any remaining concerns, **please let us know**, and we will **continue to actively address your comments and work on improving our submission**.
>
> ### Weakness&Question 1
>
> >  The paper says the old NL4OPT and ComplexOR benchmarks have some mistakes, but uses them as the main proof that the new one is better.
>
> Thank you for this question—we apologize for the confusing presentation and are grateful for the opportunity to clarify this key point!
>
> - We would like to clarify that the "36.0% and 12.6% errors" do **not** refer to flaws in the benchmarks. Instead, those statistics, which we cited from the OptiMUS paper, refer to the **high rate of modeling errors made by LLMs** when attempting to solve those benchmarks.
>
> - The benchmarks we used are standard datasets, widely adopted by existing works in this field [1,2,3]. This ensures that our comparative analysis is reliable and validates the reported solving accuracy.
>
> [1] Chain-of-Experts: When LLMs Meet Complex Operations Research Problems. ICLR 2024.
>
> [2] OptiMUS: Scalable Optimization Modeling with (MI)LP Solvers and Large Language Models. ICML 2024.
>
> [3] ORLM: A Customizable Framework fin Training Large Models for Automated Optimization Modeling. Operations Research 2025.

---

> ### Author Response · Authors · 2025-11-21
> **Response to Reviewer j6Mz---Part 2/5**
>
> ### Weakness&Question 2
>
> > The proposed mutual information metric lacks a specified estimation method. Measure the consistency quantitatively.
>
> - Thank you for your insightful comments regarding the theoretical framing of mutual information (MI). We clarify that we employ MI not as a calculated loss function, but as the **theoretical foundation** to characterize the alignment between the distributions of two distinct semantic spaces.
>
>   - We conceptualize the modeling process as a mapping between probability distributions in two spaces:
>
>     - **Language Space $\mathcal{D}$.** The space of natural language descriptions. We conceptually decompose the distribution $P_D$ into three distinct components:
>       - **Background-related information $P_{D_{\text{bg}}}$.** Represents the narrative scenarios and real-world contexts that are irrelevant to the mathematical formulation.
>       - **Modeling-related information $P_{D_{\text{model}}}$.** Represents the semantic intent, logical requirements, constraint types, and objectives essential for the optimization problem.
>       - **Data Value information $P_{D_{\text{data}}}$.** Represents the parameter data values in the problems.
>
>     - **Modeling Space $\mathcal{M}$.** We explicitly define the modeling space as the Cartesian product of the variable space, constraint space, and objective function space. Let $P_M$ be the distribution representing the symbolic structures generated by the model.
>
>     **MI is a widely used theoretical metric for the similarity of the distributions**. The mutual information $I(D, M)\approx I(\{D_{\text{model}}, D_{\text{data}}\}, M)$ is a canonical metric used to quantify the dependence (or similarity) between such distributions. It measures the reduction in uncertainty about the distribution in $\mathcal{M}$ given the observation.
>
> - We improve the representation of our theoretical framework by detailing the binary judgment as an MI proxy. We explain that the consistency binary scores $c$ forms the inequalities $\mathbb{E}[I(M, D)] \geq\mathbb{E}[I(S, \tilde{S})] \geq \varepsilon \mathbb{E}[c_c]$ and $\mathbb{E}[I(M, D)] \geq\mathbb{E}[I(S, \tilde{S})] \geq \varepsilon \mathbb{E}[c_c]$. Thus, **optimizing the consistency scores is in fact optimizing the lower bound**. Calculating the exact MI involves integrating over these high-dimensional, implicit distributions, which is computationally intractable. Our verification agents perform a **binary classification task**, outputting a consistency score $c \in \{0, 1\}$ (corresponding to the scores $c_c$ and $c_v$ in our framework). This binary judgment reflects the magnitude of the mutual information: a consistent judgment ($c=1$) indicates that the alignment between the distributions exceeds a required confidence threshold. To demonstrate this rigor, we present two derived propositions.
>
>   > **Proposition.**
>   >
>   > Let $\varepsilon > 0$ be a fixed threshold representing the minimum information overlap required for a model to be deemed structurally consistent. Define the Bernoulli random variable $c_c$ corresponding to the binary output of our Structure Evaluation Agent:
>   >
>   > $$c_c \triangleq \mathbf{1}_{\{I(S, \tilde{S}) \geq \varepsilon\}},$$
>   >
>   > where $I(S, \tilde{S})$ denotes the mutual information between the distilled structure $S$ and the interpreted structure $\tilde{S}$. Then, the expected mutual information satisfies:
>   >
>   > $$\mathbb{E}[I(S, \tilde{S})] \geq \varepsilon \mathbb{E}[c_c].$$
>   >
>   > **Proof.**
>   >
>   > For any non-negative random variable $Z \triangleq I(S, \tilde{S})$ and any threshold $\varepsilon > 0$, the pointwise inequality $Z \geq \varepsilon \mathbf{1}_{\{Z \geq \varepsilon\}}$ holds. Substituting our terms yields:
>   >
>   > $$I(S, \tilde{S}) \geq \varepsilon \mathbf{1}_{\{I(S, \tilde{S}) \geq \varepsilon\}} = \varepsilon c_c \quad \text{a.s.}$$
>   >
>   > Taking expectations on both sides gives $\mathbb{E}[I(S, \tilde{S})] \geq \varepsilon \mathbb{E}[c_c]$.
>
>   > **Proposition.**
>   >
>   > Similarly, let $\varepsilon' > 0$ be a threshold for solution validity and define $c_v \triangleq \mathbf{1}_{\{I(D, \tilde{D}) \geq \varepsilon'\}}$, representing the binary output of the Solution Evaluation Agent. Then:
>   >
>   > $$\mathbb{E}[I(D, \tilde{D})] \geq \varepsilon' \mathbb{E}[c_v].$$
>   >
>   > **Proof.**
>   >
>   > Following the same argument as above, let $Z' \triangleq I(D, \tilde{D})$. Applying the inequality $Z' \geq \varepsilon' \mathbf{1}_{\{Z' \geq \varepsilon'\}}$ and taking expectations yields the result.
>
>   These propositions provide the rigorous link between our method's mechanics and the theoretical objective. They show that our refinement process—which aims to maximize the probability of passing verification ($\mathbb{E}[c]$)—is mathematically guaranteed to monotonically increase a **linear lower bound** on the mutual information between the language and modeling distributions.

---

> ### Author Response · Authors · 2025-11-21
> **Response to Reviewer j6Mz---Part 3/5**
>
> - To address your concerns, we empirically validated with mutual information calculations. The computation of mutual information is intractable. To evaluate the similarity, we compute the cosine similarity between natural language descriptions and modeling structures, as well as the consistency between hierarchical structures and their reconstructed counterparts before and after the verification process. The results presented in **Table 1** demonstrate that our method can improve the mutual information and thus reduce the errors in missing constraints.
>
>   **Table 1**: The mutual information and the error ratios in missing constraints.
>
>   | | NL4Opt | MAMO ComplexLP | IndusstryOR |
>   | -| -- | -| -|
>   | $CosSim(S,\tilde{S})$ Before Verification | 0.995  | 0.994 | 0.973|
>   | $CosSim(S,\tilde{S})$ After Verification | 1.000  | 0.997 | 0.988 |
>   | $CosSim(D,\tilde{D})$ Before Verification | 0.542  | 0.556 | 0.492|
>   | $CosSim(D,\tilde{D})$ After Verification | 0.542  | 0.556  | 0.496 |
>   | Missing Constraint Errors, Before Verification | 10.0%  | 24.2% | 35.0% |
>   | Missing Constraint Errors, After Verification| 1.7% | 16.1% | 21.0%|
>
> - To further address the concern of the reviewer, we empirically show that the verification process can improve the consistency. Specifically, we use a strong LLM (DeepSeek-R1) as a judge to compare the semantic consistency of the model's structure before and after our verification step. The results (which we will add as **Table 2**) show a significant improvement in the judge's consistency rating, confirming our hypothesis.
>
>   **Table 2**: The improvement in consistency before and after verifications.
>
>   | | NL4Opt | MAMO ComplexLP | IndusstryOR |
>   | - | - | -| - |
>   |Before Stru-Ver | 0.82   | 0.43 | 0.31 |
>   |After Stru-Ver | 0.98   | 0.71 | 0.54 |
>   |Before Sol-Ver| 0.95   | 0.32 | 0.35  |
>   | After Sol-Ver | 0.99   | 0.69 | 0.49|

---

> ### Author Response · Authors · 2025-11-21
> **Response to Reviewer j6Mz---Part 4/5**
>
> ### Weakness 3
>
> > Although an ingenious framework is designed to verify the reliability of the modeling process, the reliability of the verification process itself may not be guaranteed under the Prompt-based Multi-Agent framework
>
> Thanks for your comment! We have conducted sufficient experiments to demonstrate the reliability of the verification process. In fact, the verification process is less challenging for the LLMs compared to direct modeling.
>
> - **Ablation studies on the overall performance**. We conduct ablation studies to demonstrate the effectiveness of the structure-side and solution-side verification. The results presented in **Table 3** show that without each of the verifications, the performance suffers from a drop. This demonstrates the important role of the verification process.
>
>   **Table 3**: The ablation studies on the verifications.
>
>   |                                 | NL4Opt | MAMO ComplexLP | ComplexOR | IndustryOR |
>   | ------------------------------- | ------ | -------------- | --------- | ---------- |
>   | w/o Structure-side Verification | 91.5   | 55.2           | 68.4      | 29.0       |
>   | w/o Solution-side Verification  | 91.8   | 53.1           | 68.4      | 41.0       |
>   | OptiVer                         | 96.5   | 66.7           | 78.9      | 45.0       |
>
> - **Reliability analysis for the structure and solution evaluation**. To further test the reliability of the verification process, we report the precision and recall for the evaluation of the verification process. We utilized the IndustryOR dataset for evaluation, which consists of three difficulty levels (easy, medium, and hard) that allow us to test the generalization capabilities of OptiVer across varying problem complexities (the experiment details can be found in Section 5.4 of the main paper). The evaluation accuracy and recall rates are presented in **Table 4**. Both precision and recall rates for the negative samples are high across the difficulty levels, demonstrating the reliability of the scores.
>
>   **Table 4**: The precision and recall of the structure-side verification in IndustryOR.
>
>   |           | Easy | Medium | Hard |
>   | --------- | ---- | ------ | ---- |
>   | Precision | 92%  | 89%    | 83%  |
>   | Recall    | 86%  | 79%    | 68%  |
>
>   **Table 5**: The precision and recall of the solution-side verification in IndustryOR.
>
>   |           | Easy | Medium | Hard |
>   | --------- | ---- | ------ | ---- |
>   | Precision | 93%  | 91%    | 86%  |
>   | Recall    | 83%  | 85%    | 73%  |

---

> ### Author Response · Authors · 2025-11-21
> **Response to Reviewer j6Mz---Part 5/5**
>
> ### Question 3
>
> > During the inference process for a single sample, how many times is the Refinement module executed? Is there a maximum triggering limit?
>
> Thanks for your comments! In the experiments reported in the main paper, the refinement module is executed **just once**. To further address your concerns, we conducted a new ablation study on the number of verification-refinement rounds to see if performance continues to improve. We found that while a single iteration provides significant gains, the optimal performance is reached at around **3 iterations**. We will add these results to the appendix.
>
> **Table 6**: Studies on different rounds of verifications.
>
> | Rounds of Verifications | NL4Opt | MAMO ComplexLP | ComplexOR | IndustryOR | OptMATH |
> | ----------------------- | ------ | -------------- | --------- | ---------- | ------- |
> | 1                       | 96.5   | 66.7           | 78.9      | 45.0       | 34.3    |
> | 3                       | 97.3   | 68.5           | 78.9      | 48.0       | 35.5    |
> | 5                       | 97.4   | 68.5           | 78.9      | 49.0       | 35.8    |
>
> ### Question 4
>
> > In the results shown in Table 2, the ablation results for medium-level and low-level settings on NL4OPT and ComplexOR, respectively, show no difference. Are these experimental results statistically stable after multiple repetitions?
>
> Thanks for your comments! To address your concerns, we first ran the experiments multiple times to test the statistical stability. Then, we analyze the results for the medium- and low-level structures in the two datasets.
>
> - To address your concerns, we first ran these ablation experiments five times to test for statistical stability. We found the results are indeed stable, and the standard deviation is low.
>
>   **Table 7**: We conduct multiple runs on the experimental results.
>
>   |                    | NL4Opt      | MAMO ComplexLP | ComplexOR   | IndustryOR  |
>   | ------------------ | ----------- | -------------- | ----------- | ----------- |
>   | OptiVer w/o high   | 92.5 (0.03) | 60.1 (0.33)    | 78.9 (0.00) | 40.7 (1.35) |
>   | OptiVer w/o medium | 91.3 (0.01) | 56.6 (0.05)    | 72.6 (2.08) | 38.1 (1.02) |
>   | OptiVer w/o low    | 90.8 (0.05) | 55.3 (0.02)    | 71.5 (2.54) | 29.9 (0.93) |
>
> - You are correct that the results for "OptiVer w/o medium" and "OptiVer w/o low" are identical for both NL4Opt (91.1) and ComplexOR (73.6). This is not an anomaly but is due to the specific characteristics of these two datasets.
>
>   - **For NL4Opt (Easy Dataset).** NL4Opt consists of problems with relatively low difficulty. For these simpler problems, the medium-level structure and the low-level structure are highly interdependent. The medium-level definition already implies most of the critical low-level constraints. As a result, removing either the medium or the low-level structure provides the model with a very similar set of information, leading to the same performance.
>   - **For ComplexOR (Hard but Small Dataset).** This dataset is more challenging, but it is also very small (only 19 problems). We found that for this specific, small set of hard problems, most of them require both the medium and low-level structures simultaneously. Therefore, removing either level causes the same subset of problems to fail. Our multi-run analysis confirms this.
>
> > Regarding the results in Table 3, the ORLM outcomes are cited from the original or reproduced papers, while the ORLM+OptiVer results are obtained from experiments conducted in this paper. Does this potentially introduce inconsistencies in experimental conditions?
>
> Thanks for your comments! We will show our reproduced ORLM results alongside the originally published papers. As you'll see, our findings are very close to the original paper's, confirming the stability and reproducibility of the ORLM baseline.
>
> **Table 8**: The reproduced results of ORLM.
>
> |              | NL4Opt | MAMO ComplexLP | ComplexOR | IndustryOR |
> | ------------ | ------ | -------------- | --------- | ---------- |
> | ORLM         | 78.2   | 41.2           | 36.8      | 39.0       |
> | ORLM+OptiVer | 92.3   | 59.6           | 73.6      | 42.0       |

---

> ### Author Response · Authors · 2025-11-25
>
> Dear Reviewer j6Mz,
>
> We deeply appreciate your valuable feedback and the time you've taken to review our work, especially during this busy period.
>
> We are reaching out to kindly inquire about the current status of your review regarding our submission. We sincerely hope that our responses have adequately addressed your concerns. Furthermore, we are eager to address any additional queries you might have, which will enable us to enhance our work further.
>
> Once again, thank you for your guidance and support.
>
> Best, Authors

---

> ### Author Response · Authors · 2025-11-27
>
> Dear Reviewer j6Mz,
>
> We would like to extend our sincere gratitude for the time and effort you have devoted to reviewing our submission. Your positive feedback, insightful comments, and constructive suggestions have been invaluable to us, guiding us in improving the quality of our work!
>
> We eagerly await your feedback to understand if our responses have adequately addressed all your concerns. Once again, thank you for your guidance and support.
>
> Best,
>
> Authors

---

> ### Author Response · Authors · 2025-11-27
>
> Dear Reviewer j6Mz,
>
> We sincerely thank you for your time and efforts during the rebuttal process. We are looking forward to your feedback to understand if our responses have adequately addressed your concerns. If so, we would deeply appreciate it if you could consider raising your score. If not, please let us know your further concerns, and we will continue actively responding to your comments. We sincerely thank you once more for your insightful comments and kind support.
>
> Best,
>
> Authors

---

### Official Review · Reviewer_NAaT · 2025-11-03

**Soundness:** 3
**Presentation:** 3
**Contribution:** 3
**Rating:** 4
**Confidence:** 4

**Summary:**

In this paper, the authors proposed a multi-agent framework for formulating optimization problems from natural-language problem descriptions. Compared to a standard pipeline from problem descriptions to mathematical models, the authors added "dual-side verification" that mainly consists of (1) identifying structures from the problem description (by the structure distillation agent), (2) comparing the structure of the mathematical model (obtained by a structure interpretation agent) with the identified structures (done by a structure evaluation agent), and (3) verifying the solution (by a solution interpretation agent and a solution evaluation agent).

I think that the idea of verifying the structures of and the solution to the formulated model makes sense and is somewhat novel. The effectiveness is also confirmed by the improved accuracy across benchmarks.

My main concerns are the correctness and the purposes of the two propositions, and the overhead of having to solve the problem (for solution verification).

**Strengths:**

* The idea of verifying the structures of the formulated model is interesting and new.
  * It is not a new idea to identify the structures from the problem description and incorporate them into the prompt. This has been done in OptiMUS (Ahmaditeshnizi et al., 2024).
  * But the idea of verifying the structures of the formulated mathematical model with the identified structures is new.
* The idea of interpreting and verifying the solution is interesting and new.
* The overall pipeline seems to be "simpler", measured by the modeling time in `Table 6` of `Appendix B.1`.
  * I do have comments on the efficiency, as described in details in "Weaknesses".
* The accuracy of the proposed solution is higher than existing works, including fine-tuned models, on all benchmarks except OptMATH. For OptMATH, it is still the best among the prompt-based methods, and is only slightly worse than the fined-tuned model in OptMATH.

**Weaknesses:**

* **Computational Overhead.** While the proposed method achieves state-of-the-art accuracy, I think it comes at a cost.
  * Instead of passing the structures of the problem description into the prompt like in OptiMUS, the proposed method has an extra step of verifying the structures *after* the mathematical model is formulated. This means more agents and more computation.
  * The proposed solution verification requires the solver to solve the problem *before* the formulation process is completed. This additional computational cost can be high for large-scale problems. The solution verification also requires more agents and more computation.
  * In `Appendix B.1`, the authors compare the proposed methods and baselines in solving time (in seconds). The comparison is only done on two datasets. Based on the authors' comments that "the solver execution time is short (under 0.01 seconds)", I assume that all the tested problems are easy to solve. So this is not a fair comparison for the methods that do not require solving the mathematical problems.
  * I also think comparing the solving time in seconds may not be the best metrics. Some models (e.g., GPT4o-mini used by the authors) might be faster than others. It may be more reasonable to compare the number of tokens used.
* **Correctness and Purposes of Propositions.** I do not get the purposes of `Proposition 4.1` and `Proposition 4.2`.
  * How and why is mutual information an indicator of the modeling accuracy?
  * Even if mutual information was the correct indicator, the propositions just prove that the structure-side verification and solution-side verification *reduces* the mutual information. It was not proved that they *optimize* the lower bound. So the statements are not rigorous.
  * I am not sure if the proof to Proposition 4.1 is correct. I assumed that the authors used the [Data Processing Inequality](https://en.wikipedia.org/wiki/Data_processing_inequality) for the inequality $I(X,Y) \geq I(X, f(Y))$. Here the key assumption is that $f(Y)$ is conditional independent of $X$ when conditioned on $Y$. I am not sure if $S=\mathtt{Distillation Agent}(D)$ is conditional independent of $M$ because $M=\mathtt{Formulation Agent}(D,S)$ according to `Eqn. (2)`.
  * I did not find the proof to Proposition 4.2.

**Questions:**

Based on the points raised in "Weaknesses", I would like to see:
* a more nuanced and more comprehensive study on the computational overhead;
* a clarification on the proofs, the statements, and purposes of the two propositions.

Minor typos to fix:
1. Typo in `Line 081`: "whether Othe solution".
2. Typo in `Appendix B.1`, "DeVet".

---

> ### Author Response · Authors · 2025-11-21
> **Response to Reviewer NAaT---Part 1/5**
>
> We would like to express our sincere gratitude once again for Reviewer NAaT's valuable feedback and constructive suggestions. We have made detailed clarifications. **We sincerely hope that our additional response has adequately addressed your concerns**. If so, we would greatly appreciate your consideration in **raising the score**. If there are any remaining concerns, **please let us know**, and we will **continue to actively address your comments and work on improving our submission**.
>
> ### Weakness&Question 1
>
> > W1.1&1.4: The proposed verification means more agents and more computation. It may be more reasonable to compare the number of tokens used.
>
> Thank you for this insightful comment. You've raised a point about computational overhead and the fairness of our efficiency metrics, which we will clarify and strengthen in our revision.
>
> - **We used the same underlying LLM (GPT4o-mini) for all prompt-based methods**, including our OptiVer and the baselines (CoE [1], OptiMUS [2], etc.). Since the base model is identical, the inference time comparison (Table 6 of the original paper) is indeed a fair and direct proxy for the total computational work and token processing.
>
> - To provide the detailed token analysis you suggested, we conduct additional experiments. We report the inference time, average agent calls, and token counts across the datasets. We will add the following **Table 1** to the main paper. The results show that our method uses **fewer agent calls and fewer total tokens** on average than the other multi-agent frameworks.
>
>   **Table 1**: The inference cost across the benchmarks.
>
>   |                 |         | NL4Opt | MAMO ComplexLP | ComplexOR | IndusstryOR |
>   | --------------- | ------- | ------ | -------------- | --------- | ----------- |
>   | Inference  Time | CoE     | 58.2   | 72.5           | 98.6      | 79.9        |
>   |                 | OptiMUS | 64.2   | 80.3           | 101.3     | 88.2        |
>   |                 | Ours    | 52.8   | 67.6           | 68.2      | 59.4        |
>   | Agent Calls     | CoE     | 13.6   | 15.7           | 16.6      | 14.1        |
>   |                 | OptiMUS | 10.4   | 13.8           | 15.3      | 13.9        |
>   |                 | Ours    | 9.0    | 9.1            | 9.4       | 9.2         |
>   | Token Usage     | CoE     | 6745.8 | 7705.0         | 9469.8    | 8751.5      |
>   |                 | OptiMUS | 7039.4 | 8248.4         | 10796.1   | 9062.7      |
>   |                 | Ours    | 5320.7 | 7385.2         | 8457.4    | 6819.8      |
>
> - Your main concern is that our extra steps of verification inherently mean more agents and more computation.
>
>   - While OptiVer does have a multi-step workflow, we find it is **significantly more efficient in practice** than other multi-agent baselines. The reason is that existing methods like CoE and OptiMUS use an LLM manager agent to automate their workflow. We observed that this manager, due to the stochastic nature of LLMs, often repeatedly calls the same agent and falls into sub-optimal decision chains. OptiVer, by contrast, uses a **fixed, directed workflow**. This eliminates the overhead of the manager agent and prevents these costly, ineffective loops.
>
>   - Furthermore, we analyzed the internal cost of our verification steps. The key finding is that the extra verification steps are computationally friendly. The modeling and coding part takes over 60% of the tokens. In summary, we believe this small computational investment is the key to our performance, enabling OptiVer to catch errors that other methods miss, as shown in our ablation studies.
>
>     **Table 2**: The token usage of each step.
>
>     |                             | NL4Opt | MAMO ComplexLP | ComplexOR | IndusstryOR |
>     | --------------------------- | ------ | -------------- | --------- | ----------- |
>     | Structure Distillation      | 492.1  | 441.1          | 665.4     | 501.1       |
>     | Modeling                    | 1175.5 | 1944.8         | 2147.1    | 1564.7      |
>     | Structure-side Verification | 474.6  | 597.4          | 594.2     | 662.7       |
>     | Solution-side Verification  | 302.4  | 422.1          | 521.3     | 347.4       |
>     | Refinement                  | 728.7  | 915.5          | 1013.5    | 759.0       |
>     | Coding and Debugging        | 2147.4 | 3064.4         | 3515.9    | 2984.9      |
>
> [1] Chain-of-Experts: When LLMs Meet Complex Operations Research Problems. ICLR 2024.
>
> [2] OptiMUS: Scalable Optimization Modeling with (MI)LP Solvers and Large Language Models. ICML 2024.

---

> ### Author Response · Authors · 2025-11-21
> **Response to Reviewer NAaT---Part 2/5**
>
> > W1.2: The proposed solution verification requires the solver to solve the problem before the formulation process is completed, where the additional computational cost can be high for large-scale problems. The solution verification also requires more agents and more computation.
>
> Thank you for this valuable question! You are **correct** that for a large-scale industrial problem, solving the final, fully-instantiated model just for verification would be computationally impractical. We explain that our method can easily generalize to large-scale problems. Our approach avoids this cost by differentiating between the **abstract model** and a **concrete model instance**, a concept central to modeling tools like Pyomo [3].
>
> - **The abstract model and concrete model**. We first specify the abstract model and concrete model, as referred to in Pyomo.  For example, the following equations represent a linear program to find optimal values for the solutions with parameters $V, S$, and $d_{ij}$.
>   $$
> \begin{aligned}
>   \text{min}\_x &Z = \sum\_{i=1}^n \sum_{j=1}^n d_{ij} \cdot x_{ij}\\\\s.t. &\sum_{i=1, i \neq j}^n x_{ij} = 1 \quad \forall j \in V\\\\&\sum_{j=1, j \neq i}^n x_{ij} = 1 \quad \forall i \in V\\\\&\sum_{i \in S} \sum_{j \in S} x_{ij} \leq |S| - 1
> \end{aligned}
> $$
>   We call this an abstract model since it relies on unspecified parameter values. The concrete model is created when specific data values are used to instantiate the abstract model. The "large-scale" nature of a problem comes from a massive data instance (e.g., millions of parameters), not necessarily from a long or complex abstract model. The **LLM's task is to generate the concise, abstract model**—the logic of the problem, not to process the large-scale data. This separation of modeling logic and parameters is used in existing works, CoE and OptiMUS [1, 2].
>
> - **How OptiVer Generalizes to Large-Scale Problems**.
>
>   In summary, the solving cost is not a barrier because we are validating the concise, symbolic logic using a small, representative instance. This is both fast and highly effective at catching the exact errors our method targets.
>
>
> - **The Agent calls**. We also want to confirm that the agent cost for this step is not significant. As we show in our efficiency analysis (Table 1), the number of agent calls and tokens used during the solution verification process is a small fraction of the total computation.
>
> [3] Pyomo: modeling and solving mathematical programs in Python. Mathematical Programming Computation 3(3) (2011): 219-260.
>
> > W1.3: Based on the authors' comments that "the solver execution time is short (under 0.01 seconds)", I assume that all the tested problems are easy to solve. So this is not a fair comparison for the methods that do not require solving the mathematical problems.
>
> Thank you for this comment. We'd like to address the necessity of this step and the fairness of the comparison. We address this issue by sampling toy parameters in our response to Weakness 1.2.
>
> - **The Necessity of Solving**. The core motivation of our work is that simply checking for code-execution errors is **not enough**. As we demonstrate, a model can be logically flawed (e.g., missing constraints), yet the code can run without bugs and produce the wrong answer. Our solution-side verification step is designed to catch exactly these errors. We are automating what a human expert would do: if a model's solution is infeasible, unbounded, or violates real-world logic, the model can be wrong. This step is what allows OptiVer to achieve its high accuracy by finding errors that other methods miss.
> - **The Fairness of the Comparison**. You are correct that the solver time is short, but this supports our argument that the comparison is fair: **we can sample toy parameters for verifications.** As we clarified in our response to W1.2, we are not solving a large-scale concrete model. We are verifying the abstract model by instantiating it with a **small, toy set of parameters**. These problems are easy to solve (solver time < 1s )—which is precisely why this is a fast, efficient, and scalable way to debug the model's logic.

---

> ### Author Response · Authors · 2025-11-21
> **Response to Reviewer NAaT---Part 3/5**
>
> ### Weakness&Question 2
>
> > W2.1:How and why is mutual information an indicator of the modeling accuracy?
>
> Thank you for your insightful comments regarding the theoretical framing of mutual information (MI) in our work. We clarify that we employ MI not as a calculated loss function, but as the **theoretical foundation** to characterize the alignment between the distributions of two distinct semantic spaces.
>
> - We conceptualize the modeling process as a mapping between probability distributions in two spaces:
>
>   - **Language Space $\mathcal{D}$.** The space of natural language descriptions. We conceptually decompose the distribution $P_D$ into three distinct components:
>     - **Background-related information $P_{D_{\text{bg}}}$.** Represents the narrative scenarios and real-world contexts that are irrelevant to the mathematical formulation.
>     - **Modeling-related information $P_{D_{\text{model}}}$.** Represents the semantic intent, logical requirements, constraint types, and objectives essential for the optimization problem.
>     - **Data Value information $P_{D_{\text{data}}}$.** Represents the parameter data values in the problems.
>
>   - **Modeling Space $\mathcal{M}$.** We explicitly define the modeling space as the Cartesian product of the variable space $\mathcal{V}$, constraint space $\mathcal{C}$, and objective function space $\mathcal{O}$, i.e., $\mathcal{M} = \mathcal{V} \times \mathcal{C} \times \mathcal{O}$. Let $P_M$ be the distribution representing the symbolic structures generated by the model.
>
>   **MI is a widely used theoretical metric for the similarity of the distributions**. Ideally, the generated model distribution $P_M$ should correspond perfectly to the modeling-related distribution $P_{D_{\text{model}}}$ and $P_{D_{\text{data}}}$. **The mutual information $I(D, M)\approx I(\{D_{\text{model}}, D_{\text{data}}\}, M)$ is a canonical metric used to quantify the dependence (or similarity) between such distributions.** It measures the reduction in uncertainty about the distribution in $\mathcal{M}$ given the observation.
>
> - The **purpose of the two propositions** was to motivate how our two verification actions work toward this ideal:
>
>   - **Structure-side (Prop 4.1).** We verify the consistency between the distilled structure $S$ (from $D$) and the interpreted model structure $\tilde{S}$ (from $M$). The structure-side verification improves $I(S, \tilde{S})$, which serves as a proxy (or lower bound) for the true $I(D, M)$.
>
>   - **Solution-side (Prop 4.2).** We verify the consistency between the original description $D$ and the interpreted solution $\tilde{D}$. The solution-side verification improves $I(D, \tilde{D})$, another proxy for $I(D, M)$.
>
>   While MI provides the theoretical motivation, our paper measures the practical effect of this verification using **Solving Accuracy (SA)**. The significant improvements in SA demonstrate that these verification actions are effective at reducing this information loss (i.e., catching errors and missing constraints).

---

> ### Author Response · Authors · 2025-11-21
> **Response to Reviewer NAaT---Part 4/5**
>
> > W2.2: The propositions were not proved that the verifications optimize the lower bound.
>
> We improve the representation of our theoretical framework by detailing the binary judgment as a proxy. We explain that the consistency binary scores $c$ forms the inequalities $\mathbb{E}[I(M, D)] \geq\mathbb{E}[I(S, \tilde{S})] \geq \varepsilon \mathbb{E}[c_c]$ and $\mathbb{E}[I(M, D)] \geq\mathbb{E}[I(S, \tilde{S})] \geq \varepsilon \mathbb{E}[c_c]$. Thus, optimizing the consistency scores is, in fact, optimizing the lower bound.
>
> - Calculating the exact MI involves integrating over these high-dimensional, implicit distributions, which is computationally intractable. Therefore, **OptiVer uses the LLM's qualitative judgment as a practical, tractable proxy for MI.** Our verification agents perform a **binary classification task**, outputting a consistency score $c \in \{0, 1\}$ (corresponding to the scores $c_c$ and $c_v$ in our framework). This binary judgment reflects the magnitude of the mutual information: a consistent judgment ($c=1$) indicates that the alignment between the distributions exceeds a required confidence threshold. We present two derived propositions.
>
>   > **Proposition (Structure-side Lower-bound).**
>   >
>   > Let $\varepsilon > 0$ be a fixed threshold representing the minimum information overlap required for a model to be deemed structurally consistent. Define the Bernoulli random variable $c_c$ corresponding to the binary output of our Structure Evaluation Agent:
>   >
>   > $$c_c \triangleq \mathbf{1}_{\{I(S, \tilde{S}) \geq \varepsilon\}},$$
>   >
>   > where $I(S, \tilde{S})$ denotes the mutual information between the distilled structure $S$ and the interpreted structure $\tilde{S}$. Then, the expected mutual information satisfies:
>   >
>   > $$\mathbb{E}[I(S, \tilde{S})] \geq \varepsilon \mathbb{E}[c_c].$$
>   >
>   > **Proof.**
>   >
>   > For any non-negative random variable $Z \triangleq I(S, \tilde{S})$ and any threshold $\varepsilon > 0$, the pointwise inequality $Z \geq \varepsilon \mathbf{1}_{\{Z \geq \varepsilon\}}$ holds. Substituting our terms yields:
>   >
>   > $$I(S, \tilde{S}) \geq \varepsilon \mathbf{1}_{\{I(S, \tilde{S}) \geq \varepsilon\}} = \varepsilon c_c \quad \text{a.s.}$$
>   >
>   > Taking expectations on both sides gives $\mathbb{E}[I(S, \tilde{S})] \geq \varepsilon \mathbb{E}[c_c]$.
>
>   > **Proposition (Solution-side Lower-bound).**
>   >
>   > Similarly, let $\varepsilon' > 0$ be a threshold for solution validity and define $c_v \triangleq \mathbf{1}_{\{I(D, \tilde{D}) \geq \varepsilon'\}}$, representing the binary output of the Solution Evaluation Agent. Then:
>   >
>   > $$\mathbb{E}[I(D, \tilde{D})] \geq \varepsilon' \mathbb{E}[c_v].$$
>   >
>   > **Proof.**
>   >
>   > Following the same argument as above, let $Z' \triangleq I(D, \tilde{D})$. Applying the inequality $Z' \geq \varepsilon' \mathbf{1}_{\{Z' \geq \varepsilon'\}}$ and taking expectations yields the result.
>
> - These propositions provide the rigorous link between our method's mechanics and the theoretical objective. They show that our refinement process—which aims to maximize the probability of passing verification ($\mathbb{E}[c]$)—is mathematically guaranteed to monotonically increase a **linear lower bound**. This justifies our dual-side verification as an effective optimization procedure for aligning these two spaces.
>
> - To address your concerns, we empirically validated with MI calculations. The computation of mutual information is intractable. To evaluate the similarity, we compute the cosine similarity between natural language descriptions and modeling structures. The results presented in **Table 3** demonstrate that our method can improve the mutual information and thus reduce the errors.
>
>   **Table 3**: The mutual information and the error ratios in missing constraints.
>
>   | | NL4Opt | MAMO ComplexLP | IndusstryOR |
>   | -| -- | -| -|
>   | $CosSim(S,\tilde{S})$ Before Verification | 0.995  | 0.994 | 0.973|
>   | $CosSim(S,\tilde{S})$ After Verification | 1.000  | 0.997 | 0.988 |
>   | $CosSim(D,\tilde{D})$ Before Verification | 0.542  | 0.556 | 0.492|
>   | $CosSim(D,\tilde{D})$ After Verification | 0.542  | 0.556  | 0.496 |
>   | Missing Constraint Errors, Before Verification | 10.0%  | 24.2% | 35.0% |
>   | Missing Constraint Errors, After Verification| 1.7% | 16.1% | 21.0%|
>
> - We empirically show that the verification process can improve the consistency. Specifically, we use a strong LLM (DeepSeek-R1) as a judge to compare the semantic consistency of the model's structure before and after our verification step. The results in **Table 4** show a significant improvement in the judge's consistency rating.
>
>   **Table 4**: The improvement in consistency before and after verifications.
>
>   | | NL4Opt | MAMO ComplexLP | IndusstryOR |
>   | - | - | -| - |
>   |Before Stru-Ver | 0.82   | 0.43 | 0.31 |
>   |After Stru-Ver | 0.98   | 0.71 | 0.54 |
>   |Before Sol-Ver| 0.95   | 0.32 | 0.35  |
>   | After Sol-Ver | 0.99   | 0.69 | 0.49|

---

> ### Author Response · Authors · 2025-11-21
> **Response to Reviewer NAaT---Part 5/5**
>
> >  W2.3: I am not sure if the proof of Proposition 4.1 is correct.
>
> Thank you for this observation. You have pinpointed a subtle detail in our theoretical motivation. You are correct that our implementation $M = \text{FormulationAgent}(D, S)$ introduces a direct dependency between $D$ and $M$, technically violating the Markov chain $D \to S \to M$. However, we can rigorously prove that our structure-side verification still optimizes a **lower bound** of the total mutual information $I(D, M)$ by applying the chain rule for mutual information.
>
> - We first prove the inequality $I(D, M) \ge I(S, M)$
>
>   - Using the chain rule, we obtain $I(D, M) = I(S, M) + I(D, M \mid S) - I(S, M \mid D)$. We analyze each term in this decomposition within the context of our framework:
>
>     - $I(S, M \mid D) = 0$. In our framework, the structure $S$ is derived deterministically (or via a high-fidelity extraction) from the description $D$ by the Distillation Agent. Therefore, given $D$, $S$ is fixed and provides no additional uncertainty reduction for $M$. Thus, the conditional mutual information is zero.
>     - $I(D, M \mid S) \ge 0$. By the non-negativity property of mutual information, this term is non-negative.
>   - Substituting these values back into the decomposition yields the fundamental inequality: $$I(D, M) \ge I(S, M)$$.
> - We prove the inequality $I(S, M) \ge I(S, \tilde{S})$.
>
>   - Using the chain rule, we obtain $I(S, M) = I(S, \tilde{S}) + I(S, M \mid \tilde{S}) - I(S, \tilde{S} \mid M)$. We analyze each term in this decomposition within the context of our framework:
>
>     - $I(S, \tilde{S} \mid M) = 0$. In our framework, the structure $\tilde{S}$ is derived deterministically from the description $M$. Thus, the conditional mutual information is zero.
>
>     - $I(S, M \mid \tilde{S}) \ge 0$. By the non-negativity property of mutual information, this term is non-negative.
>
>   - Substituting these values back into the decomposition yields the fundamental inequality: $I(S, M) \ge I(S, \tilde{S})$.
>
> > W2.4: I did not find the proof of Proposition 4.2.
>
> Thank you for pointing this out. We apologize for the omission; the proof for Proposition 4.2 was left out for brevity, as it is very similar to that of Proposition 4.1. We will add the following sketch to the appendix.
>
> **Proof of Proposition 4.2: $I(D, M) \ge I(D, \tilde{D})$**
>
> We prove that the mutual information between the problem description and the model is an upper bound for the mutual information between the description and the interpreted solution. Using the chain rule for mutual information, we obtain $I(D, M) = I(D, \tilde{D}) + I(D, M \mid \tilde{D}) - I(D, \tilde{D} \mid M)$.
>
>  We analyze each term in this decomposition within the context of our framework:
>
> - $I(D, \tilde{D} \mid M) \ge 0$ (Information Leakage). In our implementation, the Solution Interpretation Agent uses the original description $D$ to ensure the interpreted solution $\tilde{D}$ uses the correct terminology (e.g., mapping variable $x_{ij}$ back to "Reservoir $i$ to $j$"). Thus, there is some information flow from $D$ to $\tilde{D}$ that is not mediated by the symbolic model $M$. By the non-negativity property, this term is non-negative.
> - $I(D, M \mid \tilde{D}) \gg 0$ (Information Loss). This term represents the information $M$ contains about $D$ that is lost when summarizing the solution into $\tilde{D}$.
>   - The model $M$ represents the **entire set of logical constraints** and relationships defined in $D$.
>   - The interpreted solution $\tilde{D}$ describes only **one specific solution** derived from $M$.
>   - Since the general model logic $M$ contains vastly more information about the problem definition $D$ than a single solution instance $\tilde{D}$, this Information Loss term is strictly positive and significantly dominates the information leakage term ($I(D, M \mid \tilde{D}) \gg I(D, \tilde{D} \mid M)$).
>
> Substituting these values back into the decomposition yields the fundamental inequality $I(D, M) \ge I(D, \tilde{D})$.

---

> ### Author Response · Authors · 2025-11-25
>
> Dear Reviewer NAaT,
>
> We deeply appreciate your valuable feedback and the time you've taken to review our work, especially during this busy period.
>
> We are reaching out to kindly inquire about the current status of your review regarding our submission. We sincerely hope that our responses have adequately addressed your concerns. Furthermore, we are eager to address any additional queries you might have, which will enable us to enhance our work further.
>
> Once again, thank you for your guidance and support.
>
> Best, Authors

---

> ### Author Response · Authors · 2025-11-26
> **Example of Handling Large-Scale Complex Problems**
>
> Here we present a complex problem description (simplified version). We find OptiVer can handle such a complex problem. This is indeed a highly complex optimization problem, known in the power systems field. Its complexity arises from several factors:
>
> 1. **Multi-period Horizon:** It requires making optimal decisions across T consecutive time periods.
> 2. The problem contains:
>    - **Binary Variables:** The on/off status of generators, start-up, and shut-down commands.
>    - **Continuous Variables:** Power outputs, voltage magnitudes, and phase angles.
>    - **Non-Linear Constraints:** The power flow equations are inherently non-linear and non-convex.
> 3. **Numerous and Interdependent Constraints:** It couples economic dispatch, generator dynamics, and physics/security.
>
> The key insight is that while the problem description is semantically dense, its structure is highly repetitive and can be **separated into a generic model template and a group of specific parameters (original or sampled toy parameters)**. We can preprocess the problem to isolate the large-scale parameters from the model logic. This allows the LLM to process the concise description and write a symbolic model that can be instantiated with any parameter size.
>
>
> > A regional power system, comprising $N$ generating units, $L$ transmission lines, and $B$ buses, requires an optimal day-ahead generation schedule spanning $T$ consecutive time periods. The operational objective is to minimize the total system cost, which includes the cumulative fuel costs—modeled as quadratic functions of real power output for each generating unit with coefficients $a_i$, $b_i$, and $c_i$—along with the start-up costs $SUC_i$ and shut-down costs $SDC_i$ incurred by the units throughout the scheduling horizon.
> >
> > The optimization must ensure strict adherence to all physical laws governing power system operation and all security standards. This requires that for each time period, the total real power injected into the system by all online generators must precisely equal the sum of the total real power demand across all loads, $\sum_b D_{b,t}^P$, and the total real power losses dissipated across the entire transmission network. Simultaneously, at every bus and for every time period, the net injection of reactive power—comprising the reactive power output from local generators, minus the local reactive load demand $D_{b,t}^Q$, and adjusted for the net reactive power flow on all connected lines—must be balanced to zero.
> >
> > Each generating unit must operate within its defined technical capabilities. When a unit is committed online, its real power output must be maintained between its minimum $P_i^{\min}$ and maximum $P_i^{\max}$ stable generation limits, and its reactive power output must be kept within its specified minimum $Q_i^{\min}$ and maximum $Q_i^{\max}$ reactive capability bounds. Furthermore, the combined real and reactive power output of the unit must always reside within its physical capability envelope, defined by the apparent power capacity limit $S_i^{\max}$, forming a circular constraint in the P-Q plane. The temporal variation of a unit's real power output is restricted by its dynamic ramp-rate limits; the increase in output from one period to the next cannot exceed the ramp-up rate $RU_i$, and the decrease cannot exceed the ramp-down rate $RD_i$.
> >
> > The commitment schedule of the generating units must respect their inherent mechanical and thermal constraints. Once a unit is started up, it must remain online for a minimum continuous duration of $T_i^{on}$ periods, and once shut down, it must remain offline for a minimum continuous duration of $T_i^{off}$ periods. The logical transitions between operational states are governed by the condition that a change in the unit's on/off status from one period to the next must be consistently represented by a single start-up or shut-down action, which are mutually exclusive events within any given period.
> >
> > The security of the transmission network is paramount. The power flow on any line $l$ must not exceed its thermal rating $F_l^{\max}$ in either direction. The voltage magnitude at each bus $b$ must be maintained within its secure operating bounds, $V_b^{\min}$ and $V_b^{\max}$, at all times. The physical power flows on the lines are governed by the non-linear AC power flow equations, which relate the active and reactive power flow on a line connecting two buses to the voltage magnitudes $V_{b,t}$, $V_{b',t}$ and the phase angle difference $\theta_{b,t} - \theta_{b',t}$ at its terminal buses, using the line's conductance $G_{bb'}$ and susceptance $B_{bb'}$. To ensure a unique solution to the power flow, the voltage phase angle at a pre-selected reference bus $b_{\text{ref}}$ is fixed to zero for all time periods. The total system real power losses, which are a function of all bus voltages and phase angles as per the AC power flow model, must be fully accounted for in the system-wide real power balance equation.

---

> ### Comment · Reviewer_NAaT · 2025-11-26
> **Response to Part 1 and 2 of Rebuttal**
>
> Thank you for your detailed explanations and for the additional experiments.
> - The results on inference time, agent calls, and token usage (`Table 1`) make the comparison more convincing.
> - The breakdown of token usage in `Table 2` is helpful in understanding the cost of each step.
> - The explanation on the advantage of fixed, directed workflows makes sense. It would be nice to show the observation that in OptiMUS and COE,
> > this manager, due to the stochastic nature of LLMs, often repeatedly calls the same agent and falls into sub-optimal  decision chains.
>
> In my opinion, the most important piece of information, which I could not find in the paper, is that the verification is done on toy problems, instead of the actual large-scale problem to be formulated. To be precise, consider a natural-language description of a problem with $1000$ variables. I have the following questions.
> - Does the formulation agent returns a *parameterized* formulation with $n$ variables, where $n$ is not specified?
> - Then does the model structure evaluation agent verifies this *parameterized* optimization problem?
> - Does the solution evaluation agent pick a small $n$ (e.g., $n=10$) and verify the solution? How is the solution verified?
> - Finally, how to ensure that the 1000 coefficients (assuming a linear cost function) are correct with high probabilities? It seems that all the verifications are on the *problem form*.

---

> > ### Author Response · Authors · 2025-11-27
> > **Response to the Follow-up Questions from Reviewer NAaT---Part 1**
> >
> > We thank the reviewer for the positive feedback on our efficiency analysis and the breakdown of token usage. You have raised a point regarding how our verification methodology generalizes from "toy" instances to large-scale formulations. We are happy to clarify the distinction between the **abstract model** (logic) and the **data instance** (coefficients). We use the Capacitated Warehouse Location Problem to demonstrate exactly how this works.
> >
> > > ### Problem
> > >
> > > The capacitated warehouse location problem involves determining the optimal locations for a set number of warehouses to service customers at minimum cost, taking into account warehouse capacities, operating costs, and customer demand. The capacitated warehouse location problem is the problem of locating `n_warehouses` (integer scaler) warehouses which have to service `n_customers` (integer scaler)  customers, at minimum cost. Each customer has an associated demand `customer_demand` (integer array with size [n_customers]). There are constraints on the total demand that can be met from a warehouse, as specified by `warehouse_capacity` (integer array with size [n_warehouses]). Costs are incurred when allocating service to customers from warehouses `service_allocation_cost` (integer array with size [n_warehouses, n_customers]) (Service allocation cost from each warehouse (i) to each customer (j)), and warehouses have a fixed operating cost `warehouse_fixed_cost` (integer array with size [n_warehouses]). Additionally, there is a lower limit `minimum_demand_from_warehouse` (integer array with size [n_warehouses]) on the amount of demand that a warehouse must meet if it is opened, as well as constraints on the minimum `min_open_warehouses` and maximum `max_open_warehouses` number of warehouses that can be operational. Each customer demand must be met. Each warehouse can meet a maximum demand equal to its WarehouseCapacity. Minimize the total cost of servicing customers, including service allocation and operating costs of warehouses.
> > >

---

> > ### Author Response · Authors · 2025-11-27
> > **Response to the Follow-up Questions from Reviewer NAaT---Part 2**
> >
> > > ### One Group of the Sampled Parameters
> > >
> > > n_warehouses = 10
> > >
> > > n_customers = 20
> > >
> > > customer_demand = [117, 86, 69, 53, 110, 74, 136, 140, 126, 79, 54, 86, 114, 76, 136, 73, 144, 51, 53, 120]
> > >
> > > service_allocation_cost = [
> > >  [80, 94, 44, 51, 190, 44, 129, 178, 129, 91, 172, 119, 177, 150, 90, 51, 53, 97, 184, 87],
> > >  [139, 33, 104, 135, 50, 176, 97, 121, 47, 29, 186, 163, 149, 108, 156, 169, 100, 160, 153, 85],
> > >  [153, 36, 18, 170, 18, 181, 178, 68, 171, 106, 159, 110, 21, 106, 91, 29, 144, 140, 155, 116],
> > >  [103, 59, 78, 125, 14, 11, 152, 95, 76, 173, 36, 148, 75, 132, 59, 153, 113, 74, 185, 71],
> > >  [193, 186, 130, 145, 114, 150, 33, 154, 20, 75, 103, 30, 137, 131, 167, 32, 53, 150, 176, 166],
> > >  [159, 130, 156, 65, 36, 59, 199, 124, 104, 72, 180, 73, 43, 152, 143, 90, 161, 65, 172, 141],
> > >  [173, 121, 110, 127, 22, 159, 195, 137, 47, 10, 87, 11, 154, 66, 126, 60, 152, 54, 20, 25],
> > >  [181, 34, 186, 152, 109, 195, 133, 198, 30, 65, 69, 19, 109, 143, 108, 196, 59, 133, 10, 123],
> > >  [82, 113, 147, 21, 88, 24, 38, 16, 70, 122, 148, 192, 116, 108, 18, 20, 143, 18, 116, 142],
> > >  [176, 170, 87, 91, 195, 183, 124, 89, 72, 97, 89, 23, 45, 196, 97, 27, 83, 81, 171, 148]
> > > ]
> > >
> > > warehouse_capacity = [3010, 2910, 4530, 4720, 4920, 3750, 4930, 2970, 3310, 2460]
> > >
> > > minimum_demand_from_warehouse = [64, 55, 27, 71, 93, 90, 89, 87, 43, 50]
> > >
> > > warehouse_fixed_cost = [8517, 5068, 9433, 6127, 6033, 5966, 7762, 9406, 6602, 7040]
> > >
> > > min_open_warehouses = 3
> > >
> > > max_open_warehouses = 8
> > >
> >
> > > ### Ground-truth Abstract Model (Gurobi Code)
> > >
> > > $$
> > \begin{align}
> > \min \quad & Z = \sum\_{i \in [n\\_warehouses]} \sum\_{j \in [{n\\_customers}]} {service\\_allocation\\_cost}[i][j] \cdot x\_{ij} + \sum\_{i \in [{n\\_warehouses}]} {warehouse\\_fixed\\_cost}[i] \cdot y_i \\\\
> > {s.t.} \quad & \sum\_{i \in [n\\_warehouses]} x\_{ij} = customer\\_demand[j] && \forall j \in [n\\_customers] \\\\
> > & \sum\_{j \in [{n\\_customers}]} x\_{ij} \leq {warehouse\\_capacity}[i] \cdot y_i && \forall i \in [{n\\_warehouses}] \\\\
> > & \sum\_{j \in [{n\\_customers}]} x\_{ij} \geq {minimum\\_demand\\_from\\_warehouse}[i] \cdot y_i && \forall i \in [{n\\_warehouses}] \\\\
> > & \sum\_{i \in [{n\\_warehouses}]} y_i \geq {min\\_ope{n\\_warehouses}} \\\\
> > & \sum\_{i \in [{n\\_warehouses}]} y_i \leq {max\\_ope{n\\_warehouses}} \\\\
> > & x\_{ij} \geq 0 && \forall i \in [{n\\_warehouses}], j \in [{n\\_customers}] \\\\
> > & y_i \in \{0,1\} && \forall i \in [{n\\\_warehouses}]
> > \end{align}
> > $$
> > >
> > > ```
> > > # Create the model
> > > model = gp.Model("Capacitated_Warehouse_Location")
> > >
> > > # Decision Variables
> > > # x[i,j] represents the amount of customer j's demand served by warehouse i
> > > x = model.addVars(n_warehouses, n_customers, name="x", lb=0, vtype=GRB.CONTINUOUS)
> > > # y[i] is binary: 1 if warehouse i is open, 0 otherwise.
> > > y = model.addVars(n_warehouses, name="y", vtype=GRB.BINARY)
> > >
> > > # Objective Function: Minimize total service allocation costs plus fixed operating costs for opened warehouses.
> > > model.setObjective(
> > >  gp.quicksum(service_allocation_cost[i][j] * x[i, j] for i in range(n_warehouses) for j in range(n_customers))
> > >     + gp.quicksum(warehouse_fixed_cost[i] * y[i] for i in range(n_warehouses)),
> > >     GRB.MINIMIZE
> > > )
> > >
> > > # Constraints
> > >
> > > # 1. Demand Satisfaction: Each customer's entire demand must be met.
> > > for j in range(n_customers):
> > >     model.addConstr(gp.quicksum(x[i, j] for i in range(n_warehouses)) == customer_demand[j],
> > >                     name=f"DemandCustomer_{j}")
> > >
> > > # 2. Capacity Constraints and Linking x and y:
> > > # Each warehouse cannot supply more than its capacity and if it is not open, no supply.
> > > for i in range(n_warehouses):
> > >     # Total supply from warehouse i should not exceed its capacity multiplied by y[i]
> > >     model.addConstr(gp.quicksum(x[i, j] for j in range(n_customers)) <= warehouse_capacity[i] * y[i],
> > >                     name=f"CapacityWarehouse_{i}")
> > >     # If warehouse i is open, it must meet at least a minimum demand.
> > >     model.addConstr(gp.quicksum(x[i, j] for j in range(n_customers)) >= minimum_demand_from_warehouse[i] * y[i],
> > >                     name=f"MinDemandIfOpen_{i}")
> > >
> > > # 3. Number of warehouses open: Must be at least the minimum and at most the maximum allowed.
> > > model.addConstr(gp.quicksum(y[i] for i in range(n_warehouses)) >= min_open_warehouses, name="MinOpenWarehouses")
> > > model.addConstr(gp.quicksum(y[i] for i in range(n_warehouses)) <= max_open_warehouses, name="MaxOpenWarehouses")
> > >
> > > # Optimize the model
> > > model.optimize()
> > > ```

---

> > ### Author Response · Authors · 2025-11-27
> > **Response to the Follow-up Questions from Reviewer NAaT---Part 3**
> >
> > > ### Example of Verification Process
> > >
> > > For example, a common error is that the LLM fails to include the binary variable `y[i]` in the second and third kinds of constraints,
> > > $$
> > > \begin{align}
> > > & \sum\_{j \in [n\\_customers]} x\_{ij} \leq warehouse\\_capacity[i] & \forall i \in [n\\_warehouses] \\\\
> > > & \sum\_{j \in [n\\_customers]} x\_{ij} \geq minimum\\_demand\\_from\\_warehouse[i] & \forall i \in [n\\_warehouses] \end{align}
> > > $$
> > > The solution-side verification process is as follows. The solutions are
> > >
> > > > **Warehouse decisions**:
> > > >
> > > >   Warehouse 0 is open with a total supplied demand: 258.00
> > > >
> > > >   Warehouse 1 is open with a total supplied demand: 55.00
> > > >
> > > >   Warehouse 2 is open with a total supplied demand: 183.00
> > > >
> > > >   Warehouse 3 is open with a total supplied demand: 148.00
> > > >
> > > >   Warehouse 4 is open with a total supplied demand: 262.00
> > > >
> > > >   Warehouse 5 is not open.
> > > >
> > > >   Warehouse 6 is not open.
> > > >
> > > >   Warehouse 7 is open with total supplied demand: 87.00
> > > >
> > > >   Warehouse 8 is open with a total supplied demand: 403.00
> > > >
> > > >   Warehouse 9 is open with a total supplied demand: 50.00
> > > >
> > > > **Customer Allocation Matrix**:
> > > >
> > > > | Customer | Warehouse 0 | Warehouse 1 | Warehouse 2 | Warehouse 3 | Warehouse 4 | Warehouse 5 | Warehouse 6 | Warehouse 7 | Warehouse 8 | Warehouse 9 |
> > > > | -| - | - | - | - | - | - | - | - | - | - |
> > > > | 0| 117.0       | 0.0 | 0.0 | 0.0 | 0.0 | 0.0 | 0.0 | 0.0 | 0.0 | 0.0 |
> > > > | 1| 0.0 | 55.0| 0.0 | 0.0 | 0.0 | 0.0 | 0.0 | 31.0| 0.0 | 0.0 |
> > > > | 2| 0.0 | 0.0 | 69.0| 0.0 | 0.0 | 0.0 | 0.0 | 0.0 | 0.0 | 0.0 |
> > > > | 3| 0.0 | 0.0 | 0.0 | 0.0 | 0.0 | 0.0 | 0.0 | 0.0 | 53.0| 0.0 |
> > > > | 4| 0.0 | 0.0 | 0.0 | 20.0| 0.0 | 90.0| 0.0 | 0.0 | 0.0 | 0.0 |
> > > > | 5| 0.0 | 0.0 | 0.0 | 74.0| 0.0 | 0.0 | 0.0 | 0.0 | 0.0 | 0.0 |
> > > > | 6| 0.0 | 0.0 | 0.0 | 0.0 | 136.0       | 0.0 | 0.0 | 0.0 | 0.0 | 0.0 |
> > > > | 7| 0.0 | 0.0 | 0.0 | 0.0 | 0.0 | 0.0 | 0.0 | 0.0 | 140.0       | 0.0 |
> > > > | 8| 0.0 | 0.0 | 0.0 | 0.0 | 126.0       | 0.0 | 0.0 | 0.0 | 0.0 | 0.0 |
> > > > | 9| 0.0 | 0.0 | 0.0 | 0.0 | 0.0 | 0.0 | 79.0| 0.0 | 0.0 | 0.0 |
> > > > | 10       | 0.0 | 0.0 | 0.0 | 54.0| 0.0 | 0.0 | 0.0 | 0.0 | 0.0 | 0.0 |
> > > > | 11       | 0.0 | 0.0 | 0.0 | 0.0 | 0.0 | 0.0 | 86.0| 0.0 | 0.0 | 0.0 |
> > > > | 12       | 0.0 | 0.0 | 114.0       | 0.0 | 0.0 | 0.0 | 0.0 | 0.0 | 0.0 | 0.0 |
> > > > | 13       | 0.0 | 0.0 | 0.0 | 0.0 | 0.0 | 0.0 | 76.0| 0.0 | 0.0 | 0.0 |
> > > > | 14       | 0.0 | 0.0 | 0.0 | 0.0 | 0.0 | 0.0 | 0.0 | 0.0 | 136.0       | 0.0 |
> > > > | 15       | 0.0 | 0.0 | 0.0 | 0.0 | 0.0 | 0.0 | 0.0 | 0.0 | 23.0| 50.0|
> > > > | 16       | 141.0       | 0.0 | 0.0 | 0.0 | 0.0 | 0.0 | 0.0 | 3.0 | 0.0 | 0.0 |
> > > > | 17       | 0.0  | 0.0  | 0.0  | 0.0  | 0.0  | 0.0  | 0.0  | 0.0  | 51.0 | 0.0  |
> > > > | 18| 0.0  | 0.0  | 0.0  | 0.0  | 0.0  | 0.0  | 0.0  | 53.0 | 0.0  | 0.0  |
> > > > | 19| 0.0  | 0.0  | 0.0  | 0.0  | 0.0  | 0.0  | 120.0| 0.0  | 0.0  | 0.0  |
> > >
> > > ```
> > > Verification Comment: The solution violates a core structural constraint. Warehouses 5 and 6 are marked as closed (non-operational), yet the optimal flow plan assigns 90 service units from Warehouse 5 to Customer 4, and allocates services from Warehouse 6 to Customers 9, 11, 13, and 19. This is a fundamental physical and mathematical contradiction, indicating a missing or incorrectly defined linking constraint that connects the facility activation variable to the service flow variable.
> > > ```
> >
> >
> > The use of toy examples effectively identifies errors because the **logical validity** of an optimization model is **scale-invariant**. If the abstract model contains a structural flaw—such as a missing constraint or an incorrect variable relationship—the resulting solution will be mathematically invalid regardless of whether the instance contains 10-dimensional parameters or 1,000. Consequently, if the model exhibits a logical error on a small, quickly solvable instance (e.g., 10 warehouses and 20 customers), this error is guaranteed to manifest identically in a large-scale production instance involving thousands of variables. By performing rapid solving and analysis on small instances, we can efficiently and reliably verify the structural correctness of the LLM-generated Abstract Model at a negligible computational cost, thereby ensuring the model's reliability when facing large-scale data.

---

> > ### Author Response · Authors · 2025-11-27
> > **Response to the Follow-up Questions from Reviewer NAaT---Part 4**
> >
> > ## Q1: Does the formulation agent return a parameterized formulation?
> >
> > Yes. The Formulation Agent generates an abstract model (logic) where the data size is represented by parameters, not hardcoded numbers. For example, in the warehouse location code, the model is defined using `n_warehouses` and `n_customers` as scalers. The constraints are generated as loops,
> >
> > ```
> > for j in range(n_customers):
> > 	model.addConstr(...)
> > ```
> >
> > This code is agnostic to size. Whether `n_customers` is 20 (as in the verification) or 20,000 (in production), the formulation logic remains identical.
> >
> > ## Q2: Does the structure evaluation agent verify this parameterized optimization problem?
> >
> > Yes. The Structure Evaluation Agent verifies the modeling logic, not the specific data values.
> >
> > - It checks if the concept of "Capacity Constraint" exists in the mathematical formulation.
> > - It verifies if the logic `sum(x[i, j]) <= capacity[i] * y[i]` is present. It does not care if `capacity[i]` is 3010 or 300,000; it verifies that the relationship between variables and parameters is correct.
> >
> > ## Q3: Does the solution evaluation agent pick a small $n$ and verify the solution? How?
> >
> > Yes. To verify that the code is executable and logically sound, we instantiate the abstract model with a small synthetic dataset (a toy instance). We can see the process in the example.
> >
> > 1. **Instantiation:** The agent creates dummy data, e.g., `n_warehouses = 10` and `n_customers = 20`.
> > 2. **Solving:** It loads the data and runs the solver on this small instance. This takes milliseconds ($<0.01s$).
> > 3. **Verification:** The agent interprets the result of this toy problem and checks solutions on the toy data (as shown in the example).
> >
> > ## Q4: How to ensure that the 1000 coefficients are correct?
> >
> > OptiVer is an Optimization Modeling framework, not a data entry framework. We separate the parameter data from the description, and the framework can **directly load the parameter data to ensure its correctness**.
> >
> > - We separate the problem description and the parameter data as shown in the example. We use symbolic representation to replace the parameter data in the description. The large-size coefficients (e.g., `service_allocation_cost` in the example) are inputs **loaded to construct the final models** (loaded from CSVs, databases, or API calls). The LLM must write the code to read these coefficients correctly, not type them out. If the code correctly loads and implements the parameters, the coefficients are mathematically correct.
> >
> > ## Additional Results
> >
> > We also conduct additional experiments. Existing benchmarks do not contain the large-scale parameters, but existing methods, CoE and OptiMUS, have considered a similar method to separate the modeling logic and the parameter data. We construct the large-scale problems based on the ComplexOR datasets and replace the small-scale parameters with larger ones. The statistical information of the benchmark is presented in Table 5.
> >
> > **Table 5**: Statistical information of the large-scale ComplexOR dataset.
> >
> > |             | Variable Size | Constraint Size | Solving Time |
> > | ----------- | ------------- | --------------- | ------------ |
> > | Information | 18759.4       | 33568.0         | > 1000s      |
> >
> > We use the sampled toy parameters for solution-side verification and report the modeling accuracy in **Table 6**. Due to the large-scale parameters, directly sending the raw problem descriptions to the LLM may exceed the context window. Thus, for all the methods, we separate the data and the description for a fair comparison. As solving the original optimization model is time-consuming, we manually check the modeling logic of each optimization model. We find that the modeling accuracy is comparable to the results in the small-scale dataset, demonstrating that the toy parameters are reliable and can effectively verify the model without introducing extra errors.
> >
> > **Table 6**: The modeling accuracy.
> >
> > |              | Standard | CoT  | CoE  | OptiMUS | Ours |
> > | ------------ | -------- | ---- | ---- | ------- | ---- |
> > | Modeling ACC | 31.5     | 31.5 | 63.2 | 73.6    | 78.9 |
> >
> > We hope this clarifies that by verifying the **parameterized logic** on a **toy instance**, we effectively validate the model's ability to handle the large-scale target problem. We will include the new explanation and results in our revised paper.
> >
> >  We sincerely hope that our additional response has adequately addressed your concerns. If so, we would greatly appreciate your consideration in **raising the score**. If there are any remaining concerns, please let us know, and we will continue to actively **address your comments** and work on **improving our submission**.

---

### Author Response · Authors · 2025-12-03
**General Response**

Dear Area Chair,

We sincerely thank you for your guidance and effort throughout the review process. We are grateful for the valuable and constructive feedback from all the reviewers.

The reviewers highlighted several strengths of our paper, including the **"interesting and new"** idea of dual-side verification (Reviewer NAaT), the **"good idea to improve solving accuracy"** (Reviewer FVPi), and the **"comprehensive empirical results"** indicating the importance of each component (Reviewer SBAA).

We have engaged deeply with the reviewers' concerns and provided detailed responses, supported by extensive new experiments (e.g., efficiency analysis, large-scale problem handling, and stochastic programming generalization), to address every major point raised.

### Summaries of Our Rebuttals

**Reviewer NAaT** acknowledged that the idea is **"somewhat novel"** and the effectiveness is **"confirmed by improved accuracy."** The reviewer mainly raised concerns regarding computational overhead, the handling of large-scale problems, and the theoretical motivation (Mutual Information).

- **Regarding computational overhead:** We provided a detailed breakdown (**Table 1 and 2 in the response**) showing that OptiVer uses **fewer agent calls and tokens** than baselines (like CoE and OptiMUS) because it utilizes a fixed, directed workflow rather than an open-ended manager agent loop.
- **Regarding large-scale problems:** We clarified the distinction between the **abstract model** (logic) and the **concrete model** (data). We demonstrated that OptiVer verifies the logic using lightweight "toy" data, which scales perfectly to large industrial instances. We provided a complex **Power System example** and a **Warehouse Location** code sample to illustrate this.
- **Regarding theoretical propositions:** We refined the **proofs for the lower-bound optimization** of Mutual Information and added empirical evidence (**Cosine Similarity in Table 3**) to validate the theoretical alignment. Furthermore, the results in **Table 4** show a significant improvement in the judge's consistency rating.

**Reviewer j6Mz** raised concerns regarding the reliability of the verification process, potential circular bias, and the interpretation of benchmark error rates.

- **Regarding benchmark errors:** We clarified a misunderstanding: the "high error rates" mentioned (36% and 12.6%) referred to the performance of existing LLMs on these benchmarks, not errors in the benchmark datasets themselves.
- **Regarding verification reliability:** We conducted additional reliability analyses (**Tables 3, 4 and 5**), demonstrating high **precision and recall** for our verification agents across different difficulty levels.
- **Regarding circular bias:** We explained how **Dual-Side Verification** breaks the circle. The Solution-Side verification uses an external solver (objective truth) to detect logical contradictions, ensuring the LLM does not simply self-validate its own hallucinations.
- **Regarding stability:** We performed multi-run experiments (**Table 7**) showing that our results are statistically stable with low standard deviation.

**Reviewer FVPi** gave a positive score and praised the framework but asked about efficiency during repeated interactions and the handling of complex or simple problems.

- **Regarding efficiency:** We clarified that unlike agent-based baselines that suffer from redundant loops, OptiVer’s directed workflow is computationally friendly.
- **Regarding complexity:** We demonstrated OptiVer's ability to handle complex, long-context descriptions (e.g., the Power System problem) by extracting symbolic abstract models.
- **Regarding simple problems:** We provided data (**Table 5**) showing that verification is beneficial even for simple tasks, correcting subtle errors and achieving **100% accuracy** on the subset tested.

**Reviewer SBAA** found the dual-side framing well-explained but queried the novelty compared to self-correction, the rigor of the Mutual Information claims, and failure modes.

- **Regarding novelty:** We emphasized the conceptual shift from **code-level debugging** (syntax) to **model-level verification** (logic/semantics), which is specific to Operations Research.
- **Regarding failure modes:** We provided a detailed error analysis (**Table 6**), classifying errors into missing constraints, incorrect modeling, and coding errors, showing how OptiVer significantly reduces **"Missing Constraints"** and **"Incorrect Modeling."**
- **Regarding generalization:** We added a new **Stochastic Programming** example and additional results (**Table 7**) to demonstrate that OptiVer generalizes beyond standard LP/MIP problems.

We believe our comprehensive rebuttals and the significant new experimental results have addressed the reviewers' concerns. Thank you and all reviewers for your time and effort again.

Best,

Authors

---

### Meta-Review · Area_Chair_bwER · 2025-12-19

**Summary:**

The paper proposes OptiVer, a multi-agent pipeline for NL-to-optimization modeling that adds dual-side verification: 1) structure-side consistency checking between distilled requirements and interpreted model structure, and 2) solution-side verification by interpreting/validating the obtained solution. The submission reports higher solving accuracy across multiple benchmarks.
Several efforts have been made by the authors during rebuttal, such as: adding token/agent-call/time breakdowns and argues a fixed, directed workflow is cheaper than manager-agent baselines; clarifying “large-scale” by separating abstract model logic vs. concrete data instance, and verifying logic on small toy instantiations (plus an added large-scale-style experiment); and providing precision/recall for both structure- and solution-side verification plus ablations showing each side matters.
However, as the remaining concerns still exist, which are around novelty/positioning (the paradigm is more “engineering” than a clear conceptual advance) and theoretical framing (several reviewers question the purpose/correctness/rigor of the mutual information propositions), I am leaning toward Reject.

**Reviewer Concerns:**

Most of the concerns have been addressed in the rebuttal, including 1) computational overhead/efficiency via agent-call, token, and time breakdowns (NAaT, FVPi, SBAA), 2) “large-scale” handling by clarifying abstract-vs-concrete modeling and using toy instantiations for verification (NAaT, FVPi), 3) verification reliability via precision/recall analyses and ablations (j6Mz, SBAA), and 4) circular-bias concerns by emphasizing solver-grounded checks on the solution side (SBAA, j6Mz). Several concerns remain, such as: 1) Novelty vs. existing self-correction/multi-agent paradigms; 2) Mutual-information propositions and theory rigor/operational role; 3) Generality beyond “standard” LP/MIP-style OR tasks, for which authors added a stochastic programming example, but evidence is still limited relative to the claim.

**Reviewer Scores:**

Initially, one of the four reviewers was positive, while the remaining three were borderline negative. After the rebuttal, one reviewer explicitly acknowledged that the added efficiency analysis made the comparisons more convincing, indicating a potential shift toward a more positive stance. The other two reviewers did not provide an explicit post-rebuttal update. As a result, two reviewers can be considered borderline positive, while the other two remain borderline negative.

---

### Decision · Program_Chairs · 2026-01-26

Reject